# Two-dimensional Differential-form of Distributed Xinanjiang Model

Jianfei Zhao[1], Zhongmin Liang[1], Vijay P. Singh[2,3,4], Taiyi Wen[1], Yiming Hu[1], Binquan Li[1], Jun Wang[1]

[1]College of Hydrology and Water Resources, Hohai University, Nanjing, 210024, China
[2]Department of Biological & Agricultural Engineering, Texas A & M University, College Station, TX 77843-2117, USA
[3]Zachry Department of Civil & Environmental Engineering, Texas A & M University, College Station, TX 77843-3127, USA
[4]National Water and Energy Center, UAE University, Al Ain, P.O. Box 15551, UAE

*Correspondence to*: Zhongmin Liang (zmliang@hhu.edu.cn)

**Abstract.** The distributed hydrologic models (DHMs) evolved from lumped hydrologic models, inheriting their modeling philosophy along with persistent numerical error issues. Historically, these models tend to use established one-dimensional (1D) methods for slope concentration, which often struggle to effectively represent complex terrains. In this study, we formulated a purely differential-form of mathematical equations for the distributed Xinanjiang model, and developed a fully-coupled numerical solution framework. We also introduced two-dimensional (2D) diffusion wave equations for surface slope concentration, and derived 2D linear reservoir equations for subsurface slope concentration, to replace their 1D counterparts. This culminated in the development of a Two-dimensional Differential-form of Distributed Xinanjiang (TDD-XAJ) model. Two numerical experiments and its application in a humid watershed were conducted to demonstrate the model. Our result suggested that: (a) numerical errors in the existing distributed Xinanjiang model were significant and may be exacerbated by a potential terrain amplification effect, which could be effectively controlled by the fully-coupled numerical framework within the TDD-XAJ model; (b) the 2D slope concentration methods showed enhanced terrain capture ability, and eliminated the reliance on flow direction algorithms used in 1D methods; and (c) the TDD-XAJ model exhibited improved simulation capabilities compared to the existing model when applied in Tunxi watershed, particularly for flood volume. This study emphasizes the need to revisit DHMs which stemming from lumped hydrological models, focusing on model equations and numerical implementations, which could enhance model performance and benefit the hydrological modeling community.

## 1 Introduction

The distributed hydrological model (DHM) is a sophisticated perceptual, mathematical, and computational framework that simulates the spatial and temporal dynamics of hydrological processes across watersheds, particularly addressing the rainfall-runoff partitioning (Fatichi et al., 2016; Kampf and Burges, 2007; Loritz et al., 2021; Paniconi and Putti, 2015). By effectively accounting for climate conditions and critical zone characteristics, DHM has been widely used for hypothesis testing (Clark et al., 2016; Nippgen et al., 2015), flood forecasting (Perrini et al., 2024), future scenario projections (Ul Hassan et al., 2024) and other applications. The development of DHM benefits from advancements in observation methods,

computational techniques, and crucial insights gained from hydrological experiments (Bisht and Riley, 2019; Dongarra and Keyes, 2024; Paniconi and Putti, 2015). The equations used in DHMs can be classified into two categories: rigorous mathematical-physical equations simplified from the Navier-Stokes equations, and parameterized equations from the lumped

hydrological models. The emphasis on mathematical-physical equations underscores their theoretical grounding, while parameterized equations highlight their consistency with observations (Beven, 2002; Freeze and Harlan, 1969; Reggiani et al., 1998). This integration of both approaches expands the available equation space for DHMs by absorbing the merits of lumped hydrological models (Beven, 2002; Tran et al., 2018).

The Xinanjiang (XAJ) model (Zhao, 1992) is widely regarded as a standard hydrological model in China (Chen et al., 2023;

Zhao et al., 2023) and has been recognized worldwide (He et al., 2023; Knoben et al., 2020; Taheri et al., 2023; Wang et al., 2023) The XAJ model assumes that runoff can only occur once the soil tension water capacity is satisfied, following a saturation-excess runoff generation mechanism (Nan et al., 2024; Zhao, 1992). A distinctive feature of the XAJ model is its use of the Pareto distribution to effectively represent the spatial distribution of soil tension water capacity within a watershed in a parsimonious manner (Taheri et al., 2023). The development of the XAJ model has evolved through three distinct phases

over the past 60 years:

**Phase 1 (1963-1980):** This phase marks the initial formulation of the XAJ model. It began in 1963 when the saturation-excess runoff mechanism was first introduced (Zhao and Zhuang, 1963) and concluded in 1980 when the model was formally named the Xinanjiang model (Zhao et al., 1980). Key contributions during this phase included the establishment of a runoff generation module based on probability distribution curves and a top-down procedure that separated the total

runoff into distinct components—surface runoff and groundwater—before routing them to the channel.

**Phase 2 (1980-2002):** The second phase focused on enhancing the structure and accuracy of the XAJ model. Three major improvements were made: (a) the evapotranspiration module was refined by dividing the soil horizon into three layers, addressing the issue of underestimating evapotranspiration after prolonged dry periods; (b) interflow was introduced as a new runoff source, inspired by the progress in hillslope hydrology during the International Hydrological Decade (McGuire

et al., 2024); and (c) the original hydrograph method for slope and channel concentration was replaced with linear reservoir and lag-routing method, leading to the formation of the widely-used three-water-sources lumped XAJ model (Zhao, 1992).

**Phase 3 (2002-present):** The third phase is characterized by efforts to transition the XAJ model from a lumped to a distributed version, aligning it with other contemporary models like TOPMODEL and HBV (Beven et al., 2021; Seibert et

al., 2022). It is noteworthy that the original lumped XAJ model has undergone continuous evolution alongside (Ouyang et al., 2025). Although initial efforts began earlier (Lu et al., 1996), 2002 was a notable milestone due to Beven's alternative blueprint (Beven, 2002). This blueprint emphasized the importance of observational consistency as a core requirement of the physical-based property in DHMs, enabling the integration of parameterized equations from lumped hydrological models. The transformation involved the application of the runoff generation modules to smaller computational units (e.g.,

sub-basins or grids) and implementing distributed hydrological or hydraulic runoff concentration methods (Chen et al.,

2024; Fang et al., 2017; Liu et al., 2009; Su et al., 2003). Grid-based models, which utilize structured grids for data input and output, are easier to preprocess, implement, and visualize (Shu et al., 2024). This approach fosters connections with other disciplines and benefits from ongoing community support (Chen et al., 2023; Wu et al., 2024; Zhang et al., 2024), making grid-based distributed models more popular and a focal point of this study. The Grid-XAJ (GXAJ) model is a

representative achievement of the existing distributed Xinanjiang model (Yao et al., 2009; Yao et al., 2012).

A limitation of the current generation of the distributed XAJ model is its reliance on a one-dimensional slope concentration method. This method assumes that lateral flow characteristics that deviate from the dominant flow direction can be neglected (Hong and Mostaghimi, 1997). Consequently, one-dimensional (1D) methods for channel concentration can be adapted to represent slope concentration, ranging from simpler approaches like linear reservoir or Muskingum methods to more

complex options, such as kinematic and diffusion wave models (Clark et al., 2015; Paniconi and Putti, 2015; Todini, 2007). Typically, the 1D slope concentration method is combined with a single flow direction algorithm, such as the well-known eight-direction (D8) algorithm (O'Callaghan and Mark, 1984), to determine the dominant flow direction (Zhu et al., 2013). This practice is common in grid-based distributed hydrological models, such as WECOH (Nippgen et al., 2015) and HydroPy (Stacke and Hagemann, 2021). However, the 1D slope concentration method has long been criticized for its

inability to accurately simulate water flow in complex terrains and its failure to adequately represent the effects of microtopography (Hong and Mostaghimi, 1997; Liu et al., 2004).

Another major deficiency of the current generation of the distributed XAJ model is the numerical error issue inherited from its lumped counterparts, a common phenomenon in lumped models (Clark and Kavetski, 2010; Gupta et al., 2012). These lumped models typically rely on the difference-form of equations formulated over discrete time periods using simple

numerical techniques, such as the first-order explicit Euler method and operator splitting (Clark and Kavetski, 2010; Santos et al., 2018; Schoups et al., 2010; Woldegiorgis et al., 2023). Although more advanced methods, such as the MacCormack scheme (MacCormack, 1982), had been applied for runoff concentration, numerical errors persisted during the development of the distributed XAJ model. This ongoing issue can be attributed to the direct application of the difference-form of runoff generation equations from the lumped XAJ model. Moreover, a loosely-coupled numerical implementation framework had

been developed, where the total amount of different runoff sources was calculated separately using difference-form equations, assuming their intensities remained constant over discrete time intervals when fed into the differential-form of runoff concentration equations (Yao et al., 2012). This framework further complicated the existing numerical error problem. Such numerical errors can deteriorate model performance (Clark and Kavetski, 2010; Woldegiorgis et al., 2023), complicate parameter calibration (Kavetski and Clark, 2010), and even impact the effectiveness of physically informed machine

learning methods (Song et al., 2024). Furthermore, the effect of numerical errors tends to intensify with increased precipitation, leading to greater uncertainty in model applications (La Follette et al., 2021).

A differential-form of mathematical framework that describes the runoff generation and concentration process for the distributed XAJ model is required to address the numerical error issue, which further enables a fully-coupled numerical implementation framework. This effort is based on two key ideas. First, Zhao et al. (2023) introduced a differential-form of

lumped XAJ model, which provides ordinary differential equations (ODEs) to describe the grid-scale runoff generation process. These ODEs can be integrated with partial differential equations (PDEs) and other ODEs that represent the runoff concentration process, resulting in complete mathematical equations of the distributed XAJ model. Second, we employed Qu's strategy for handling hydrological processes involving mixed PDEs and ODEs, utilizing the finite volume method (FVM) to spatially discretize the PDEs (Qu and Duffy, 2007). This strategy resulted in a system of ODEs that can be solved in a unified manner.

The specific objectives of this study were as follows:

1. To introduce or derive 2D slope concentration methods for surface and subsurface runoff, and to compare their performance with 1D methods.

2. To evaluate and control the numerical errors arising from the loosely-coupled numerical implementation framework in the current generation of the distributed XAJ model.

3. To propose a new distributed XAJ model incorporating differential-form of equations, a fully coupled numerical implementation, and 2D slope concentration methods.

The remainder of this paper is organized as follows. Section 2 outlines the distributed XAJ model we developed. Section 3 introduces numerical experiments and model application, including the details of the test cases and study area used. Section 4 discusses the results of numerical experiments and model application. Section 5 provides the summary and conclusions of this study.

## 2 Model Theory and Development

### 2.1 General overview

Two-dimensional differential-form of distributed XAJ (TDD-XAJ) model inherits the merits of the original lumped XAJ model, incorporating two decades of efforts to transform it into a distributed model. TDD-XAJ model uses a grid-based structure, assuming meteorological forcing and model parameters are uniform within each grid but vary between grids. The key hydrological processes in the model are categorized into runoff generation, which is calculated at the grid level, and runoff concentration, which is calculated between grids. The soil column within each grid is divided into three layers for evapotranspiration calculation. Precipitation reaches the surface as net precipitation after accounting for actual evapotranspiration losses. Total runoff is calculated based on net precipitation using the saturation-excess runoff mechanism and is then partitioned into surface runoff, interflow, and groundwater runoff for further concentration. Water moves in two dimensions across the slope, with surface and subsurface runoff (interflow and groundwater) draining into the channels. The water in the channels is routed to the outlet in a one-dimensional flow pattern. Bidirectional water exchange between slope surface and channel is considered under varying hydraulic conditions. Specifically, when the channel water surface elevation exceeds that of the slope surface, water flows from the channel back onto the slope surface. The diagram of the TDD-XAJ is shown in Fig. 1.

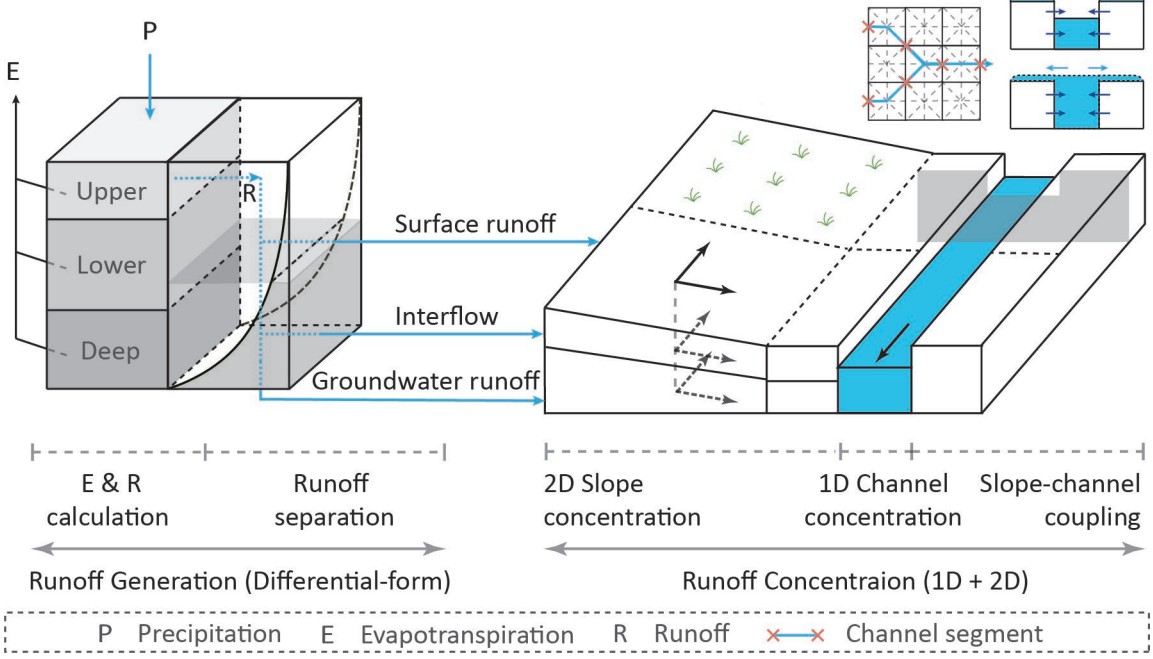

**Figure 1.** Diagram of the two-dimensional differential-form of distributed Xinanjiang model.

The square DEM cells are used as fundamental computational units with a grid size of $\Delta x$ (m). The channel is aligned either along the axis or diagonal through the center of grid cell and divided into segments corresponding to the grid cell it intersects. The river network consists of several channels, each assumed to have uniform hydraulic elements. Branching is not considered, meaning a channel may receive flow from multiple upstream channels and discharge into no more than one downstream channel. Only the length of the mainstream segment is considered, while tributary inflows are treated equivalent to source term. The length of a channel segment $\Delta l$ can be $\Delta x$, $(1 + \sqrt{2})\Delta x/2$ or $\sqrt{2}\Delta x$, based on its flow direction. The distance between two neighboring grid centers $\delta l$ is constrained to either $\Delta x$, or $\sqrt{2}\Delta x$.

There are five key differences between the TDD-XAJ model and the GXAJ model, which is a representative of the existing distributed Xinanjiang model:

1. **Revised module partitioning method**: The evapotranspiration calculation module and runoff generation module from the lumped XAJ model have been combined into a single module in the TDD-XAJ model, as they share the same state variable. This adjustment benefits mathematical description and maintains consistency with previous work.

2. **Differential-form runoff generation equations**: Unlike existing models that use difference-form equations derived with simple numerical techniques, the TDD-XAJ model adopts differential-form runoff generation equations for improved numerical accuracy.

3. **Two-dimensional diffusion wave equations**: Instead of the one-dimensional (1D) diffusion wave equations, the TDD-XAJ model employs two-dimensional (2D) diffusion wave equations to better represent surface water flow across the slope.

4. **Extended subsurface runoff concentration equations**: The 1D linear reservoir method used for subsurface runoff concentration previously has been extended to 2D in the TDD-XAJ model, allowing for better representation of subsurface flow and aligning with the surface runoff concentration method for consistency.

5. **Fully-coupled numerical implementation framework**: TDD-XAJ model uses a fully-coupled numerical approach, where the model's equations are assembled into a system of ODEs and solved simultaneously. In contrast, the GXAJ model employed a loosely-coupled method, with difference-form of equations for runoff generation and differential-form of equations for runoff concentration.

## 2.2 Mathematical structure

### 2.2.1 Evapotranspiration and runoff calculation

The actual evapotranspiration is calculated by applying the three-layer evapotranspiration (TLE) formulas of the lumped XAJ model. The TLE concept divides the soil horizon into upper, lower, and deep layers. Each layer has a tension water storage and storage capacity. Depending on the magnitude of precipitation intensity and potential evapotranspiration intensity, the tension water storages of these soil layers are replenished by precipitation or lost by evapotranspiration in a

top-to-bottom sequence. When precipitation exceeds potential evapotranspiration intensity, the surplus (net precipitation) immediately enters the upper soil layer. If the tension water storage of the upper or lower soil layer exceeds its water storage capacity, the excess is transferred to the lower or deep soil layer further. Conversely, during conditions of less precipitation, evaporative losses are sequentially deducted from the soil layers, starting from the upper to the deep layer. The governing differential equations of TLE formulas are written as:

$$\frac{d}{dt}\begin{pmatrix} W_{\mathrm{u}} \\ W_{\mathrm{l}} \\ W_{\mathrm{d}} \end{pmatrix} = \begin{pmatrix} P_{\mathrm{n}} - R - E_{\mathrm{u}} - I_{\mathrm{u}} \\ I_{\mathrm{u}} - E_{\mathrm{l}} - I_{\mathrm{l}} \\ I_{\mathrm{l}} - E_{\mathrm{d}} \end{pmatrix},\tag{1}$$

where $t$ is time (s), $W_{\mathrm{u}}$, $W_{\mathrm{l}}$, and $W_{\mathrm{d}}$ are three state variables which represents the tension water storage of the upper, lower, and deep soil layer (mm). The variables on the right-hand side of Eq. (1) are all model fluxes with units of mm s$^{-1}$, which include net precipitation intensity ($P_{\mathrm{n}}$), total runoff intensity ($R$), actual evapotranspiration intensity from the upper ($E_{\mathrm{u}}$), lower ($E_{\mathrm{l}}$), and deep ($E_{\mathrm{d}}$) soil layer, recharge intensity from the upper ($I_{\mathrm{u}}$) and lower ($I_{\mathrm{l}}$) soil layer to next soil layer when its

tension water capacity is satisfied.

The tension water storage capacity curve (TWSCC) is used to calculate $R$, which depicts the influence of the spatial heterogeneity of tension water storage capacity on the runoff generation process using the Pareto distribution. The constitutive equations of $R$, which consider the effect of the impervious area, are written as:

$$f_{\mathrm{w}} = A_{\mathrm{imp}} + \left(1 - A_{\mathrm{imp}}\right)\left[1 - \left(1 - \frac{W_{\mathrm{u}} + W_{\mathrm{l}} + W_{\mathrm{d}}}{W_{\mathrm{um}} + W_{\mathrm{lm}} + W_{\mathrm{dm}}}\right)^{b/(1+b)}\right],\tag{2}$$

$$R = P_{\mathrm{n}}f_{\mathrm{w}},\tag{3}$$

where $f_w$ denotes the runoff coefficient or the ratio of areas where tension water capacity is satisfied (-), $A_{imp}$ is the ratio of the impervious area (-), $b$ is the TWSCC exponent (-), $W_{um}$, $W_{lm}$, and $W_{dm}$ are the tension water storage capacity of three soil layers (mm).

The constitutive equations of the remainder model fluxes in Eq. (1) are given as:

$$\begin{pmatrix} P_n \\ E_n \\ E_u \\ I_u \\ E_l \\ I_l \\ E_d \end{pmatrix} = \begin{pmatrix} \max(P_{obs} - K_e E_{obs}, 0) \\ \max(K_e E_{obs} - P_{obs}, 0) \\ \max(W_u, 0) \cdot E_n \\ \max[\text{sgn}(W_u - W_{um}), 0] \cdot (P_n - R - E_u) \\ \max(W_l, 0) \cdot \max(c, W_l/W_{lm}) \cdot (E_n - E_u) \\ \max[\text{sgn}(W_l - W_{lm}), 0] \cdot (I_u - E_l) \\ \max(W_d, 0) \cdot \max[c(E_n - E_u) - E_l, 0] \end{pmatrix}, \tag{4}$$

where $P_{obs}$ and $E_{obs}$ are observed precipitation and pan evaporation intensity (mm s$^{-1}$), $K_e$ is the coefficient of potential evapotranspiration to pan evaporation (-), $E_n$ denotes the net evapotranspiration intensity (mm s$^{-1}$), $c$ is the coefficient of deep soil layer evapotranspiration (-).

### 2.2.2 Runoff separation

According to the top-down modeling philosophy of the XAJ model, the total runoff is calculated first and then separated into different runoff sources. A distinction is made between impervious and pervious areas when separating the total runoff; the total runoff on impervious areas is treated as surface runoff directly, while the total runoff on pervious areas is further divided into surface runoff, interflow, and groundwater runoff. The total amount of surface runoff is the summation of surface runoff from both the impervious and pervious areas. The separation procedure in pervious areas combines the free water storage capacity curve (FWSCC) and linear reservoir method. According to the Dunne saturation-excess runoff mechanism, it is assumed that the surface runoff only occurs when free water storage capacity is satisfied. The spatial heterogeneity of free water storage capacity is accounted for with the same Pareto distribution used in TWSCC. The interflow and groundwater runoff are assumed to outflow linearly depending on free water storage. The governing equation and constitutive equations of the runoff separation module are written as:

$$\frac{dS_0}{dt} = P_n - (R_{ps} + R_i + R_g)/(f_w - A_{imp}), \tag{5}$$

$$\begin{pmatrix} R_{ps} \\ R_i \\ R_g \end{pmatrix} = (f_w - A_{imp}) \begin{pmatrix} P_n - P_n(1 - S_0/S_m)^{ex/(1+ex)} \\ -K_i S_0 \ln(1 - K_i - K_g)/(K_i \Delta T + K_g \Delta T) \\ -K_g S_0 \ln(1 - K_i - K_g)/(K_i \Delta T + K_g \Delta T) \end{pmatrix}, \tag{6}$$

$$R_s = P_n A_{imp} + R_{ps}, \tag{7}$$

where $S_0$ is free water storage (mm), $S_m$ is free water storage capacity (mm), $R_{ps}$, $R_i$, and $R_g$ are surface runoff intensity from the pervious areas, interflow intensity, and groundwater intensity (mm s$^{-1}$), $ex$ is the FWSCC exponent (-), $K_i$ and $K_g$ are interflow and groundwater outflow coefficient (-), $\Delta T$ is the time interval of input forces (s), $R_s$ is total surface runoff intensity (mm s$^{-1}$).

### 2.2.3 Surface runoff concentration

The governing equations for the slope concentration process of the surface runoff is the two-dimensional diffusion wave equations (Gottardi and Venutelli, 2008), which are given as:

$$\frac{\partial U}{\partial t} + \frac{\partial F(U)}{\partial x} + \frac{\partial G(U)}{\partial y} = S(U),$$ (8)

$$U = \begin{bmatrix} h_s \\ 0 \\ 0 \end{bmatrix} \quad F = \begin{bmatrix} h_s u \\ gh_s^2/2 \\ 0 \end{bmatrix} \quad G = \begin{bmatrix} h_s v \\ 0 \\ gh_s^2/2 \end{bmatrix} \quad S = \begin{bmatrix} \phi_s \\ gh_s(S_{ox} - S_{fx}) \\ gh_s(S_{oy} - S_{fy}) \end{bmatrix},$$ (9)

where $h_s$ is the surface water depth (m), $u$ and $v$ are the surface flow velocity in $x$ and $y$ direction respectively (m s$^{-1}$), $g$ denotes the acceleration due to gravity (m s$^{-2}$) and is taken as 9.81, $S_{ox}$ and $S_{oy}$ are the surface bottom slope term in $x$ and $y$ direction respectively (-), $S_{ox} = -\partial z_s/\partial x$ and $S_{oy} = -\partial z_s/\partial y$, $z_s$ represents the surface elevation (m), $S_{fx}$ and $S_{fy}$ denotes the surface friction term in $x$ and $y$ direction respectively (-), $\phi_s$ is the source term (m s$^{-1}$).

The cell-centered finite volume method (FVM) is used to spatially discretize the two-dimensional (2D) diffusion wave equations based on a square structured mesh (Jain and Singh, 2005). The mesh layout and stencils used are shown in Fig. 2a.

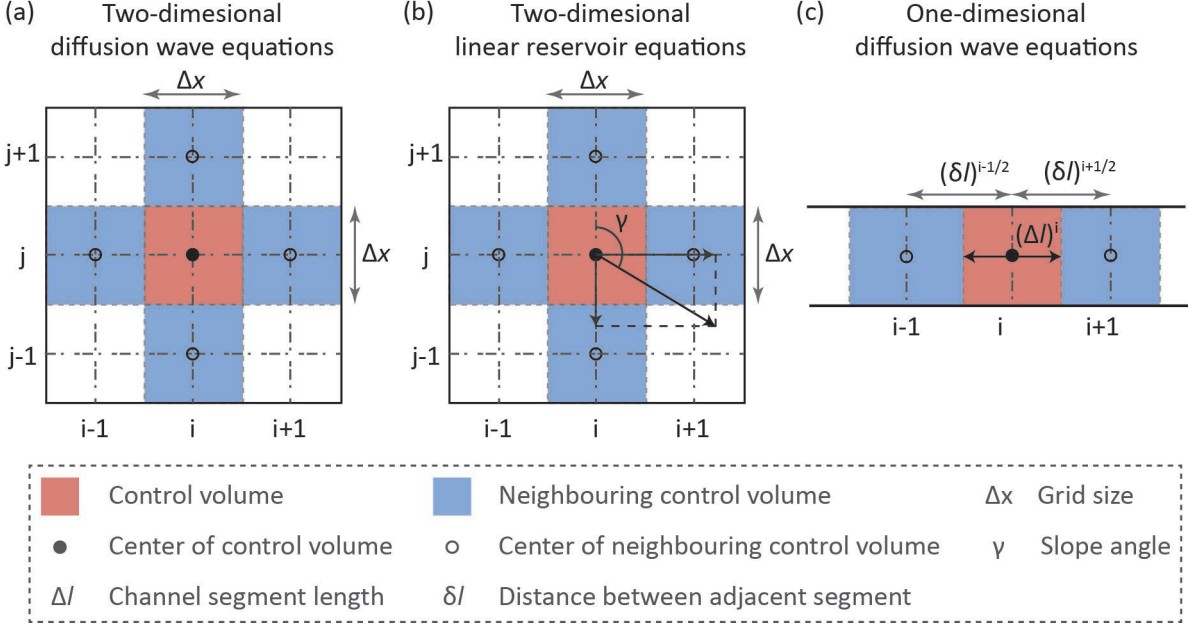

**Figure 2.** Diagram of spatial discretization for runoff concentration equations.

Within the framework of the FVM, we integrate the Eq. (8) over the control volume ( $\Omega$, red square in Fig. 2a), and introduce Green's theorem to transform the double integral over $\Omega$ into a line integral around its boundary $\Gamma$, which turns out to be:

$$\frac{\partial}{\partial t}\int_\Omega U \, d\Omega + \oint_\Gamma F(U) \cdot n \, dl + \oint_\Gamma G(U) \cdot n \, dl = \int_\Omega S(U) \, d\Omega.$$ (10)

Incorporating Eq. (9), and further assuming the surface water is evenly distributed in $\Omega$. Equation (8) could be written in the ODE form after the spatial discretization of FVM, which could be expressed as:


$$\frac{dh_s^{i,j}}{dt} = -\left(\frac{h_{s,\text{eff}}^{i-1/2,j}u^{i-1/2,j}-h_{s,\text{eff}}^{i+1/2,j}u^{i+1/2,j}}{\Delta x} + \frac{h_{s,\text{eff}}^{i,j-1/2}v^{i,j-1/2}-h_{s,\text{eff}}^{i,j+1/2}v^{i,j+1/2}}{\Delta x}\right) + \phi_s^{i,j}, \tag{11}$$

where $i$ and $j$ denotes the index of $\Omega$, $h_{s,\text{eff}}^{i+1/2,j}$ and $h_{s,\text{eff}}^{i,j+1/2}$ are the effective water depth at the east and north boundary of $\Omega$ (m), $u^{i+1/2,j}$ and $v^{i,j+1/2}$ are the surface flow velocity in the direction of the external normal vector at the eastern and northern boundary of $\Omega$ (m s$^{-1}$), $\Delta x$ is the grid size (m). $\phi_s$ including total surface runoff and interaction discharge between surface and channel $Q_{sc}$ (m$^3$ s$^{-1}$), $\phi_s = R_s/1000 - Q_{sc}/\Delta x^2$. The effective water depth at the boundary is determined

according to the water depth and surface elevation on the left and right sides of the boundary (Bates et al., 2010), and the equations are written as:

$$h_{s,\text{eff}}^{i+1/2,j} = \max(\eta_s^{i,j},\eta_s^{i+1,j}) - \max(z_s^{i,j},z_s^{i+1,j}), \tag{12}$$

$$h_{s,\text{eff}}^{i,j+1/2} = \max(\eta_s^{i,j},\eta_s^{i,j+1}) - \max(z_s^{i,j},z_s^{i,j+1}), \tag{13}$$

where $\eta_s$ denotes water surface elevation (m), $\eta_s = h_s + z_s$. The friction term $S_{fx}$ and $S_{fy}$ in Eq. (9) are evaluated with

Manning's equation (Jain and Singh, 2005), and the equations of flow velocity at the boundary of $\Omega$ could be given as:

$$u^{i+1/2,j} = -\text{sgn}(\eta_s^{i+1,j} - \eta_s^{i,j})\frac{2}{n_s^{i+1,j}+n_s^{i,j}}\left(h_s^{i+1/2,j}\right)^{2/3}\left|\frac{\eta_s^{i+1,j}-\eta_s^{i,j}}{\Delta x}\right|^{1/2}, \tag{14}$$

$$v^{i,j+1/2} = -\text{sgn}(\eta_s^{i,j+1} - \eta_s^{i,j})\frac{2}{n_s^{i,j+1}+n_s^{i,j}}\left(h_s^{i,j+1/2}\right)^{2/3}\left|\frac{\eta_s^{i,j+1}-\eta_s^{i,j}}{\Delta x}\right|^{1/2}, \tag{15}$$

where $n_s$ is surface roughness coefficient (s m$^{-1/3}$). Wall boundary conditions are applied, indicating that no surface runoff can flow through the simulation domain.

**2.2.4 Subsurface runoff concentration**

The subsurface runoff, which includes interflow and groundwater runoff, is routed with 2D linear reservoir equations. The governing equations of the original one-dimensional (1D) linear reservoir equations at grid scale are given as:

$$\frac{d}{dt}\begin{pmatrix}o_i^{i,j}\\o_g^{i,j}\end{pmatrix} = \begin{pmatrix}Q_{i,\text{up}}^{i,j}-Q_i^{i,j}+\phi_i^{i,j}\\Q_{g,\text{up}}^{i,j}-Q_g^{i,j}+\phi_g^{i,j}\end{pmatrix}, \tag{16}$$

$$\begin{pmatrix}Q_i^{i,j}\\Q_g^{i,j}\end{pmatrix} = \begin{pmatrix}-o_i^{i,j}\ln(c_i^{i,j})/\Delta T\\-o_g^{i,j}\ln(c_g^{i,j})/\Delta T\end{pmatrix}, \tag{17}$$

where $O_i$ and $O_g$ are the amount of interflow and groundwater storage respectively (mm), $Q_i$ and $Q_g$ are the outflow intensity of the interflow and groundwater storage separately (mm s$^{-1}$), $Q_{i,\text{up}}$ and $Q_{g,\text{up}}$ are the summation of $Q_i$ and $Q_g$ from multiple upstream grids (mm s$^{-1}$), $C_i$ and $C_g$ are interflow and groundwater storage recession coefficient, $\phi_i$ and $\phi_g$ are source term of interflow and groundwater storage respectively (mm s$^{-1}$), which including the corresponding generated runoff and discharge into the channel, $\phi_i = R_i - \varepsilon_i Q_i$, $\phi_g = R_g - \varepsilon_g Q_g$. When a channel is present in the grid, all outflow of the interflow and

groundwater storage are assumed to drain into the channel, resulting in both $\varepsilon_i$ and $\varepsilon_g$ being equal to 1. In the no-channel case, $\varepsilon_i$ and $\varepsilon_g$ are set to 0.

To extend the linear reservoir equations from 1D to 2D, we follow the approach outlined by Liu et al. (2004). This involves decomposing the outflow intensity derived from the 1D method into $x$- and $y$-directional components based on the actual flow direction (see Fig. 2b). The state variables are then updated within the FVM framework. For clarity and simplicity, we will illustrate this extension using interflow, given the similarity between the formulas for interflow and groundwater runoff concentration. The $x$- and $y$-directional components of $Q_i$, which maintain the total amount, could be expressed as:

$$\begin{pmatrix} Q_{i,x}^{i,j} \\ Q_{i,y}^{i,j} \end{pmatrix} = \begin{pmatrix} \left(1 - \varepsilon_i^{i,j}\right) Q_i^{i,j} \sin\gamma^{i,j} / (\sin\gamma^{i,j} + \cos\gamma^{i,j}) \\ \left(1 - \varepsilon_i^{i,j}\right) Q_i^{i,j} \cos\gamma^{i,j} / (\sin\gamma^{i,j} + \cos\gamma^{i,j}) \end{pmatrix}, \tag{18}$$

where $Q_{i,x}$ and $Q_{i,y}$ are the $x$- and $y$-directional components of $Q_i$ (mm s$^{-1}$), $\gamma$ is the slope aspect and calculated clockwise with 0 in the positive direction of the $y$-axis, $0° \le \gamma < 360°$. The governing equation of the 2D linear reservoir method is given as:

$$\frac{\mathrm{d}o_i^{i,j}}{\mathrm{d}t} = -F_i^{i+1/2,j} + F_i^{i-1/2,j} - F_i^{i,j+1/2} + F_i^{i,j-1/2} + \phi_i^{i,j}, \tag{19}$$

where $F_i^{i+1/2,j}$ and $F_i^{i,j+1/2}$ denote the interflow intensity (mm s$^{-1}$) in the direction of the external normal vector at the eastern and northern boundary of the control volume. The equations of $F_i^{i+1/2,j}$ and $F_i^{i,j+1/2}$ could be expressed as:

$$\begin{pmatrix} F_i^{i+1/2,j} \\ F_i^{i,j+1/2} \end{pmatrix} = \begin{pmatrix} \max\left(Q_{i,x}^{i,j}, 0\right) - \min\left(Q_{i,x}^{i+1,j}, 0\right) \\ \max\left(Q_{i,y}^{i,j}, 0\right) - \min\left(Q_{i,y}^{i,j+1}, 0\right) \end{pmatrix}. \tag{20}$$

The same wall boundary conditions are applied to Eq. (19), which means no subsurface runoff can flow through the simulation domain.

### 2.2.5 Channel concentration

The water within the channel is routed to the watershed outlet by using one-dimensional diffusion wave equations (Kazezyılmaz-Alhan and Medina, 2007), and the governing equations are as follows:

$$\frac{\partial U}{\partial t} + \frac{\partial F(U)}{\partial x} = S(U), \tag{21}$$

$$U = \begin{bmatrix} h_c \\ 0 \end{bmatrix} \quad F = \begin{bmatrix} h_c w \\ g h_c^2/2 \end{bmatrix} \quad S = \begin{bmatrix} \phi_c \\ g h_c (S_{oc} - S_{fc}) \end{bmatrix}, \tag{22}$$

where $h_c$ is channel water depth (m), $w$ is channel flow velocity (m s$^{-1}$), $S_{oc}$ is the channel bottom slope term (-), $S_{oc} = -\partial z_c / \partial x$, $z_c$ is the channel bottom elevation (m), $S_{fc}$ is the channel friction term (-), $\phi_c$ is the source term of channel (m s$^{-1}$), including interflow recharge, groundwater recharge, and interaction discharge between slope surface and channel.

The cell-centered FVM is used to discretize the 1D diffusion wave equations. The control volumes are constructed based on channel segment, with $h_c$ positioned at the center of the control volumes and $w$ assigned along the boundaries. The mesh layout and stencils used are shown in Fig. 2c. The spatial discretization form of Eq. (21) and (22) could be expressed as:

$$\frac{\mathrm{d}h_c^i}{\mathrm{d}t} = -\frac{1}{A_c^i}\left[Q_{c,up}^i - Q_c^i + \phi_c^i\right], \tag{23}$$

where $i$ is the index of the channel control volume (-), $A_c^i$ is open water surface area of the channel control volume (m$^2$), $A_c^i = [B(h_{c,eff}^{i-1/2}) + B(h_{c,eff}^{i+1/2})](\Delta l)^i/2$, $B(\cdot)$ is the formula of water surface width (m), $(\Delta l)^i$ denotes the length of the channel control volume (m), $Q_c^i$ is the channel discharge in the direction of the external normal vector at the eastern boundary of the channel control volume (m$^3$ s$^{-1}$). $Q_{c,up}$ is the summation of $Q_c$ from multiple upstream channel segments. The formula of $Q_c^i$ is derived based on Manning's equation, which is written as:

$$Q_c^i = -\text{sgn}(\eta_c^{i+1} - \eta_c^i)\frac{2}{n_c^{i+1}+n_c^i}\frac{A(h_{c,eff}^{i+1/2})^{5/3}}{\chi(h_{c,eff}^{i+1/2})^{2/3}}\left|\frac{\eta_c^{i+1}-\eta_c^i}{(\delta l)^{i+1/2}}\right|^{1/2}, \tag{24}$$

where $A(\cdot)$ denote the formula of cross-sectional area (m$^2$), $\chi(\cdot)$ is the formula of wetted perimeter (m), $\eta_c$ is the elevation of channel water surface elevation (m), $\eta_c = h_c + z_c$, $n_c$ is channel roughness coefficient (s m$^{-1/3}$), $(\delta l)^{i+1/2}$ is the distance between the center of the $i$-th and $i+1$-th channel control volume (m). The cross-section is generalized into a trapezoid, and the formulas of cross-sectional hydraulic elements including $A$, $B$, and $\chi$ can be derived accordingly (see Eq. S1, S2, and S3 in the Supplement). The effective water depth at the east boundary of the channel control volume $h_{c,eff}$ is evaluated with the upwind scheme, which could be expressed as:

$$h_{c,eff}^{i+1/2} = \begin{cases} h_c^{i+1} & \eta_c^{i+1} > \eta_c^i \\ h_c^i & \eta_c^{i+1} \le \eta_c^i. \end{cases} \tag{25}$$

The exchange discharge between the slope surface and the channel is calculated based on their water surface elevation and Manning's equation (Shen and Phanikumar, 2010), which could be expressed as:

$$Q_{sc} = \text{sgn}(h_s - h_{c,eff})\frac{\max(h_s, h_{c,eff})^{5/3}\Delta l}{n_s}\left|\frac{h_s-h_{c,eff}}{\Delta x/2}\right|^{1/2}, \tag{26}$$

where $Q_{sc}$ is exchange discharge between the slope surface and the channel (m$^3$ s$^{-1}$), $h_{c,eff}$ is the effective channel water depth (m) which excess the channel bank elevation $z_{bank}$ (m), $h_{c,eff} = \max(\eta_c - z_{bank}, 0)$. The source term in Eq. (23) turns out to be:

$$\phi_c = Q_{sc} + (Q_i + Q_g)\Delta x^2/1000. \tag{27}$$

The upstream boundary of the headwater channel is the wall boundary condition, while the downstream boundary of the channel at the watershed outlet is the zero-depth gradient (ZDG) condition (Panday and Huyakorn, 2004). The equation of the ZDG condition is written as:

$$Q_{ZDG} = \frac{1}{n_c^{down}}\frac{A(h_c^{down})^{5/3}}{\chi(h_c^{down})^{2/3}}\left(\frac{2h_c^{down}}{\Delta l^{down}} + S_{oc}^{down}\right)^{1/2}, \tag{28}$$

where $Q_{ZDG}$ is the outflow discharge at the watershed outlet (m$^3$ s$^{-1}$), and the superscript down refers to the information stored at the channel control volume where the watershed outlet is located.

## 2.3 Numerical implementation

In the GXAJ model, the difference-form equations for runoff generation from the original lumped XAJ model are directly applied at the grid scale. These equations are derived based on the time interval of input force, denoted as $\Delta T$. However, owing to the Courant-Friedrichs-Lewy (CFL) stability criterion, the model time step $\Delta t$ used for runoff concentration module is usually smaller than $\Delta T$. To address this discrepancy, a double-layer time loop strategy (DTLS) is adopted.

The DTLS consists of an outer-layer time loop that iterates forcing data and an inner-layer time loop that determines $\Delta t$ for advancing the solution in time. Initially, the total simulation time $T$ is read to define the duration of the outer-layer time loop. During this loop, forcing data is processed in chronological order. At the start of each outer-layer time loop, the forcing data and $\Delta T$ are loaded. The total amounts of three runoff sources (surface runoff, interflow, and groundwater runoff) are calculated using the difference-form of the runoff generation equations. These amounts are then averaged to determine the intensities for three runoff sources based on $\Delta T$. Following this, the inner-layer time loop begins, with $\Delta T$ serving as its time duration. The value of $\Delta t$ is constrained by the CFL conditions and a user-defined maximum timestep length (e.g., 10 minutes). The runoff concentration module is advanced in each inner time loop. To ensure that the end time of the last inner loop aligns with the start time of the next outer-layer iteration, $\Delta t$ may be shortened as necessary. The solving process concludes once the total simulation time $T$ is reached.

Given that the runoff generation and concentration processes are calculated separately and using equations in different forms, the numerical implementation is referred to as a loosely-coupled numerical implementation framework. The diagram of this framework is shown in Fig. 3a.

The mathematical equations of TDD-XAJ, after spatial discretization, form a set of ODEs. All state variables are assembled into a vector and are advanced simultaneously within a single time step to determine their values at the next time point. The equation of the coupled ODEs is written as:

$$\frac{\mathrm{d}\boldsymbol{Y}}{\mathrm{d}t} = \boldsymbol{F}(\boldsymbol{Y}, t), \tag{29}$$

where $\boldsymbol{Y}$ is the vector of model state variables, $\boldsymbol{Y} = [W_\mathrm{u} \quad W_\mathrm{l} \quad W_\mathrm{d} \quad S_0 \quad O_\mathrm{i} \quad O_\mathrm{g} \quad h_\mathrm{s} \quad h_\mathrm{c}]^\mathsf{T}$, $\boldsymbol{F}$ is the function vector which represent right-hand side of ODEs, which is a combination of Eq. (1), (5), (11), (19), and (23). The first seven terms of $\boldsymbol{Y}$ are used to store the state variables of the slope, and the number of elements for each term corresponds to the total number of valid DEM grids. In contrast, the last term of $\boldsymbol{Y}$ stores the state variable related to the channel, and thus the number of elements for it matches the number of DEM grids occupied by river system.

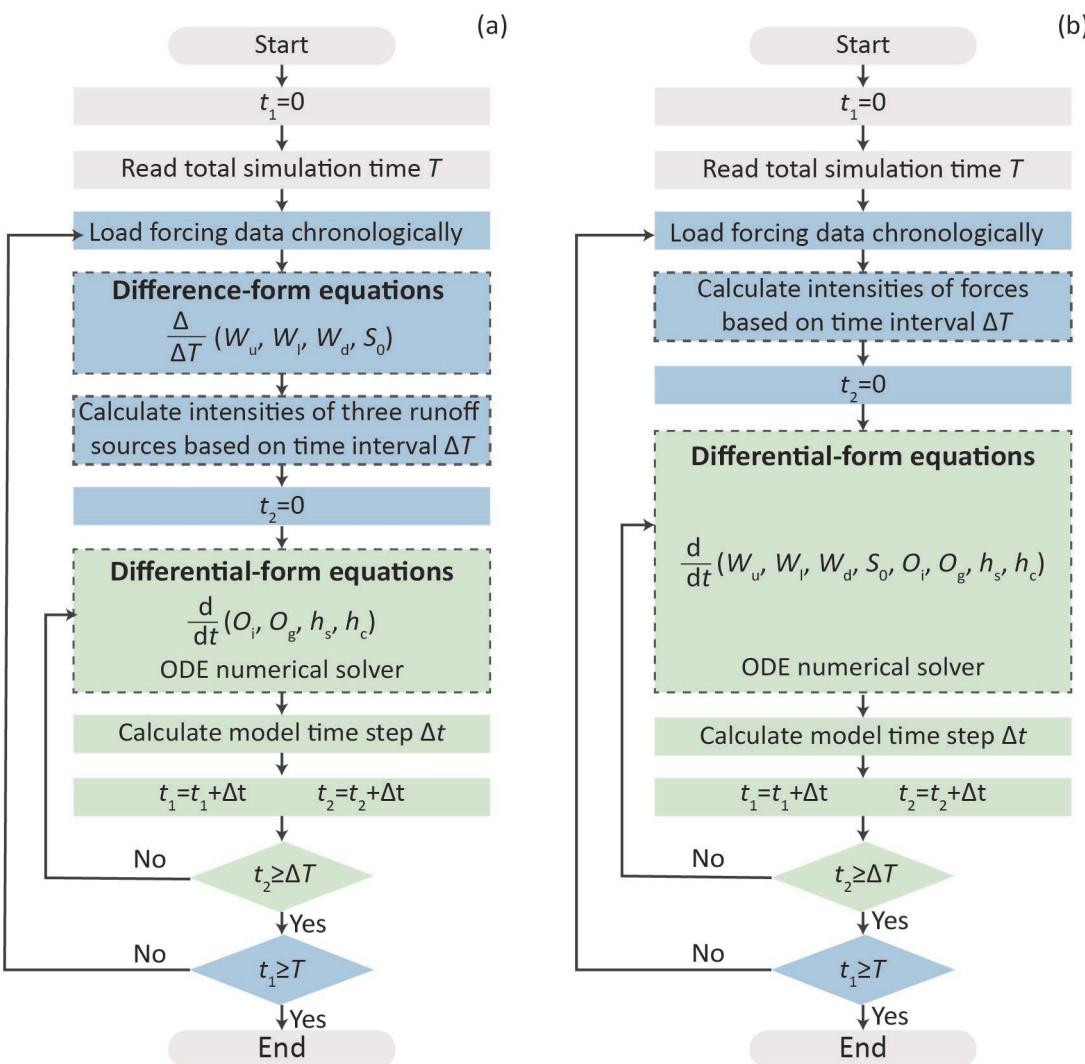

**Figure 3.** Diagram of the loosely-coupled (a) and fully-coupled (b) numerical implementation framework. $t_1$ and $t_2$ are two variables to store time. $W_u$, $W_l$, $W_d$, and $S_0$ are state variables related to runoff generation process, while $O_i$, $O_g$, $h_s$, and $h_c$ are state variables related to runoff concentration process. The differences between the two frameworks are highlighted with dash boxes.

This global coupling approach offers high numerical accuracy and flexibility for future expansion, while ensuring conservation and temporal continuity of all state variables (Qu and Duffy, 2007; Shu et al., 2020). Most importantly, it enables a fully-coupled numerical implementation framework (see Fig. 3b). The DLTS are also used in this framework to address the mismatch between $\Delta T$ and $\Delta t$. The main differences between the fully- and loosely-coupled numerical implementation framework are as follow: (a) the total amount of input forces, rather than runoff sources, is averaged in the outer time loop, and (b) both the runoff generation and concentration processes are calculated together using the differential-form of equations.

The ODEs in both fully- and loosely-coupled numerical implementation frameworks are solved with the explicit Heun scheme (Clark and Kavetski, 2010). The combined restriction of CFL conditions for surface and channel flow is written as:

$$\Delta t_{\max} = \alpha \min\left[\Delta x / \left(\sqrt{(u^2 + v^2)}\right)_{\max}, \Delta l / w_{\max}\right], \tag{30}$$

where $\Delta t_{\max}$ denotes the maximum timestep without further user-defined or programmatic constrain (s), $\alpha$ is the CFL condition coefficient (-), the rest subscript max denotes the maximum flow velocity across all corresponding elements.

## 2.4 Model Parameters

The TDD-XAJ model has 15 tunable parameters, which is one more than in the original lumped Xinanjiang model (Zhao et al., 2023). The signification and value range of these parameters are listed in Table 1. Parameters that can be spatially uniform are consistent with those from the original lumped model. The methods for determining spatially distributed parameters are provided in Sect. S2 of the Supplement.

**Table 1.** Parameters and parameter range of the TDD-XAJ model.

| Components | Symbol | Signification | Range | Spatially | Unit |
|---|---|---|---|---|---|
| Evapotranspiration and runoff calculation | $K_e$ | Coefficient of potential evapotranspiration to pan evaporation | [0.6, 1.5] | Uniform | - |
| | $c$ | Coefficient of deep soil layer evapotranspiration | [0.01, 0.2] | Uniform/ Distributed | - |
| | $W_{um}$ | Tension water storage capacity of upper soil layer | [5, 30] | Uniform/ Distributed | mm |
| | $W_{lm}$ | Tension water storage capacity of lower soil layer | [60, 90] | Uniform/ Distributed | mm |
| | $W_{dm}$ | Tension water storage capacity of deep soil layer | [15, 60] | Uniform/ Distributed | mm |
| | $A_{imp}$ | The ratio of the impervious area | [0.01, 0.2] | Uniform/ Distributed | - |
| | $b$ | Tension water storage capacity curve exponent | [0.1, 0.4] | Uniform/ Distributed | - |
| Runoff separation | $S_m$ | Free water storage capacity | [10, 50] | Uniform/ Distributed | mm |
| | $ex$ | Free water storage capacity curve exponent | [1.0, 1.5] | Uniform/ Distributed | - |
| | $K_i$ | Interflow outflow coefficient | [0.10, 0.55] | Uniform/ Distributed | - |
| | $K_g$ | Groundwater outflow coefficient | $0.7 - K_i$ | Uniform/ Distributed | - |
| Slope concentration | $n_s$ | Surface roughness coefficient | [0.01, 0.80] | Distributed | m s$^{-1/3}$ |
| | $C_i$ | Interflow storage recession coefficient | [0.5, 0.9] | Uniform/ Distributed | - |
| | $C_g$ | Groundwater storage recession coefficient | [0.98, 0.998] | Uniform/ Distributed | - |
| Channel concentration | $n_c$ | Channel roughness coefficient | [0.01, 0.05] | Distributed | m s$^{-1/3}$ |

## 3 Numerical Experiments and Model Application

Two numerical experiments were conducted to evaluate the TDD-XAJ model, which was further applied in a typical humid watershed. The experiments compared 1D and 2D slope concentration methods, as well as loosely- and fully-coupled numerical implementation frameworks. These experiments were designed to demonstrate the theoretical effectiveness of the TDD-XAJ model, while the application was designed to assess the model's performance in real-world watersheds.

### 3.1 Numerical experiments

###### 360 3.1.1 Slope concentration methods comparison experiment

The performance of the 1D and 2D diffusion wave equations for surface runoff concentration, and the 1D and 2D linear reservoir equations for subsurface runoff concentration, was compared separately using two test cases. A total of 8 simulations were conducted, categorized into surface and subsurface slope concentration comparison scenarios. The models used differ only in their slope concentration methods. The evapotranspiration and runoff calculation module, as well as the

365 runoff separation module, were disabled. The subsurface slope concentration module was turned off in the surface slope concentration comparison scenario and vice versa. The same 1D diffusion wave method was applied for channel concentration across all simulations.

The synthetic V-catchment, first proposed by Overton and Brakensiek (1970), is commonly used to verify runoff concentration components in the hydrological modeling community (Kollet et al., 2017; Maxwell et al., 2014; Shen and

370 Phanikumar, 2010; Shu et al., 2020). It consists of two symmetric hillslopes with a channel in between. Each hillslope has a length of 1000 m and a width of 800 m, with a surface roughness coefficient $n_s$ of 0.015 s m$^{-1/3}$. The length and bottom slope of the channel $S_{oc}$ are 1,000 m and 0.02, respectively, and the channel roughness coefficient $n_c$ is set to be 0.015 s m$^{-1/3}$. The channel has a uniform square cross-section, with a width measuring 20 m. This study used two test cases based on the synthetic V-catchment: the single- and double-slope case (Fig. 4), differentiated by the hillslope gradient along the $y$-

375 direction $S_{oy}$ (parallel to the channel). Both cases have an $x$-directional gradient $S_{ox}$ (perpendicular to the channel) of 0.05, while the $S_{oy}$ is 0 for the single-slope case and 0.02 for the double-slope case.

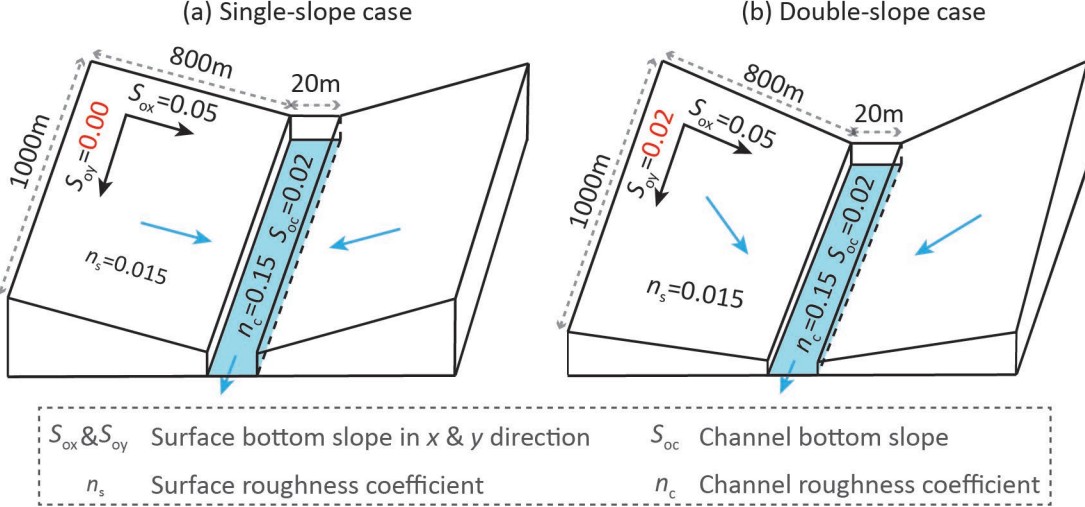

**Figure 4.** Diagram of single-slope (a) and double-slope (b) synthetic V-catchment test case.

In both test cases, precipitation lasted 180 minutes, with constant intensity during the first 90 minutes, followed by 90

minutes without precipitation, totaling 16.2 mm. Precipitation was uniformly distributed over the hillslopes, and the channel received no direct precipitation. Evapotranspiration was set to zero. Only the left-side hillslope was modeled considering the

symmetry, and the hillslope was discretized into a structured grid with 5 m×5 m cells, totaling 32000 cells, and the channel was discretized into 5 m long segments. The initial water depth of the hillslope and channel were set to zero. The wall boundary condition is applied to the hillslope, except for the boundary connected to the channel, allowing water to drain into the channel. The upstream boundary of the channel was set as the wall boundary condition, while the ZDG boundary condition was applied to the downstream outlet.

In the surface slope concentration comparison scenario, all precipitation was assumed to contribute to surface runoff with no losses. The simulated results using 1D and 2D diffusion wave equations, were compared against analytical solutions and previous published results, respectively. The single-slope case has an analytical solution for hillslope and channel outflow, with a hillslope outflow duration of 180 minutes and a channel outflow duration of 90 minutes (Di Giammarco et al., 1996). No analytical solution exists for the double-slope case, so the results of the Integrated Finite Difference Model (IFDM) were used as benchmarks (Di Giammarco et al., 1996). The Nash-Sutcliffe efficiency coefficient (NSE) (Nash and Sutcliffe, 1970) was used to assess the consistency of the simulation results with these benchmarks.

In the subsurface slope concentration comparison scenario, all precipitation was assumed to be transformed into interflow. The 1D and 2D linear reservoir equations are used to represent the interflow and groundwater runoff slope concentration process, with interflow serving as the example in this experiment. The interflow storage recession coefficient $C_i$ was set to 0.5, meaning 50% of the interflow reservoir storage would remain after an hour if no additional interflow entered.

### 3.1.2 Numerical implementation framework comparison experiment

The second numerical experiment aimed to investigate the effect of the numerical implementation framework on simulation results. It was conducted on the same synthetic V-catchment test cases as the first numerical experiment. The runoff calculation, runoff separation, surface slope concentration, and channel concentration process are considered. Two different models emerged from the use of the loosely- or fully-coupled numerical implementation frameworks (see Sect. 2.3 for details), referred to hereinafter as the loosely- and the fully-coupled model.

The model parameters used in this experiment are presented in Table 1. Five hundred sets of parameters were generated using the Symmetric Latin Hypercube (SLH) method (Gong et al., 2015), accounting for potential parameter variability. The parameters for hillslope and channel concentration were consistent with the previous experiments. The model parameters $K_e$, $A_{imp}$, $K_i$, $K_g$, $C_i$, and $C_g$ were set to 0, ensuring that only surface runoff was involved and the derivation of the corresponding analytical solution. The total amounts of simulated surface runoff during the entire simulation using both loosely- and fully-coupled models were evaluated with mean absolute error (MAE) (Hodson, 2022) against the analytical value. The equation of this analytical value (Zhao et al., 2023) could be given as:

$$R_s^* = P - \frac{S_{mm}}{1+ex}[1 - (1 - P/S_{mm})^{1+ex}] - \frac{W_{mm}}{1+b}[1 - (1 - P/W_{mm})^{1+b}] + \frac{W_{mm}}{1+ex+b}[1 - (1 - P/W_{mm})^{1+ex+b}], \quad (31)$$

where $R_s^*$ is the total surface runoff amount during the entire simulation (mm), $P$ is the total amount of precipitation (mm), $W_{mm}$ and $S_{mm}$ are maximum single-point tension and free water storage capacity respectively (mm), $W_{mm} = W_m(1 + b)/$

$(1 - A_{\text{imp}})$, $S_{\text{mm}} = S_{\text{m}}(1 + ex)$. The parameter $S_{\text{m}}$ in each parameter set was adjusted to ensure $W_{\text{mm}} = S_{\text{mm}}$, a precondition to be satisfied for the derivation of the analytical value of $R_s^*$. The $R_s^*$ for each parameter set is fixed at 16.2 mm for ease of comparison, so the value of $P$ associated with each parameter set is determined through a reverse calculation of Eq. (31). However, there is no analytical solution for hillslope or channel outflow, making direct comparison challenging. To address this, we evaluated the convergence of the loosely-coupled model by progressively reducing the time interval of input forcing $\Delta T$. Theoretically, as $\Delta T$ decreases, the results of the loosely-coupled model should converge to those of the fully-coupled model. The initial $\Delta T$ was set to 90 min and reduced to 45 and 15 min for the loosely-coupled model, while $\Delta T$ used for the fully-coupled model was 90 min. The consistency of the loose-coupled model with the fully-coupled model was evaluated using the MAE metric.

## 3.2 Model application

The TDD-XAJ model was applied in the Tunxi watershed (Fig. 5) to assess its performance. The Tunxi watershed is located at the headwater of the Xinanjiang river system in China, where the XAJ model is named accordingly. The Tunxi watershed is a typical humid watershed, with annual precipitation of 1,750 mm and an area of 2,670 km$^2$. Its elevation ranges from 121 to 1,614 m. The watershed is characterized by a hilly landscape comprising mountains, high and low hills, and intermountain basins. Due to the intense convective activity influenced by the unique terrain, precipitation can be heavy, resulting in rapid rises and falls floods.

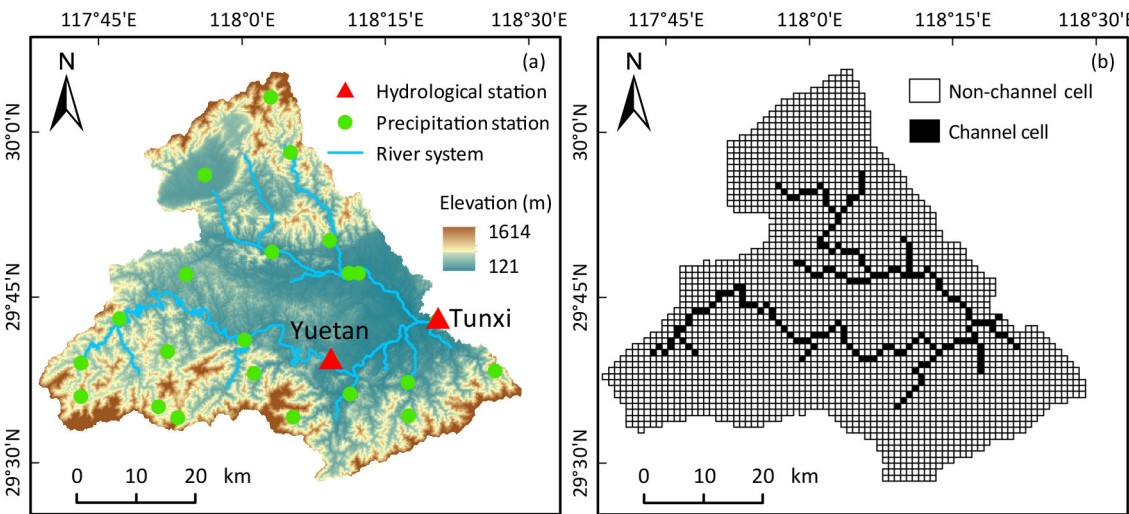

**Figure 5.** Location and gauging station distribution of the Tunxi watershed (a), and the spatial discretization of the watershed, including channel and non-channel cells (b).

Daily scale hydrological data were used in this study. The discharge and precipitation series were observed by the gauging network within the Tunxi watershed, which consists of the Tunxi hydrological station, the Yuetan hydrological station, and 23 precipitation stations (Fig. 5a). The duration ranges from 1 January 2007 to 31 December 2019, and the data source is the

Annual Hydrological Report of China (Volume VI, Book XVIII). The daily pan evaporation series was collected from the daily surface climatological data for China (V3.0) within the same period. The spatial distribution of precipitation within the Tunxi watershed was obtained using the ordinary Kriging method provided by PyKrige (Murphy et al., 2024), while the watershed averaged value of pan evaporation was used for simulation.

A structured grid of 1000 m×1000 m was used for simulation (Fig. 5b). The Digital Elevation Model (DEM) utilized was a 90 meter resolution SRTMDEMUTM data product, while the land use information was obtained from the 30 meter resolution GlobeLand30 (V2020) data product. To simplify the research and parameter calibration process, the parameters related to the runoff generation process were assumed to be spatially uniform and calibrated manually. The surface roughness coefficient $n_s$ was derived from the land use product (Shu et al., 2024). Based on DEM data and GIS analysis, 17

river channels were extracted for simulation. The channel roughness coefficient $n_c$ was estimated based on the actual characteristics of the river channel. The range for $n_c$ is between 0.025 and 0.040. The hydrometeorological data in the first year (1 January 2007 to 31 December 2007) were used for model spin-up, while the data in the following 7 years (1 January 2008 to 31 December 2014) were used for calibration, and the remaining data were used for validation. Model performance was evaluated with NSE, Kling-Gupta efficiency coefficient (KGE), flood volume relative error (FVRE), and the coefficient

of determination ($R^2$) (Jackson et al., 2019). The equation of FVRE could be given as:

$$\text{FVRE} = \left(\sum_{i=1}^{n} Q_{\text{sim,i}} - \sum_{i=1}^{n} Q_{\text{obs,i}}\right) / \sum_{i=1}^{n} Q_{\text{obs,i}} * 100\% \tag{32}$$

where $Q_{\text{sim,i}}$ and $Q_{\text{obs,i}}$ represent simulated and observed discharge at time step i (m$^3$ s$^{-1}$), and $n$ is the length of the sequence.

## 4 Results and Discussion

### 4.1 Slope concentration method comparison

The surface and subsurface slope concentration methods were compared in single- and double-slope case, which resulted in four sets of simulation results. The simulated hydrographs of hillslope and channel outflow are presented in Fig. 6. The state variable distribution on the left hillside at the 60 minute mark are shown in Fig. 7.

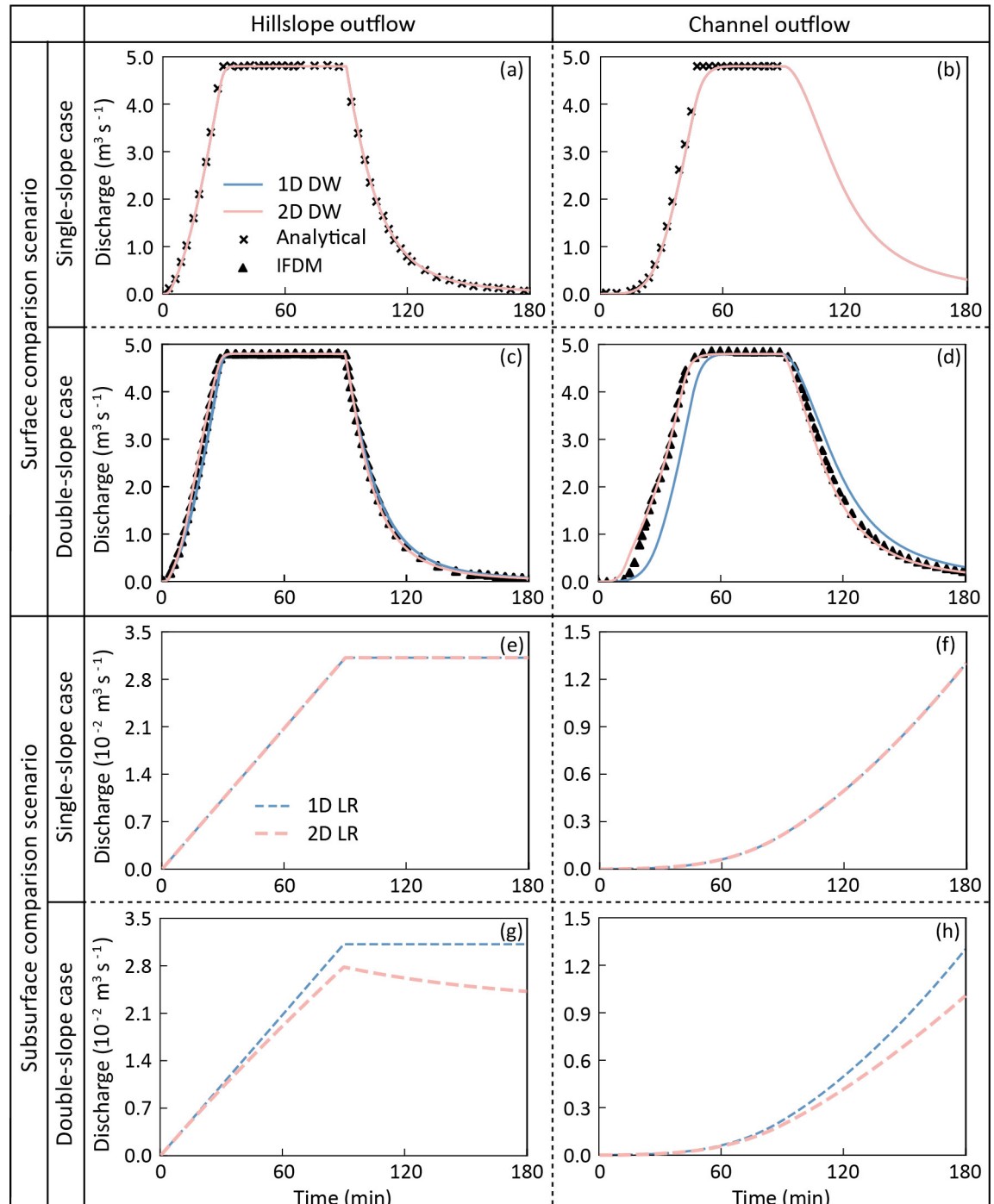

**Figure 6.** The hillslope and channel outflow hydrograph of the single- and double-slope synthetic V-catchment test case in surface and subsurface comparison scenario. In the surface comparison scenario (a-d), the one-dimensional (1D) and two-dimensional (2D) diffusion wave equations (DW) were evaluated, and the analytical solution for single-slope case and the IFDM solution for the double-slope case

were derived from (Di Giammarco et al., 1996). The 1D and 2D linear reservoir equations (LR) were compared in the subsurface comparison scenario (e-h).

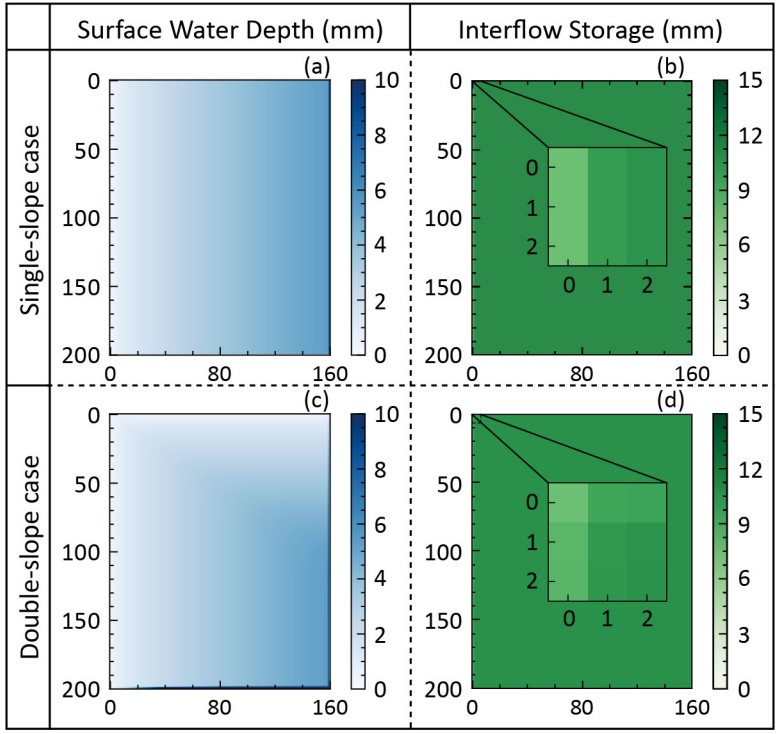

**Figure 7.** The spatial distribution of surface water depth ($h_s$) and interflow storage ($O_i$) on the left-side hillslope of both single-slope (a-b) and double-slope (c-d) synthetic V-catchment test cases at the 60 minute mark. The state variable distributions shown are simulated using two-dimensional (2D) slope concentration methods. The corresponding results of 1D methods are identical to those obtained from the single-slope case simulated with 2D methods, regardless of the test case used. For a clear comparison, the spatial distribution of $O_i$ in the upper left corner has been zoomed.

As shown in Fig. 6a and 6b, the simulation results of the 1D and 2D diffusion wave methods coincide with each other, and it can be attributed to the consistency of the actual overland flow direction between the one analyzed by the D8 method, which is perpendicular to the channel. The NSE values for the hillslope and channel outflow exceed 0.999 when compared with the analytical solution, demonstrating the applicability of 1D and 2D diffusion wave methods in the single-slope case. Significant differences exist in the hydrographs corresponding to the 1D and 2D diffusion wave method in the double-slope

case, with a notably pronounced lag in the channel outflow for the 1D diffusion wave method, presented in Fig. 6d. It is worth noting that for the 1D diffusion wave method, the flow directions analyzed using the D8 method in both single- and double-slope cases are identical, consistent with the simulation results. Using the IFDM solution as a comparative reference, the NSE of the hillslope outflow is 0.991 and 0.998 for the 1D and 2D diffusion wave method, while the NSE for channel outflow is 0.865 and 0.992, respectively. The poor performance of the 1D diffusion wave method can be explained by its

neglect of the $y$-directional flow component, which is rooted in its inability to capture the surface microtopography. For

further illustration, the surface water depth of the left-side hillslope in the double-slope case for the 1D diffusion wave method is shown in Fig. 7a, and the corresponding result of the 2D diffusion wave method is displayed in Fig. 7c.

The subsurface slope concentration comparison scenario shows the same pattern, i.e., the simulation results of the 1D and 2D linear reservoir method are the same in the single-slope case, but there is a certain discrepancy in the double-slope case. In the single-slope case, the intensity of hillslope interflow outflow increases with the continuation of the precipitation in the first 90 min and remains constant in the following 90 min, while the channel outflow increased throughout the simulation, which is shown in Fig. 6e and 6f. The discrepancy in the double-slope case is also attributed to the mismatch of the flow direction used and the actual flow direction in the 1D linear reservoir approach. The outflow intensity from the hillslope and channel using the 2D linear reservoir method is slower than that of the 1D method, and the hillslope interflow outflow keeps decaying after the precipitation ceases, as depicted in Fig. 6g. The flow length of the subsurface flow path in the double-slope case is longer than that of the single-slope case, which means the interflow slope concentration process is subject to a stronger storage effect, resulting in a slower outflow rate. Meanwhile, the simulation results of the 1D linear reservoir are identical in the single- and double-slope case. Hence, the simulation results of the 2D linear reservoir method are more reasonable in the double-slope case. The comparison of interflow linear reservoir storage in the upper left corner of the left-side hillslope at 60 minute mark also supports the ignorance of $y$-directional interflow component for the 1D linear reservoir method in the double-slope case (Fig. 7b and 7d).

The result of this numerical experiment indicate that 1D slope concentration methods perform relatively poorly in double-slope case, regardless of the surface or subsurface comparison scenario evaluated. This limitation partly arises from the inability of the D8 method to capture the actual flow direction accurately, highlighting the need for a more suitable single flow direction algorithm. Essentially, this reflects the shortcomings of the 1D method in characterizing complex terrains. In contrast, the 2D slope concentration methods exhibit good simulation accuracy in both the single- and double-slope case. These 2D methods effectively represent flow direction by synthesizing flow velocities in both the $x$- and $y$-direction, without reliance on specific flow direction algorithms. In summary, the findings from this numerical experiment underscore the advantages of using 2D slope concentration methods within the TDD-XAJ model.

## 4.2 Numerical implementation framework comparison

The loosely- and fully-coupled models were conducted on two V-catchment test cases with different $\Delta T$. Model parameters and the amount of precipitation were configured to ensure a total of 16.2 mm $R_s^*$ were generated. The MAE metric was employed to evaluate the simulated model fluxes of the loosely-coupled model against the analytical values or the solutions from the fully-coupled model. A total of 500 parameter sets were utilized, resulting in 500 MAE values for each model flux, as detailed in Table 2.

**Table 2.** MAE statistics of model fluxes in numerical implementation comparison experiment.

| Model | $\Delta T$ (min) | Statistics | MAE (mm) $R_s^*$ | MAE (m$^3$/s) Hillslope outflow Single-slope | Hillslope outflow Double-slope | Channel outflow Single-slope | Channel outflow Double-slope |
|---|---|---|---|---|---|---|---|
| Loosely-coupled | 90 | Max | 5.93 | 2.19 | 2.20 | 2.01 | 2.06 |
| | | Average | 4.57 | 1.85 | 1.87 | 1.68 | 1.72 |
| | 45 | Max | 2.01 | 1.60 | 1.62 | 1.31 | 1.41 |
| | | Average | 1.21 | 0.81 | 0.82 | 0.70 | 0.73 |
| | 15 | Max | 0.33 | 0.56 | 0.61 | 0.36 | 0.39 |
| | | Average | 0.14 | 0.16 | 0.17 | 0.11 | 0.12 |
| Fully-coupled | 90 | Max | $4.19\times10^{-3}$ | —[a] | — | — | — |
| | | Average | $2.84\times10^{-4}$ | — | — | — | — |

a. The results of the fully-coupled model are used as references to calculate the MAE values for hillslope and channel outflow, so the corresponding value is empty.

As shown in Table 2, the maximum and average MAE values of $R_s^*$ simulated by the fully-coupled model are $4.19\times10^{-3}$ and $2.84\times10^{-4}$ mm, respectively, indicating a good alignment with the analytical solution. It is worth noting that the analytical solution of $R_s^*$ using the same parameter set should be identical for both single- and double-slope case. When $\Delta T$ is set to 90 minutes, the maximum and average MAE values of $R_s^*$ simulated by the loosely-coupled model are 5.93 and 4.57 mm, which account for 36.6% and 28.2% of their analytical value. As $\Delta T$ is reduced from 90 min to 15 min for the loosely-coupled model, the maximum and average MAE values for $R_s^*$ become 2.0% and 0.9% of their analytical value, demonstrating a significant decrease trend but remaining relatively large compared to the fully-coupled model.

As $\Delta T$ decreases, the MAE values for both hillslope and channel outflow simulated by the loosely-coupled model show a consistent downward tendency for both single- and double-slope case. These MAE values, benchmarked against the fully-coupled model's simulation results, suggest that the outputs of the loosely-coupled model are gradually converging towards those of the fully-coupled model. As an example, we illustrated the outflow hydrograph of various model configurations in the double-slope case (Fig. 8), using the first set of the SLH-generated parameters.

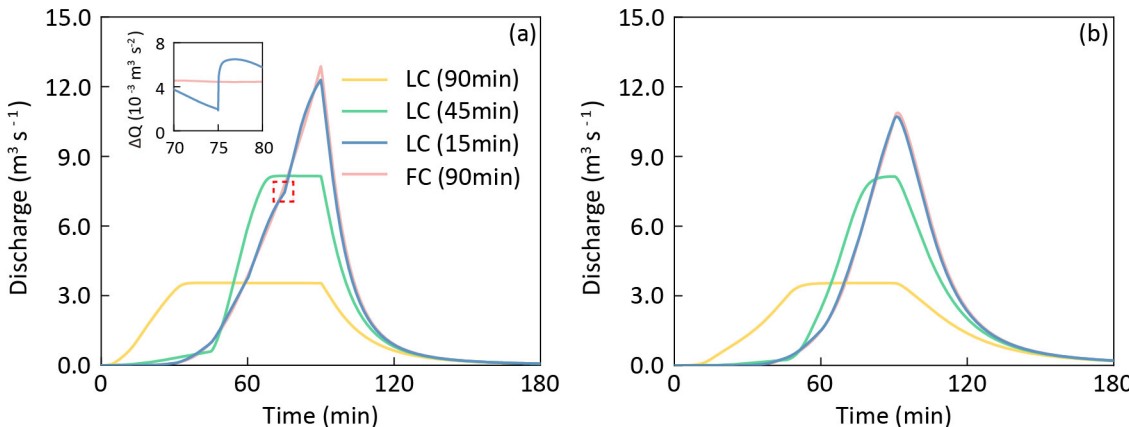

**Figure 8.** The simulated hillslope outflow (a) and channel outflow (b) of the loosely-coupled (LC) model and fully-coupled (FC) model in the double-slope case. The time in parentheses denotes the time interval of input forces ($\Delta T$) used for the model. The FC model uses 90 min only, while the LC model uses 90 min, 45 min, and 15 min for convergence test. The parameters used are the first set of the parameters sampled with the SLH method from their ranges listed in Table 2. The inflection point at 75 minute mark in the hillslope outflow hydrograph of the LC model (using $\Delta T$=15min) is highlighted with a red dashed box, along with timeseries of the first-order numerical derivative of hillslope outflow ($\Delta Q$).

As shown in Fig. 8, the hillslope and channel outflow hydrograph of the loosely-coupled model increasingly converge with those of the fully-coupled model as $\Delta T$ decreases. Consequently, the impact of the numerical implementation framework on simulation results is analyzed based on the solution of the loosely-coupled and fully-coupled model ($\Delta T$= 90 min for both models). The average MAE values for channel outflow are 2.01 and 2.06 m³ s⁻¹ in single- and double-slope case, lower than those for hillslope outflow, respectively. This could be explained by the cancel-out phenomenon of numerical error. The numerical error alternates between positive and negative values, and delayed effects are introduced in the runoff concentration process. The channel outflow at a specific moment is influenced by the cumulative effects of the numerical error from the current and previous time steps, resulting in a reduction in magnitude of numerical error. Although the average MAE value for channel outflow shows a certain decrease, its overall magnitude remains comparable to that of hillslope outflow, indicating that the numerical errors still require control. Additionally, the mean MAE values for hillslope outflow and channel outflow in the double-slope test case are 1.87 and 1.72 m³ s⁻¹, higher than those for the single-slope test case. This could be attributed to the concentration of overland flow on the downstream side of the channel in the double-slope case, where the overall numerical error is more pronounced. This suggests that more complex terrain may be more prone to amplifying numerical errors, a phenomenon that, to the best of our knowledge, has not been reported in previous studies.

It can also be seen from Fig. 8, the simulation results of the loosely-coupled model show a steady state phenomenon when $\Delta T$ equals 90 and 45 min, and there is a clear inflection point in simulated hillslope outflow hydrograph when $\Delta T$ equals 15 min. The false steady outflow phenomenon and inflection point observed indicate a potential limitation in capturing transient

behaviors of the loosely-coupled numerical implementation framework, and highlight the advantages of the fully-coupled model's continuous updating mechanism.

This experiment highlights that significant numerical error arises from the loosely-coupled numerical implementation framework previously employed. Additionally, the numerical error may be exacerbated owing to a potential terrain amplification effect. In contrast, the fully-coupled numerical framework can effectively control numerical errors and capture transient behaviors. In summary, the results of this numerical experiment emphasize the benefits of utilizing purely differential mathematical equations and a fully-coupled numerical framework within the TDD-XAJ model.

## 4.3 Model application result in Tunxi watershed

The TDD-XAJ model was applied in the Tunxi watershed, a typical watershed in the humid area, to simulate daily hydrograph. The calibrated parameters of the model are listed in Table 3. The simulated result was evaluated with four performance metrics annually against the observations, which are listed in Table 4.

**Table 3.** Model parameters of the TDD-XAJ in the Tunxi watershed.

| $K_c$ | $W_{um}$ | $W_{lm}$ | $W_{dm}$ | $b$ | $c$ | $A_{imp}$ |
|-------|----------|----------|----------|------|------|-----------|
| 0.73 | 18.23 | 69.32 | 30.32 | 0.14 | 0.14 | 0.20 |
| $S_m$ | $ex$ | $K_i$ | $K_g$ | $C_i$ | $C_g$ | — |
| 14.19 | 1.37 | 0.18 | 0.52 | 0.505 | 0.995 | — |

**Table 4.** Annually evaluated simulation performance metrics of the TDD-XAJ model in the Tunxi watershed.

| Period | Year | Tunxi station | | | | Yuetan station | | | |
|--------|------|-----|-----|---------|-------|-----|-----|---------|-------|
| | | NSE | KGE | FVRE(%) | $R^2$ | NSE | KGE | FVRE(%) | $R^2$ |
| Calibration | 2008 | 0.94 | 0.91 | -8.26 | 0.94 | 0.90 | 0.89 | -1.33 | 0.90 |
| | 2009 | 0.88 | 0.90 | -6.95 | 0.88 | 0.82 | 0.80 | -16.43 | 0.83 |
| | 2010 | 0.85 | 0.78 | -16.88 | 0.90 | 0.82 | 0.78 | -18.99 | 0.87 |
| | 2011 | 0.89 | 0.78 | 7.53 | 0.89 | 0.78 | 0.76 | -3.46 | 0.80 |
| | 2012 | 0.82 | 0.84 | -7.64 | 0.83 | 0.74 | 0.80 | -5.97 | 0.74 |
| | 2013 | 0.87 | 0.80 | -10.75 | 0.92 | 0.88 | 0.79 | -3.99 | 0.92 |
| | 2014 | 0.88 | 0.79 | 0.20 | 0.91 | 0.84 | 0.75 | 0.72 | 0.87 |
| Validation | 2015 | 0.85 | 0.77 | -9.92 | 0.92 | 0.85 | 0.80 | -8.48 | 0.88 |
| | 2016 | 0.88 | 0.78 | -6.92 | 0.92 | 0.86 | 0.79 | -4.37 | 0.89 |
| | 2017 | 0.88 | 0.76 | 1.62 | 0.92 | 0.84 | 0.72 | 4.66 | 0.86 |
| | 2018 | 0.87 | 0.77 | 1.58 | 0.89 | 0.87 | 0.78 | -2.75 | 0.90 |
| | 2019 | 0.85 | 0.74 | -2.24 | 0.89 | 0.79 | 0.74 | -3.70 | 0.81 |

For Tunxi station at the outlet of Tunxi watershed, Table 4 indicates that the values of the FVRE metric are all within ±20 %, with the absolute values of the FVRE (|FVRE|) averaging 8.3% and 4.5% for the calibration and validation period, respectively. In terms of hydrograph evaluation, the average values of NSE and KGE are 0.88 and 0.83 for the calibration period and 0.87 and 0.76 for the validation period, which is slightly better for the calibration period than for the validation period. The minimum value of $R^2$ is 0.83 for all years, and the average value for all years is 0.90. In a direct comparison,

Tong (2022) conducted a similar daily simulation in the same watershed using the GXAJ model, reporting average NSE and
|FVRE| values of 0.85 and 11.0% between 2008 and 2017, respectively. In contrast, the TDD-XAJ model achieved average
values of 0.87 for NSE and 7.7% for |FVRE| in the same period. For Yuetan station within Tunxi watershed, Table 4 shows
that FVRE metric values remain within ±20 %. The average |FVRE| is 7.3% and 4.8% for the calibration and validation
periods, respectively. Meanwhile, the average value of NSE is 0.82 for the calibration period and 0.84 for the validation
period, and the average KGE is 0.80 and 0.77 for calibration period and validation period, respectively. The average $R^2$
Across all years is 0.86. Fig. 9 provides an example of the simulated hydrograph at Tunxi and Yuetan station of the TDD-
XAJ model in 2008.

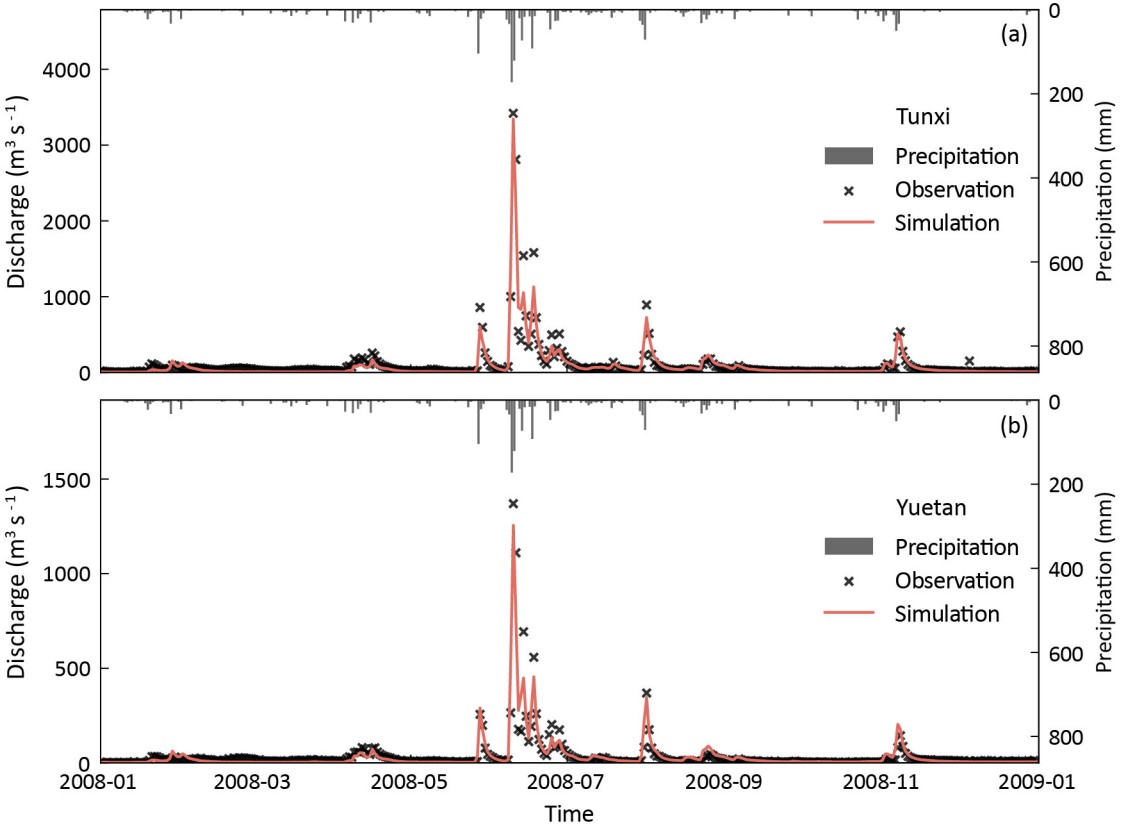

**Figure 9.** The simulated hydrograph at Tunxi (a) and Yuetan (b) station of the Tunxi watershed in 2008 using the TDD-XAJ model.

We further analyzed the spatial simulation capability of the TDD-XAJ model using precipitation events from June 6-8, 2008,
which occurred just before the flood peak and exhibited distinct spatial patterns (Fig. 10a-c). On June 6, almost no rainfall
was recorded; on June 7, precipitation was concentrated in the upper-left corner; and on June 8, it was focused in the lower-
right corner. The simulation results for tension and free water storage reflect these patterns. Specifically, on June 7 (Fig. 10d
and 10g), the overall tension water storage was unsaturated with minimal free water storage. Subsequently, saturation of
tension water occurred in the upper-left part of the watershed, and free water storage increased significantly in areas with

heavier rainfall (Fig. 10e and 10h). On June 8, the tension water in the lower-right corner became fully saturated due to the coverage of precipitation (Fig. 10f). For free water (Fig. 10i), the storage in the same area also approached saturation, whereas the saturated areas in the upper part quickly dissipated due to the faster recession of free water.

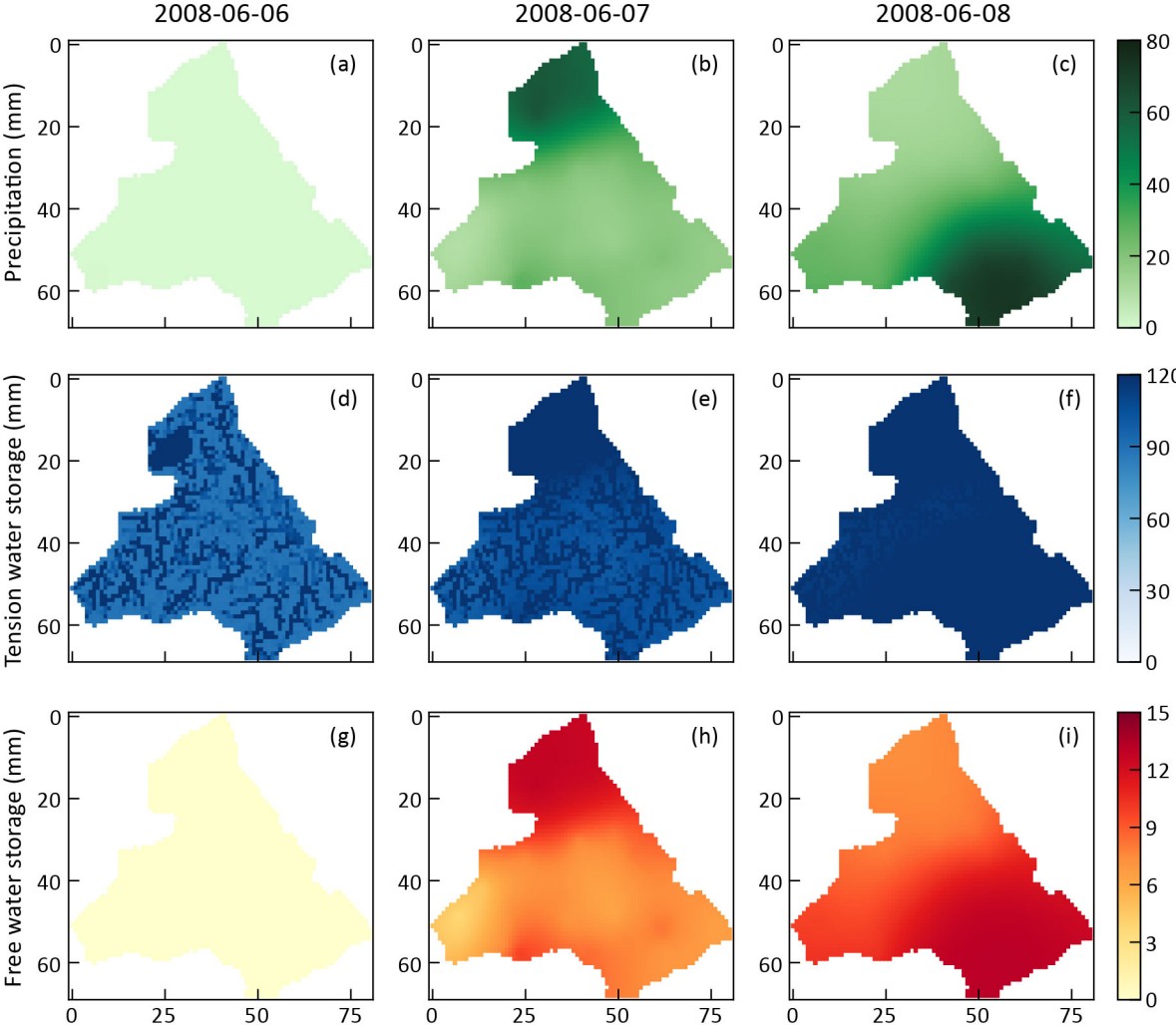

**Figure 10.** The spatial distribution of precipitation for three days in 2008 (a-c), along with the spatial distribution of tension (d-f) and free water storage (g-i) simulated by TDD-XAJ model for the three days.

The application results indicate a good and consistent agreement between the observation and the simulations of the TDD-XAJ model in the Tunxi watershed, demonstrating a slight improvement in performance metrics compared to the GXAJ model, particularly for flood volume. Additionally, the spatial analysis of simulation results demonstrates that the TDD-XAJ model captures spatial variability in water storage effectively in response to varying precipitation patterns.

## 4.4 Computational efficiency of the model

The implementation of the TDD-XAJ model is detailed in Appendix A. The Python language was chosen for its robust ecosystem in hydrology and earth system sciences (Stacke and Hagemann, 2021; Murphy et al., 2024), which facilitates rapid development and experimentation. While Python, as an interpreted language, is generally less computationally efficient than compiled languages, it benefits from libraries like NumPy, which rely on compiled languages for underlying computations. To enhance computational performance, we leveraged NumPy's vectorized operations and optimized functions, which are generally faster than native Python code.

To compare the computational efficiency of the 1D and 2D slope runoff concentration methods, we repeated the models used in corresponding comparison experiment for 10 times, and recorded the runtime for each. The average runtime values, obtained using an AMD 5950X CPU (3.4 GHz) in single-core mode, are summarized in Table 5.

**Table 5**. Average runtime recorded in the slope runoff concentration methods comparison numerical experiment.

| Surface comparison scenario (s) | | | Subsurface comparison scenario (s) | | |
|---|---|---|---|---|---|
| Method | Single-slope case | Double-slope case | Method | Single-slope case | Double-slope case |
| 1D diffusion wave | 28.41 | 28.32 | 1D linear reservoir | 2.10 | 2.06 |
| 2D diffusion wave | 29.02 | 29.80 | 1D linear reservoir | 1.13 | 1.12 |

As shown in Table 5, replacing 1D slope runoff concentration methods with 2D methods incurs minimal computational overhead while potentially enhancing efficiency. For diffusion wave method used for surface runoff concentration, the runtime for the 2D form increases 2.1% and 5.2% compared to the 1D form in the single-slope and double-slope case, respectively. Conversely, for linear reservoir method applied to subsurface runoff concentration, switching from 1D to 2D reduces runtime by at least 45.6% across both test cases. The numerical computation procedure for the slope runoff concentration methods mainly involves two steps: flux calculation and state variable update. In flux calculation step, the 1D and 2D diffusion wave method require calculating once per cell (based on GIS-derived flow direction) or twice (in both $x$ and $y$ directions), respectively. For linear reservoir method, both the 1D and 2D forms require calculation only once per cell, and the outflow flux evaluated is further decomposed into $x$- and $y$- directional components based on slope aspect for the 2D form. When updating state variables, both the diffusion wave and linear reservoir methods in their 1D form require logical judgments based on GIS-derived flow direction to identify the inflow cells. In their 2D form, this can be directly determined using cell indexing. The element-wise logical judgment is relatively time-consuming and is comparable to a one-time flux calculation per cell. Therefore, for the diffusion wave method, transitioning from 1D to 2D form does not result in a significant increase in runtime, but for the linear reservoir method, the computational overhead is effectively reduced.

## 4.5 Limitations and future research

The main limitation of the TDD-XAJ model is that it addresses only the numerical errors on the time scale from the ODE's numerical solution, while neglecting errors arising from the spatial discretization of the PDE via the FVM method. Future research should consider using methods like manufactured solutions (Bisht and Riley, 2019) as an alternative for evaluating

numerical errors arising from both spatial and temporal discretization when exact solutions are difficult to obtain. Additionally, exploring more advanced spatial discretization techniques, such as second-order FVM or discontinuous Galerkin method (Shaw et al., 2021), along with more sophisticated temporal integration methods, could help control numerical errors more effectively.

In terms of computational efficiency, although we make extensive use of the NumPy library, which is generally faster than native Python code, we recognize that NumPy may still be slower in certain operations (e.g., element-wise calculations) compared to compiled languages. This is a well-known limitation within the Python community, and solutions like Just-In-Time (JIT) compilation have been proposed (Lam et al., 2015), which convert frequently executed script code into machine code with further automatic optimizations. Although this manuscript primarily focuses on presenting the theoretical aspects of the TDD-XAJ model, we plan to optimize the code, including the implementation of parallelization, in future work.

It is essential to validate the model with more real-world watershed data and evaluate its uncertainty, particularly for watersheds with varying scales, diverse underlying physical conditions, and hydrological data at different temporal resolutions. The differential-form mathematical equations established for the TDD-XAJ model provide a solid foundation for future research, including combining deep learning for better model parameterization and process understanding (Höge et al., 2022; Li et al., 2024).

## 5 Summary and Conclusions

The Xinanjiang model has been widely applied in China, and its modeling philosophy is referenced in the development of several other hydrological models commonly used worldwide. However, the development of a distributed version of the XAJ model has been somewhat stagnant. In this study, we propose a new grid-based distributed Xinanjiang model (TDD-XAJ). The features of the TDD-XAJ model include the usage of 2D slope concentration methods, the establishment of purely differential-form mathematical equations, and a fully-coupled numerical framework for model solution. The performance of the TDD-XAJ model was tested and compared with the existing distributed Xinanjiang model [GXAJ, proposed by Yao et al. (2009)], leading to the following conclusions:

1. The TDD-XAJ model replaces 1D diffusion wave equations for surface slope concentration with 2D diffusion wave equations; the new 2D linear reservoir equations were derived for subsurface slope concentration. The combination of the 2D methods forms the slope concentration module of the model. Comparisons conducted on two V-catchment test cases reveal that the 2D methods enhance the model's ability to capture the microtopography of complex terrain. Additionally, using 2D methods eliminates the need for careful selection of a single flow direction algorithm, which is required in 1D methods.

2. The ODE form equations describing the runoff generation process are integrated with 2D slope and 1D channel concentration equations. This integration establishes a purely differential-form mathematical framework of the TDD-XAJ model. The framework facilitates a fully-coupled numerical implementation, resulting in a system of ODEs that can be

solved simultaneously. A comparison experiment of the numerical implementation indicates that numerical error introduced by the previous loosely-coupled manner in the GXAJ model is evident and likely to be amplified by complex terrains, while the use of fully-coupled manner in the TDD-XAJ model can effectively mitigate this issue.

3. The TDD-XAJ model was applied in the Tunxi watershed, a typical humid study area in China, and its results were compared with those from the GXAJ model. The findings indicated that the TDD-XAJ model outperforms the GXAJ model in key hydrological skill metrics, particularly in its improved ability to simulate flood volume. These results preliminarily demonstrate the effectiveness and applicability of the TDD-XAJ model for real-world watershed simulations.

## Appendix A: Implementation details of the TDD-XAJ model

The TDD-XAJ model is implemented in Python 3.9, taking advantage of its extensive ecosystem as well as cross-platform
capabilities. To maintain readability and facilitate understanding, only essential libraries are used, ensuring the model is both accessible and easily extendable in the future. A list of these libraries is provided in Table A1. NumPy plays a crucial role, providing vectorized operations and optimized computational functions that enhance performance compared to native Python code. Other libraries serve supporting roles, such as model configuration and handling file input/output.

**Table A1.** Libraries utilized in the implementation of the TDD-XAJ Model

| Category | Name | Version | Description |
|---|---|---|---|
| Core | NumPy | 1.26.4 | A basic scientific computing library for efficient numerical operations |
| Auxiliary | PyYAML | 6.0.1 | Library for reading YAML files used for model configuration and settings |
| | Pandas | 2.2.1 | A high-level library for data manipulation and analysis |
| | OpenPyxl | 3.1.5 | Backend support for reading and writing Excel files in Pandas |
| | treelib | 1.7.0 | Efficient implementation of tree data structure |

## Appendix B: Inputs and outputs of the TDD-XAJ model

The inputs and outputs of the TDD-XAJ model are listed in Table B1. Its inputs consist of four categories: (1) model configuration dictionary, (2) spatiotemporal multidimensional arrays of precipitation and evaporation as forcing inputs, (3) raster datasets providing spatial attributes (including elevation, river network, aspect, etc.), and (4) lookup tables storing hydraulic parameters for surface runoff and channel concentration. The outputs comprise spatiotemporal distributions of
675 state variables and model fluxes at pre-defined temporal resolutions (instantaneous or time-averaged values). Compared to the original XAJ model, which is a lumped hydrological model, its inputs are watershed-scale parameters and timeseries for precipitation and evaporation, with output mainly including simulated discharge hydrograph. The TDD-XAJ model shares largely similar input and output information with existing distributed XAJ model, the aspect data introduced by the 2D linear reservoir method can be derived from elevation raster.

**Table B1.** Detailed information on input and output files of the TDD-XAJ model.

| Class | Category | Name | Format | Description |
|---|---|---|---|---|
| Input | Dictionary | Model configuration | YAML | (1) Paths for the other files in this table, temp and output folders<br>(2) Numerical calculation settings (e.g., CFL coefficient)<br>(3) Model parameters<br>(4) Location of hydrological stations |
| | Array | Forcings | NPY | Spatiotemporal distribution of precipitation and evaporation |
| | Raster | Elevation | Esri ASCII | Use projected coordinate system, with unit of m |
| | Raster | Aspect | Esri ASCII | Derived from elevation raster via GIS analysis |
| | Raster | Flow accumulation | Esri ASCII | Derived from elevation raster via GIS analysis |
| | Raster | Flow direction | Esri ASCII | Derived from elevation raster via GIS analysis |
| | Raster | River network | Esri ASCII | Mark the grid where channel is located and corresponding code |
| | Raster | Land use | Esri ASCII | Aligned with elevation raster, and store land use code |
| | Lookup table | Surface hydraulics | CSV/XLSX | Land use code and corresponding surface roughness coefficient |
| | Lookup table | Channel hydraulics | CSV/XLSX | Channel code and corresponding attributes (e.g., longitudinal slope and channel roughness coefficient) |
| Output | Array | State variables | NPY | Spatiotemporal distributions of state variables at pre-defined temporal resolutions (instantaneous or time-averaged values) |
| | Array | Model fluxes | NPY | Spatiotemporal distributions of model fluxes at pre-defined temporal resolutions (instantaneous or time-averaged values) |

## Code availability

The documentation of PyKrige is provided at https://pykrige.readthedocs.io (last access: 20 November 2024). The code used in this study are available through Zenodo via https://doi.org/10.5281/zenodo.14227068 (Zhao, 2024a).

## Data availability

The daily surface climatological data for China (V3.0) is downloaded from https://data.cma.cn (last access: 13 February 2022). The cross-section data of the channels in the Tunxi watershed are obtained from the local hydrological water information website (http://yc.wswj.net/ahsxx/LOL/, last access: 19 November 2024). The SRTMDEMUTM data can be downloaded from https://www.gscloud.cn (last access: 19 November 2024). The GlobeLand30 (V2020) data is available at https://www.webmap.cn (last access: 19 November 2024). The inputs and outputs of numerical experiments and model application are stored in the Zenodo data repository at https://doi.org/10.5281/zenodo.14226969 (Zhao, 2024b).

## Author contribution

JZ had the original idea, and developed the conceptualization of this study with ZL. TW, JZ, and JW collected the data. The methodology was designed by ZL, JZ, and VS. JZ and TW developed the software and conducted all the model simulations and their formal analysis. The results were discussed and interpreted between ZL, VS, and YH. The visualizations and the

original draft of the manuscript were prepared by JZ, and the revisions and editing were done by VS, YH, and BL. The project was administrated by ZL. All the authors have read and agreed to the current version of the paper.

**Competing interests**

The authors declare that they have no known competing financial interests or personal relationships that could have appeared to influence the work reported in this paper.

**Acknowledgements**

The authors are grateful to the editor and reviewers for their valuable comments and constructive feedback, which have greatly enhanced the quality of this manuscript.

**Financial support**

This study has been supported by the Fundamental Research Funds for the Central Universities (B240201164), Jiangsu Provincial Outstanding Postdoctoral Program (2024ZB603), the Postdoctoral Fellowship Program of CPSF (GZC20240380), the China Postdoctoral Science Foundation (2024M760740), and the National Natural Science Foundation of China (523043811).

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
