# Peer review of "Two-dimensional Differential-form of Distributed Xinanjiang Model"

_Hydrology and Earth System Sciences, 2024_

## Community Comment (CC2)

**Response to Reviewer RC1**

**Title**: Two-dimensional Differential-form of Distributed Xinanjiang Model
**Authors**: Jianfei Zhao, Zhongmin Liang\*, Vijay P. Singh, Taiyi Wen, Yiming Hu, Binquan Li, Jun Wang
**Manuscript ID**: hess-2024-377

We would like to sincerely thank you for thoroughly reviewing our manuscript. All your comments have been carefully addressed, and a point-by-point response is provided below.

For better readability, the point-by-point response is formatted as follows:

- Reviewer's comments are shown in black
- Authors' responses are shown in blue
- Revisions to be incorporated in the revised manuscript are highlighted in red
- Figures used only in this response will have the prefix "R", such as Figure R1
- Figures without prefix "R" are numbered as they appear in the manuscript

**Overall comments:**

This work is a typical representation of the forward development in the current hydrological modeling field. By leveraging existing advanced numerical techniques, it reconstructs the traditional empirical, conceptual, and lumped hydrological model, transforming it into modern distributed hydrological model that can represent spatial heterogeneity and are based on complete systems of differential equations. Overall, I think this work is of great practical significance. In addition, the author's writing is also quite mature and concise. Nevertheless, I still believe that this work should undergo major revision before it is suitable for publication, as the presentation of the results section remains insufficient for a mature study.

Thank you for your positive remarks on our work, particularly for acknowledging and highlighting our efforts. Your comments are invaluable, and we have carefully considered each point to enhance the clarity and maturity of the subsequent revised manuscript. We hope that these changes could effectively address your concerns.

**Major concerns:**

1. I have reviewed the model code provided by the author and noticed that it is relatively concise pure Python code and does not include many additional dependencies. I believe this is a strong advantage, as it suggests broader usage scenarios and greater potential for expansion. However, as an interpreted language, Python has a typical limitation in comparison to compiled languages like C, namely its lower efficiency. As a potential application for large domain modeling, I am concerned about the computational

efficiency of the TDD-XAJ model. The author should at least provide a reference for efficiency, and clarify whether transitioning from a 1D modeling framework to a 2D framework would result in a significant decrease in computational efficiency. I believe the author should add a section to address these issues.

In addition, please include a description of the programming environment of the model, as well as the dependencies used, to help users clearly understand the usage scenarios.

Thank you for taking the time and effort to review the code we provided. The model was implemented solely in Python, and to ensure readability, understandability, and ease of application and potential future extensions, we used only essential libraries, keeping the code concise while maintaining completeness. We are grateful for your recognition of this manner and your positive feedback.

We understand your concern regarding the computational efficiency of the TDD-XAJ model. Just as you pointed out, Python is an interpreted language, does lag behind compiled languages such as C or Fortran in terms of execution speed. To address this, we will add a section in Result and Discussion part of the revised manuscript, which is shown below:

**"4.4 Computational efficiency of the model**

The implementation of the TDD-XAJ model is detailed in Appendix A. The Python language was chosen for its robust ecosystem in hydrology and earth system sciences (Stacke and Hagemann, 2021; Murphy et al., 2024), which facilitates rapid development and experimentation. While Python, as an interpreted language, is generally less computationally efficient than compiled languages, it benefits from libraries like NumPy, which rely on compiled languages for underlying computations. To enhance computational performance, we leveraged NumPy's vectorized operations and optimized functions, which are generally faster than native Python code. However, we recognize that NumPy may still be slower in certain cases, such as element-wise calculations, compared to compiled languages. This is a well-known limitation within the Python community, and solutions like Just-In-Time (JIT) compilation have been proposed (Lam et al., 2015), which convert frequently executed script code into machine code with further automatic optimizations. Although this manuscript primarily focuses on presenting the theoretical aspects of the TDD-XAJ model, we plan to optimize the code, including exploring parallelization, in future work."

Transitioning from a 1D modeling framework to a 2D framework does not result in a significant decrease in computational efficiency. To clarify this, we repeated the models used in the slope concentration methods comparison experiment and recorded the runtime. The details and analysis will be added to the new Section 4.4 of the revised manuscript, which is provided below:

"To compare the computational efficiency of the 1D and 2D slope runoff concentration methods, we repeated the models used in corresponding comparison experiment for 10 times, and recorded the runtime for each. The average runtime values, obtained using an AMD 5950X CPU (3.4 GHz) in single-core mode, are summarized in Table 5.

**Table 5**. Average runtime recorded in the slope runoff concentration methods comparison numerical experiment.

| Surface comparison scenario (s) | | | Subsurface comparison scenario (s) | | |
|---|---|---|---|---|---|
| Method | Single-slope case | Double-slope case | Method | Single-slope case | Double-slope case |
| 1D diffusion wave | 28.41 | 28.32 | 1D linear reservoir | 2.10 | 2.06 |
| 2D diffusion wave | 29.02 | 29.80 | 2D linear reservoir | 1.13 | 1.12 |

As shown in Table 5, replacing 1D slope runoff concentration methods with 2D methods incurs minimal computational overhead while potentially enhancing efficiency. For diffusion wave method used for surface runoff concentration, the runtime for the 2D form increases 2.1% and 5.2% compared to the 1D form in the single-slope and double-slope case, respectively. Conversely, for linear reservoir method applied to subsurface runoff concentration, switching from 1D to 2D form reduces runtime by at least 45.6% across both test cases. The numerical computation procedure for the slope runoff concentration methods mainly involves two steps: flux calculation and state variable update. In flux calculation step, the 1D and 2D diffusion wave method require calculating once per cell (based on GIS-derived flow direction) or twice (in both $x$ and $y$ directions), respectively. For linear reservoir method, both the 1D and 2D forms require calculation only once per cell, and the outflow flux evaluated is further decomposed into $x$- and $y$- directional components based on slope aspect for the 2D form. When updating state variables, both the diffusion wave and linear reservoir methods in their 1D form require logical judgments based on GIS-derived flow direction to identify the inflow cells. In their 2D form, this can be directly determined using cell indexing. The element-wise logical judgment is relatively time-consuming and is comparable to a one-time flux calculation per cell. Therefore, for the diffusion wave method, transitioning from 1D to 2D form does not result in a significant increase in runtime, but for the linear reservoir method, the computational overhead is effectively reduced."

As suggested, an appendix will be included in the revised manuscript detailing the programming environment and library dependencies used in the model, which is shown below:

**"Appendix A: Implementation details of the TDD-XAJ model**
The TDD-XAJ model is implemented in Python 3.9, taking advantage of its extensive ecosystem, as well as cross-platform capabilities. To maintain readability and facilitate understanding, only essential libraries

are used, ensuring the model is both accessible and easily extendable in the future. A list of these libraries is provided in Table A1. NumPy plays a crucial role, providing vectorized operations and optimized computational functions that enhance performance compared to native Python. Other libraries serve supporting roles, such as model configuration and handling file input/output.

**Table A1.** Libraries utilized in the implementation of the TDD-XAJ Model.

| Category | Name | Version | Description |
|---|---|---|---|
| Core | NumPy | 1.26.4 | A basic scientific computing library for efficient numerical operations |
| Auxiliary | PyYAML | 6.0.1 | Library for reading YAML files used for model configuration and settings |
| | Pandas | 2.2.1 | A high-level library for data manipulation and analysis |
| | OpenPyxl | 3.1.5 | Backend support for reading and writing Excel files in Pandas |
| | treelib | 1.7.0 | Efficient implementation of tree data structure |

,,

2. I have some confusion regarding the channel unit modeling in Figure 4. In the previous sections (Section 2.1 and Figure 1), the channel units are explicitly defined as segments, with their lengths clearly specified in three ways. In these cases, the width of the channel units is significantly smaller than the grid size so that the rainfall can be neglected. However, in this test case (Figure. 4), the grid size (5m x 5m) is significantly larger than the channel width (20m). Does this mean that the channel units need to encompass multiple grids? I would ask the author to provide a corresponding diagram to clarify how spatial discretization and channel unit modeling are handled in the test cases.

A similar issue also applies to real basin modeling. Could you provide a diagram to illustrate how spatial discretization is carried out in the Tunxi watershed, including the definition of grid units and channel units?

Thanks for your comments. The TDD-XAJ model uses DEM grid cells as its fundamental computational units, which means the simulation domain is spatially discretized with square structure grid (see Figure R1 below). These cells are explicitly categorized into two types: (1) channel cells, which containing river channels, where both slope concentration and channel concentration processes are considered, including water exchange between the slope and the channel, and (2) non-channel cells, which do not contain river channels and only slope concentration process is considered. In channel cells, the channel concentration process is governed by one-dimensional diffusion wave equations. This allows the channel to be conceptualized as a line feature within the model, with its width treated as a cross-sectional attribute of the one-dimensional equations. The spatial relationship between the channel line feature and channel cells adopts two topologies: intersecting (channel passes through cell interiors) and adjacent (channel aligns with cell boundaries). In real-world watershed applications, the channel width is generally smaller than the grid size. Thus, we implement the intersecting topology by positioning the channel line through cell centers (Figure R1a). In the synthetic V-shaped watershed test case, where the channel width (20m) is

significantly larger than the grid resolution (5m), we adopt the adjacent topology to keep physical realism and avoid representing the channel with multiple cells (Figure R1b).

[Figure]

**Figure R1.** Diagram of spatial discretization and the arrangement of channel line features (with two spatial topological relationships between the channels and slopes).

Figure 5 in our manuscript initially presented the location of the Tunxi watershed and its gauging stations. To illustrate the spatial discretization in the Tunxi watershed, including the definition of grid (non-channel) units and channel units, we have made further extensions based on this (see Figure 5b) and the revised Figure 5 is shown below:

" Line 421:

[Figure]

Figure 5. Location and gauging station distribution of the Tunxi watershed (a), and the spatial discretization of the watershed, including channel and non-channel cells (b). "

3. Since the author emphasizes the advantages of the 2D method in capturing microtopography and

presents it as a typical case of distributed modeling, the author should at least provide some results for spatial simulations. This would better illustrate the model's advantages and provide supporting evidence for the attribution.

Specifically, the author could provide some degree of spatial validation (for example), which is highly valued in the hydrological modeling community. If this proves challenging (due to data limitations or other reasons), an alternative could be to include a test case with more varied slopes (such as two or three different slope changes in the y-direction). In this more complex test case, testing the consistency between slope variations and simulation results would better highlight the advantages of 2D modeling.

Thanks for your valuable comments. To address this, we have further included the spatial simulation results in addition to the simulated hydrograph in 2008 (Figure 9 in manuscript, it's revised version is shown below). We focused on the precipitation events from June 6th, 7th, and 8th, 2008 (just before the flood peak, with distinctive spatial precipitation distribution). On June 6th, there was almost no precipitation (Figure 9a), on June 7th, the precipitation was concentrated in the upper-left corner of the watershed (Figure 9b), and on June 8th, it was concentrated in the lower-right corner (Figure 9c). We provide the spatial simulation results for tension and free water storage on the three days. As shown in Figure 9d and 9g, the tension water storage of the watershed as a whole was not yet fully saturated, and the free water storage was almost zero. After June 7th, the tension water storage in the upper-left part of the watershed became saturated, and the rest of the watershed also became deeper due to precipitation (Figure 9e). The overall free water storage was also increased, with more significant increases in areas where precipitation was heavier, and vice versa (Figure 9h). On June 8th, the tension water in the lower-right corner of the watershed became fully saturated due to the coverage of precipitation in corresponding area (Figure 9f). For free water (Figure 9i), the storage in the lower-right part of the watershed also approached saturation, while the saturated areas in the upper part of the watershed quickly dissipated due to the faster recession of free water.

[Figure]

Figure 9. The spatial distribution of precipitation for three days in 2008 (a-c), along with the spatial distribution of tension (d-f) and free water storage (g-i) simulated by TDD-XAJ model for the three days, and the simulated hydrograph of the Tunxi watershed in 2008 using the model (j)."

Due to data limitations, we regret that we are unable to provide further spatial validation. As an alternative, and based on your suggestion, we have expanded the range of slope in the $y$-direction ($S_{oy}$). The original used $S_{oy}$ of 0.00 (single-slope case) and 0.02 (double-slope case) have been extended to 0.00, 0.01, 0.02,

0.03, 0.04, and 0.05, while the slope in the $x$-direction ($S_{ox}$) is fixed to 0.05. For each case, both 1D and 2D runoff concentration methods are applied. The 1D methods use GIS-derived flow directions, and the comparison between the calculated and real flow directions is shown in Figure R2, and they are identical only when $S_{oy}$ equals 0.00 and 0.05.

[Figure]

Figure R2. The comparison of real and GIS-derived flow direction under different $y$-directional slope ($S_{oy}$).

For surface runoff concentration, the 1D and 2D forms of diffusion wave (DW) method are used. Figure R3 and R4 illustrate the comparison of hillslope and channel outflow hydrograph for different $S_{oy}$, respectively. In the case of subsurface runoff concentration, the 1D and 2D forms of linear reservoir method (LR) are implemented. The hillslope outflow hydrograph is demonstrated in Figure R5, with channel outflow hydrograph further detailed in Figure R6. We used mean absolute error (MAE) to assess the difference between hydrographs simulated by the 1D and 2D forms of runoff concentration methods. From these figures, we can see that when the GIS-derived flow direction aligns with the actual flow direction ($S_{oy}$=0.00, and $S_{oy}$=0.05), the results of 1D form methods match those of the 2D form methods (see Figure R3a and R3f, Figure R4a and R4f, Figure R5a and R5f, Figure R6a and R6f). However, when there are discrepancies in derived and real flow directions, certain components in 1D form methods are neglected and its results show deviations.

[Figure]

**Figure R3.** The hillslope outflow hydrograph of synthetic V-catchment test case with varying $y$-directional slope ($S_{oy}$) in surface comparison scenario. The one-dimensional (1D) and two-dimensional (2D) form diffusion wave methods (DW) were evaluated in this scenario. The difference between the outflow of 1D and 2D form methods was assessed with mean absolute error (MAE).

[Figure]

**Figure R4.** The channel outflow hydrograph of synthetic V-catchment test case with varying $y$-directional slope ($S_{oy}$) in surface comparison scenario. The one-dimensional (1D) and two-dimensional (2D) form diffusion wave methods (DW) were evaluated in this scenario. The difference between the outflow of 1D and 2D form methods was assessed with mean absolute error (MAE).

[Figure]

**Figure R5.** The hillslope outflow hydrograph of synthetic V-catchment test case with varying $y$-directional slope ($S_{oy}$) in subsurface comparison scenario. The one-dimensional (1D) and two-dimensional (2D) form linear reservoir methods (LR) were evaluated in this scenario. The difference between the outflow of 1D and 2D form methods was assessed with mean absolute error (MAE).

[Figure]

**Figure R6.** The channel outflow hydrograph of synthetic V-catchment test case with varying $y$-directional slope ($S_{oy}$) in subsurface comparison scenario. The one-dimensional (1D) and two-dimensional (2D) form linear reservoir methods (LR) were evaluated in this scenario. The difference between the outflow of 1D and 2D form methods was assessed with mean absolute error (MAE).

4. The differences between Figure. 7b and Figure. 7c suggest that 1D modeling overlooks the interflow component in the y-direction, although this is not particularly clear. Could you include one or two time profile plots corresponding to different stages in Figure. 6g to further support this conclusion?

Thanks for your valuable feedback. In Figure 7, we illustrate the differences between the 1D and 2D forms of the linear reservoir method for subsurface runoff concentration through Figure 7b and Figure 7d. As you pointed out, the contrast between these two sub-figures is not very clear. To address this, we have plotted the distribution of interflow storage completely and then zoomed in on the areas where differences are more noticeable. This visualization approach helps to highlight how the 1D model overlooks the interflow component in the $y$-direction. The updated Figure 7 is shown below:

"Line 452:

[Figure]

**Figure 7.** The spatial distribution of surface water depth ($h_s$) and interflow storage ($O_i$) on the left-side hillslope of both single-slope (a-b) and double-slope (c-d) synthetic V-catchment test cases at the 60 minute mark. The state variable distributions shown are simulated using two-dimensional (2D) slope concentration methods. The corresponding results of 1D methods are identical to those obtained from the single-slope case simulated with 2D methods, regardless of the test case used. For a clear comparison, the spatial distribution of $O_i$ in the upper left corner has been zoomed."

Furthermore, we selected three points (A, B, and C, see Figure R7) from the zoomed area in the updated Figure 7. For both single-slope and double-slope cases, we generated time profile plots of the interflow storage using both the 1D and 2D linear reservoir method at these points (Figure R8). The comparative results clearly reveal the simulated differences between the 1D and 2D linear reservoir methods in the double-slope test case (Figures R8d to R8f).

[Figure]

**Figure R7**. Diagram of the spatial location for points A, B, and C.

[Figure]

**Figure R8**. Timeseries of interflow storage ($O_i$) simulated by 1D and 2D linear reservoir (LR) methods for single-slope (a-c) and double-slope (d-f) synthetic V-catchment test cases (at points shown in Figure R7).

Below are some of my suggestions regarding the writing or presentation (although the paper is already well-written):

**Specific issues:**

1. Line 13-14: Please emphasize both the 2D diffusion wave and the 2D linear reservoir method in this sentence to enhance consistency with the paper.

Thanks for your suggestion. We will revise the sentence into:

"We also introduced two-dimensional (2D) diffusion wave equations for surface slope concentration and derived 2D linear reservoir equations for subsurface slope concentration, to replace their 1D counterparts."

2. Line 58-59: Beven's alternative blueprint encompasses several key concepts, including Bayesian philosophy, model equivalence, and an emphasis on uniqueness. It would be beneficial to clarify which of these are closely related to the development of XAJ, although this may be considered a matter of fine-tuning.

Thanks for your suggestion. Beven's alternative blueprint indeed encompasses several key perspectives, including those you mentioned, as well as an evaluation on the physical-based property of distributed hydrological model (Beven, 2002). Before the alternative blueprint, physical-based model generally emphasized on applying rigorous mathematical-physical equations deriving from established physical principles under appropriate assumptions, such as different simplified form of the Navier-Stokes equations. However, the alternative blueprint, while still adhering to fundamental physical principles and conservation laws, places greater emphasis on consistency with observational data. This shift effectively relaxes the strict dependence on mathematical-physical equations, enabling the integration of conceptual parameterization equations from the lumped hydrological model, thereby expanding the available equation space for distributed hydrological model. This laid the foundation for the development of the distributed version of the XAJ model, making it closely linked to the evolution of the XAJ model.

According to your suggestion, we will revise it to:

"Although initial efforts began earlier (Lu et al., 1996), 2002 was a notable milestone due to Beven's alternative blueprint (Beven, 2002). This blueprint emphasized the importance of observational consistency as a core requirement of the physical-based property in DHMs, enabling the integration of parameterized equations from lumped hydrological models."

3. Line 342: Please clarify the model's inputs and outputs, and whether they differ from the original XAJ model.

As suggested, an appendix will be included in the revised manuscript to clarify the inputs and outputs of the TDD-XAJ model, along with an analysis of whether they differ from the original XAJ model. The appendix is provided below:

**"Appendix B: Inputs and outputs of the TDD-XAJ model**

The inputs and outputs of the TDD-XAJ model are listed in Table B1. Its inputs consists of four categories: (1) model configuration dictionary, (2) spatiotemporal multidimensional arrays of precipitation and evaporation as forcing inputs, (3) raster datasets providing spatial attributes (including elevation, river network, aspect, etc.), and (4) lookup tables storing hydraulic parameters for surface and channel routing. The outputs comprise spatiotemporal distributions of state variables and model fluxes at pre-defined temporal resolutions (instantaneous or time-averaged values). Compared to the original XAJ model, which is a lumped hydrological model, its inputs are watershed-scale parameters and timeseries for precipitation and evaporation, with output mainly including simulated discharge hydrograph. The TDD-XAJ model share largely similar input and output information with existing distributed XAJ model, the aspect information introduced by the 2D linear reservoir method can be derived from elevation raster.

**Table B1.** Detailed information on input and output files of the TDD-XAJ model.

| Class | Category | Name | Format | Description |
|-------|----------|------|--------|-------------|
| Input | Dictionary | Model configuration | YAML | (1) Paths for the other files in this table, temp and output folders (2) Numerical calculation settings (e.g., CFL coefficient) (3) Model parameters |
| | Array | Forcing | NPY | Spatiotemporal distribution of precipitation and evaporation |
| | Raster | Elevation | Esri ASCII | Use projected coordinate system, with unit of m |
| | Raster | Aspect | Esri ASCII | Derived from elevation raster via GIS analysis |
| | Raster | Flow accumulation | Esri ASCII | Derived from elevation raster via GIS analysis |
| | Raster | Flow direction | Esri ASCII | Derived from elevation raster via GIS analysis |
| | Raster | River network | Esri ASCII | Mark the location of channel cells and corresponding code |
| | Raster | Land use | Esri ASCII | Aligned with elevation raster, and store land use code |
| | Lookup table | Surface hydraulics | CSV/XLSX | Land use code and corresponding surface roughness coefficient |
| | Lookup table | Channel hydraulics | CSV/XLSX | Channel code and corresponding attributes (e.g., longitudinal slope and channel roughness coefficient) |
| Output | Array | State variables | NPY | Spatiotemporal distributions of state variables at pre-defined temporal resolutions (instantaneous or time-averaged values) |
| | Array | Model fluxes | NPY | Spatiotemporal distributions of model fluxes at pre-defined temporal resolutions (instantaneous or time-averaged values) |

"

4. Line 359: "The synthetic V-catchment, first proposed by (Overton and Brakensiek, 1970)", This appears to be an incorrect citation format.

 Thanks for pointing this out, we will revise it into:

"The synthetic V-catchment, first proposed by Overton and Brakensiek (1970)"

5. Figure 6: The lower-left corner should be "subsurface" rather than "surface."

Thank you for pointing this out, we will correct this.

6. Figure 8: Highlight the inflection.

Thanks for your suggestion. We have highlighted the inflection point in Figure 8 (a) using a red dashed box and plotted the first-order numerical derivative to emphasize this point. The updated Figure 8 is shown below:

[Figure]

**Figure 8.** The simulated hillslope outflow (a) and channel outflow (b) of the loosely-coupled (LC) model and fully-coupled (FC) model in the double-slope case. The time in parentheses denotes the time interval of input forces (Δ$T$) used for the model. The FC model uses 90 min only, while the LC model uses 90 min, 45 min, and 15 min for convergence test. The parameters used are the first set of the parameters sampled with the SLH method from their ranges listed in Table 2. The inflection point at 75 minute mark in the hillslope outflow hydrograph of the LC model (using Δ$T$=15min) is highlighted with a red dashed box, along with timeseries of the first-order numerical derivative of hillslope outflow (Δ$Q$)."

**References mentioned in the response**

Beven, K., 2002. Towards an alternative blueprint for a physically based digitally simulated hydrologic response modelling system. Hydrological Processes, 16(2): 189-206.

Lam, S. K., Pitrou, A., and Seibert, S.: Numba: a llvm-based python jit compiler, in: Proceedings of the Second Workshop on the LLVM Compiler Infrastructure in HPC, Austin, Texas, 7, https://doi.org/10.1145/2833157.2833162, 2015.

Lu, M., Kioke, T., and Hayakawa, N.: Distributed XinAnJiang model using radar measured rainfall data, in: Proceedings of International Conference on Water Resources & Environmental Research: Towards the 21st Century, Kyoto, Japan, 1996.

Murphy, B., Yurchak, R., and Müller, S.: GeoStat-framework/PyKrige (1.7.2), Zenodo [code], https://doi.org/https://doi.org/10.5281/zenodo.11360184, 2024.

Overton, D. E. and Brakensiek, D. L.: A kinematic model of surface runoff response, in: Proceedings of the Wellington Symposium, Wellington, New Zealand, 1970.

Stacke, T. and Hagemann, S.: HydroPy (v1.0): a new global hydrology model written in python, Geoscientific Model Development, 14, 7795-7816, https://doi.org/10.5194/gmd-14-7795-2021, 2021.

---

## Author Comment (AC2)

**Response to Reviewer RC2**

**Title**: Two-dimensional Differential-form of Distributed Xinanjiang Model

**Authors**: Jianfei Zhao, Zhongmin Liang\*, Vijay P. Singh, Taiyi Wen, Yiming Hu, Binquan Li, Jun Wang

**Manuscript ID**: hess-2024-377

We deeply appreciate your detailed review on our manuscript. All your comments have been carefully addressed, and a point-by-point response is provided below.

For better readability, the point-by-point response is formatted as follows:

- Reviewer's comments are shown in black
- Authors' responses are shown in blue
- Revisions to be incorporated in the revised manuscript are highlighted in red

**Overall comments:**

This manuscript not only summarizes the progress achieved during past decades but also advances the field by establishing a complete system of differential equations for the distributed XAJ model, along with its numerical implementation. The manuscript provides a thorough analysis of both one-dimensional and two-dimensional concentration methods while addressing numerical error issues. The manuscript is well-written with promising application prospects. To strengthen the manuscript, I would like to propose several comments in the order of line numbers.

Thank you very much for your concise paper summary and positive feedback on our research. We have carefully considered all of your comments and addressed them in the subsequent specific comments section. We hope that these changes could effectively address your concerns and enhance our manuscript.

**Specific comments:**

1. Line 55: It is more appropriate to describe this as "the widely-used three-water-sources lumped XAJ model."

Thanks for your suggestion. Including the term "three-water-sources" makes the sentence more precise. We will revise the sentence into:

"…, leading to the formation of the widely-used three-water-sources lumped XAJ model (Zhao, 1992)."

2. Line 56-66: While I agree the third phase highlighted the development of the distributed XAJ model,

it is important to note that the original lumped XAJ model has continued to evolve in parallel. The authors should briefly acknowledge this progression.

Thanks for pointing this out. We agree that it is important to acknowledge that the original lumped XAJ model has continued to evolve in parallel with the development of the distributed XAJ model. According to your suggestion, we will revise it to:

"**Phase 3 (2002-present):** The third phase is characterized by efforts to transition the XAJ model from a lumped to a distributed version, aligning it with other contemporary models like TOPMODEL and HBV (Beven et al., 2021; Seibert et al., 2022). It is noteworthy that the original lumped XAJ model has undergone continuous evolution alongside (Ouyang et al., 2025)."

3. Line 61: I believe the term "concentration method" is more appropriate here to maintain terminology consistency throughout the manuscript.

Thank you for your feedback regarding terminology consistency. We agree that aligning terminology strengthens clarity. In addition to Line 61, we have checked the terminology throughout the manuscript. The principle is to replace "route" with "runoff concentration" when used in conjunction with "runoff generation". According to your suggestion, we will revise it to:

"The transformation involved the application of the runoff generation modules to smaller computational units (e.g., sub-basins or grids) and implementing distributed hydrological or hydraulic runoff concentration methods (Chen et al., 2024; Fang et al., 2017; Liu et al., 2009; Su et al., 2003)."

4. Line 126: "Deeper" in Figure 1 should be corrected into "Deep", as referenced in Section 2.2.1. Besides, the slope-channel coupling diagram in this figure should be described in the text for clarity.

Thanks for pointing this out. As you suggested, we will revise it to:

"Line 122-127:
Water moves in two dimensions across the slope, with surface and subsurface runoff (interflow and groundwater) draining into the channels. The water in the channels is routed to the outlet in a one-dimensional flow pattern. Bidirectional water exchange between slope surface and channel is considered under varying hydraulic conditions. Specifically, when the channel water surface elevation is higher than that of the slope surface, water flows from the channel back onto the slope surface. The diagram of the TDD-XAJ is shown in Fig. 1.

[Figure]

**Figure 1.** Diagram of the two-dimensional differential-form of distributed Xinanjiang model."

5. Line 145: The term "slope" is recommended instead of "hillslope".

Thanks for your suggestion. We will replace "hillslope" with "slope" accordingly and revise it into:

"Instead of the one-dimensional (1D) diffusion wave equations, the TDD-XAJ model employs two-dimensional (2D) diffusion wave equations to better represent surface water flow across the slope."

6. Line 153: Given the number of equations and variables, it is suggested to include a nomenclature section.

We agree that incorporating a nomenclature section would enhance readability, and we will include it in the form of supplementary material. Thanks for your suggestion.

7. Line 343: The determination methods of 15 model parameters listed in the Table 1 requires detailed explanation. In addition, it is also necessary to explain the differences in model parameter from the original XAJ model.

Thanks for your comments. The TDD-XAJ model comprises 15 parameters, which is one parameter more than the original lumped Xinanjiang model (Zhao et al., 2023), with 11 hydrological parameters for runoff generation and 4 parameters (2 hydrological and 2 hydraulic) for runoff concentration. These 13 hydrological parameters are consistent with those from the original lumped model. The remaining hydrological parameter of the original lumped model is used to represent the channel concentration

process. The channel concentration process is represented with one-dimensional diffusional wave equations in the TDD-XAJ model, thus transforming the hydrological parameter into a hydraulic parameter (channel roughness coefficient, $n_c$). The additional hydraulic parameter (surface roughness coefficient, $n_s$) in the TDD-XAJ model is included to facilitate two-dimensional diffusional wave equations for slope surface concentration.

We have added a column to the model parameter table (Table 5), to distinguish whether the parameters are spatially uniform or spatially distributed. For the spatially distributed parameters, we introduced their determination methods in the supplementary material (see Section S2). As suggested, we will revise it to:

"Line 343-344:

The TDD-XAJ model has 15 tunable parameters, which is one more than in the original lumped Xinanjiang model (Zhao et al., 2023). The signification and value range of these parameters are listed in Table 1. Parameters that can be spatially uniform are consistent with those from the original lumped model. The methods for determining spatially distributed parameters are provided in Sect. S2 of the Supplement.

**Table 1.** Parameters and parameter range of the TDD-XAJ model.

| Components | Symbol | Signification | Range | Spatially | Unit |
|---|---|---|---|---|---|
| Evapotranspiration and runoff calculation | $K_e$ | Coefficient of potential evapotranspiration to pan evaporation | [0.6, 1.5] | Uniform | - |
| | $c$ | Coefficient of deep soil layer evapotranspiration | [0.01, 0.2] | Uniform/ Distributed | - |
| | $W_{um}$ | Tension water storage capacity of upper soil layer | [5, 30] | Uniform/ Distributed | mm |
| | $W_{lm}$ | Tension water storage capacity of lower soil layer | [60, 90] | Uniform/ Distributed | mm |
| | $W_{dm}$ | Tension water storage capacity of deep soil layer | [15, 60] | Uniform/ Distributed | mm |
| | $A_{imp}$ | The ratio of the impervious area | [0.01, 0.2] | Uniform/ Distributed | - |
| | $b$ | Tension water storage capacity curve exponent | [0.1, 0.4] | Uniform/ Distributed | - |
| Runoff separation | $S_m$ | Free water storage capacity | [10, 50] | Uniform/ Distributed | mm |
| | $ex$ | Free water storage capacity curve exponent | [1.0, 1.5] | Uniform/ Distributed | - |
| | $K_i$ | Interflow outflow coefficient | [0.10, 0.55] | Uniform/ Distributed | - |
| | $K_g$ | Groundwater outflow coefficient | $0.7 - K_i$ | Uniform/ Distributed | - |
| Slope concentration | $n_s$ | Surface roughness coefficient | [0.01, 0.80] | Distributed | m s$^{-1/3}$ |
| | $C_i$ | Interflow storage recession coefficient | [0.5, 0.9] | Uniform/ Distributed | - |
| | $C_g$ | Groundwater storage recession coefficient | [0.98, 0.998] | Uniform/ Distributed | - |
| Channel concentration | $n_c$ | Channel roughness coefficient | [0.01, 0.05] | Distributed | m s$^{-1/3}$ |

"

"**Section S2 of the Supplement:** The determination methods of spatially distributed model parameter

To determine spatially distributed model parameters, the process is generally based on spatially quantified data of watershed physical characteristics. This work is primarily carried out in two ways:

(1) **Lookup table-based method**. Parameters are determined from tables based on watershed physical

attributes. Specifically, the ratio of the impervious area ($A_{\text{imp}}$) and coefficient of deep soil layer evapotranspiration ($c$) are determined according to land use types (Yao et al., 2012), while the determination of tension water storage capacity curve exponent ($b$) and free water storage capacity curve exponent ($ex$) are assigned based on soil types. The value of surface roughness coefficient ($n_{\text{s}}$) is assigned based on the land use type of each grid cell, with different land uses corresponding to different roughness coefficients, which are derived from existing literature (Miao et al., 2016; Perrini et al., 2024). For channel roughness coefficient ($n_{\text{c}}$), values are obtained from a roughness coefficient table for river channels (Arcement and Schneider, 1989).

(2) **Physical meaning-based method**. Parameter values are calculated using quantitative watershed physical characteristics according to the physical meaning of the parameters. Specifically:

**a**. Tension water storage capacity of the upper, lower, and deep soil layer ($W_{\text{um}}$, $W_{\text{lm}}$, and $W_{\text{dm}}$). The summation of $W_{\text{um}}$, $W_{\text{lm}}$, and $W_{\text{dm}}$ represents the tension water capacity of the entire soil layer ($W_{\text{m}}$), and it can be determined according to soil hydrological parameters and soil layer depth (Yao et al., 2012), which could be expressed as:

$$W_{\text{m}} = (\theta_{\text{f}} - \theta_{\text{r}})D_{\text{s}}, \tag{S4}$$

where $\theta_{\text{f}}$ is field capacity, $\theta_{\text{r}}$ is residual water content, $D_{\text{s}}$ is soil layer depth (mm). Subsequently, two watershed-scale uniform coefficients ($K_{\text{um}}$ and $K_{\text{lm}}$) and their derived value ($1 - K_{\text{um}} - K_{\text{lm}}$) are used to divide $W_{\text{m}}$ into $W_{\text{um}}$, $W_{\text{lm}}$, and $W_{\text{dm}}$ accordingly, which are given as:

$$W_{\text{um}} = W_{\text{m}}K_{\text{um}}, \tag{S5}$$

$$W_{\text{lm}} = W_{\text{m}}K_{\text{lm}}, \tag{S6}$$

$$W_{\text{dm}} = W_{\text{m}}(1 - K_{\text{um}} - K_{\text{lm}}). \tag{S7}$$

**b**. Free water storage capacity ($S_{\text{m}}$). $S_{\text{m}}$ usually represents the capacity of free water in the humus layer. Thus, it can be determined according to soil hydrological parameters and the humus layer depth (Yao et al., 2012), which could be expressed as:

$$S_{\text{m}} = (\theta_{\text{s}} - \theta_{\text{f}})D_{\text{h}}, \tag{S8}$$

where $\theta_{\text{s}}$ is saturated water content, $\theta_{\text{f}}$ is field capacity, $D_{\text{h}}$ is humus layer depth (mm).

**c**. Interflow and groundwater outflow coefficient ($K_{\text{i}}$ and $K_{\text{g}}$). $K_{\text{i}}$ and $K_{\text{g}}$ represent the outflow rate of interflow and groundwater. The method for determining $K_{\text{i}}$ and $K_{\text{g}}$ involves converting the free water storage to corresponding saturated water depth, based on the hillslope storage-discharge theory and steady-state assumptions, which is then multiplied by the slope gradient and saturated hydraulic conductivity using the kinematic wave assumption (Tong, 2022). $K_{\text{i}}$ and $K_{\text{g}}$ are finally expressed as the ratios of corresponding flow distance in the time interval of input forces to the slope length, which could be given as:

$$K_{\text{i}} = \frac{2S_0 K_{\text{su}} S_{\text{hill}} \Delta T}{1000(\theta_{\text{s}} - \theta_{\text{f}})L_{\text{hill}}^2}, \tag{S9}$$

$$K_{\mathrm{g}} = \frac{2S_0 K_{\mathrm{sl}} S_{\mathrm{hill}} \Delta T}{1000(\theta_{\mathrm{s}} - \theta_{\mathrm{f}}) L_{\mathrm{hill}}^2}, \tag{S10}$$

where $S_0$ is free water storage (mm), $K_{\mathrm{su}}$ and $K_{\mathrm{sl}}$ is saturated hydraulic conductivity of the upper (representing interflow) and lower (representing groundwater) soil layer respectively (m s$^{-1}$), $S_{\mathrm{hill}}$ is the gradient of the slope, $\Delta T$ is the time interval of input forces (s), and $L_{\mathrm{hill}}$ is the length of the slope (m).

**d**. Interflow and groundwater storage recession coefficient ($C_{\mathrm{i}}$ and $C_{\mathrm{g}}$). $C_{\mathrm{i}}$ and $C_{\mathrm{g}}$ represent the time delay for interflow and groundwater runoff as they travel from specific locations on the slope to the river channel. These parameters are determined based on the theory of spatially distributed unit hydrograph (Maidment et al., 1996; Tong, 2022). The grid cells that form the flow path extending from specific locations on the slope to the river channel is first identified using GIS. Then, using the kinematic wave assumption, the flow velocity of interflow and groundwater runoff through each grid cell is computed based on the saturated hydraulic conductivity of the upper and lower layers and the slope gradient. Finally, the time taken for flow through each grid cell is accumulated, which could be expressed as:

$$T_{\mathrm{i}} = \sum_{j=1}^{N_{\mathrm{hill}}} L_{\mathrm{hill}}^j / \left(K_{\mathrm{su}}^j S_{\mathrm{hill}}^j\right), \tag{S11}$$

$$T_{\mathrm{g}} = \sum_{j=1}^{N_{\mathrm{hill}}} L_{\mathrm{hill}}^j / \left(K_{\mathrm{sl}}^j S_{\mathrm{hill}}^j\right), \tag{S12}$$

where $T_{\mathrm{i}}$ and $T_{\mathrm{g}}$ is the accumulated travel time from specific locations on the slope to the river channel through interflow and groundwater respectively (s), $N_{\mathrm{hill}}$ is the count of grid cells that form the flow path. $C_{\mathrm{i}}$ and $C_{\mathrm{g}}$ for each grid cell are further derived using theoretical conversion, which could be given as:

$$C_{\mathrm{i}} = \exp(-\Delta T / T_{\mathrm{i}}), \tag{S13}$$

$$C_{\mathrm{g}} = \exp\left(-\Delta T / T_{\mathrm{g}}\right). \tag{S14}$$

The primary data used to determine spatially distributed model parameters include soil physical and hydraulic properties, slope gradient, and land use. These can be obtained from open-source datasets, such as Harmonized World Soil Database v2.0 (HWSD v2.0) (FAO and IIASA, 2023), China dataset of soil properties for land surface modelling version 2 (CSDLv2) (Shi et al., 2025), and Global land cover mapping at 30m resolution (GlobeLand30) (Chen et al., 2015)."

8. Line 439: The equation of FVRE should be provided. Similarly, it is also suggested to provide the formulas of three channel cross-sectional hydraulic elements mentioned in Line 282.

Thanks for your comment. As you suggested, the equation of FVRE is provided below:

"The equation of FVRE could be given as:

$$FVRE = \left(\sum_{i=1}^{n} Q_{sim,i} - \sum_{i=1}^{n} Q_{obs,i}\right) / \sum_{i=1}^{n} Q_{obs,i} * 100\% \tag{32}$$

where $Q_{sim,i}$ and $Q_{obs,i}$ represent simulated and observed discharge at time step i (m$^3$ s$^{-1}$), and $n$ is the length of the sequence."

Furthermore, we will provide the formulas of three channel cross-sectional hydraulic elements (cross-sectional area, water surface width, and wetted perimeter) in supplementary material (see Section S1).

"**Section S1 of the Supplement:** Cross-sectional generalization and hydraulic parameters of river channel

In the two-dimensional differential-form of distributed Xinanjiang model (TDD-XAJ), the cross-section of river channel is generalized into a trapezoid (Fig. S1). Thus, the formulas for cross-sectional area, water surface width, and channel wetted perimeter could be given as:

$$A = \begin{cases} \varsigma h_c + h_c{}^2/\tan\beta & 0<\beta<90° \\ \varsigma h_c & \beta = 90° \end{cases}, \tag{S1}$$

$$B = \begin{cases} \varsigma + 2h_c/\tan\beta & 0<\beta<90° \\ \varsigma & \beta = 90° \end{cases}, \tag{S2}$$

$$\chi = \begin{cases} \varsigma + 2h_c/\sin\beta & 0<\beta<90° \\ \varsigma + 2h_c & \beta = 90° \end{cases}, \tag{S3}$$

where $A$ is cross-sectional area (m$^2$), $B$ is water surface width (m), $\chi$ is channel wetted perimeter (m), $\varsigma$ is channel bottom width (m), $h_c$ is channel water depth (m), $\beta$ is river bank slope gradient (°).

[Figure]

**Figure S1.** Diagram of trapezoidal cross-sectional generalization of river channel."

9. Line 452: In the slope concentration methods comparison experiment, the authors systematically compared the 1D and 2D forms of the diffusion wave and linear reservoir methods based on idealized test cases. For the diffusion wave method, significant differences were observed between the 1D and 2D form, both in terms of hydrographs and surface storage. However, for the linear reservoir method, while the differences in hydrographs were noticeable (Figure 6g and 6h), the contrast in storage was less evident (Figure 7b and 7d). The authors should improve the visualization approach for Figure 7 such as by changing color schemes to make the comparison more clear.

We are sorry for not clearly presenting the differences between the spatial distributions of interflow

storage simulated by the 1D and 2D linear reservoir methods, as also noted by other reviewers. Following the suggestion of reviewer CC1, we provide the complete spatial distribution of interflow storage and then zoom in on areas with significant differences. This approach allows for a clearer illustration while preserving data integrity. The revised Figure 7 is shown below:

[Figure]

**Figure 7.** The spatial distribution of surface water depth ($h_s$) and interflow storage ($O_i$) on the left-side hillslope of both single-slope (a-b) and double-slope (c-d) synthetic V-catchment test cases at the 60 minute mark. The state variable distributions shown are simulated using two-dimensional (2D) slope concentration methods. The corresponding results of 1D methods are identical to those obtained from the single-slope case simulated with 2D methods, regardless of the test case used. For a clear comparison, the spatial distribution of $O_i$ in the upper left corner has been zoomed."

10. Line 547-565: The authors analyzed the model's performance by applying it to the Tunxi watershed and examining the flow hydrograph at the outlet station, and the overall simulation results were satisfactory. However, as a distributed hydrological model, the authors should provide more details regarding the spatial simulation. Furthermore, it is suggested that the authors could include comparative results from stations within the watershed, if possible, as this would provide a more comprehensive evaluation of the performance of the TDD-XAJ model.

Thanks for your feedback on enhancing the spatial evaluation of the TDD-XAJ model. To address this, we have further included the spatial simulation results in addition to the simulated hydrograph in 2008 (Figure 10). Moreover, to strengthen the assessment of model performance, we introduced the Yuetan hydrological station—a station within the Tunxi watershed (Figure 5), and compared its simulation results

with observed data. Details of performance metrics are provided in Table 4. As shown in Table 4, the average values of the Nash-Sutcliffe efficiency (NSE), Kling-Gupta efficiency (KGE), the absolute flood volume relative error (|FVRE|), and the coefficient of determination ($R^2$) for Yuetan station (across all years) are 0.83, 0.78, 6.2%, and 0.86, respectively. The corresponding values for Tunxi station are 0.87, 0.80, 6.7%, and 0.90. In summary, these metrics indicate that the TDD-XAJ model provides robust streamflow simulations at both stations in the Tunxi watershed. The revisions to be implemented are shown below:

"Line 421:

[revised manuscript text omitted]

11. Line 591-595: In the final paragraph of the conclusion, the authors summarize the limitations of this study. The manuscript encompasses extensive research efforts. I understand that the journey from proposing a model to its refinement and maturity is a lengthy process, and this manuscript has done an excellent job methodologically, providing a solid foundation for future application and research. I recommend relocating this paragraph to the discussion part, where the potential application scenarios and future research directions of the model can be further explored.

Thanks for your encouragement. As suggested, we will add a section in Result and Discussion part of the revised manuscript, which is shown below:

"4.5 Limitations and further research
The main limitation of the TDD-XAJ model is that it addresses only the numerical errors on the time scale from the ODE's numerical solution, while neglecting errors arising from the spatial discretization of the PDE via the FVM method. Future research should consider using methods like manufactured solutions

(Bisht and Riley, 2019) as an alternative for evaluating numerical errors arising from both spatial and temporal discretization when exact solutions are difficult to obtain. Additionally, exploring more advanced spatial discretization techniques—such as second-order FVM or discontinuous Galerkin method (Shaw et al., 2021)—and more sophisticated temporal integration methods could help control numerical errors more effectively.

In terms of computational efficiency, although we make extensive use of the NumPy library—which is generally faster than native Python code— we acknowledge that NumPy may still be slower in certain operations (e.g., element-wise calculations) compared to compiled languages. This is a well-known limitation within the Python community, and solutions like Just-In-Time (JIT) compilation have been proposed (Lam et al., 2015), which convert frequently executed script code into machine code with further automatic optimizations. Although this manuscript primarily focuses on presenting the theoretical aspects of the TDD-XAJ model, we plan to optimize the code—potentially including parallelization—in future work.

It is essential to validate the model with more real-world watershed data and evaluate its uncertainty, particularly for watersheds with varying scales, diverse underlying physical conditions, and hydrological data at different temporal resolutions. The differential-form mathematical equations established for the TDD-XAJ model provide a solid foundation for future research, including combining deep learning for better model parameterization and process understanding (Höge et al., 2022; Li et al., 2024)."

**References mentioned in the response**

Arcement, G. J. and Schneider, V. R.: Guide for selecting manning's roughness coefficients for natural channels and flood plains, 2339, https://doi.org/10.3133/wsp2339, 1989.

Bisht, G. and Riley, W. J.: Development and verification of a numerical library for solving global terrestrial multiphysics problems, J. Adv. Model. Earth Syst., 11, 1516-1542, https://doi.org/10.1029/2018MS001560, 2019.

Beven, K. J., Kirkby, M. J., Freer, J. E., and Lamb, R.: A history of TOPMODEL, Hydrol. Earth Syst. Sci., 25, 527-549, https://doi.org/10.5194/hess-25-527-2021, 2021.

Chen, J., Chen, J., Liao, A., Cao, X., Chen, L., Chen, X., He, C., Han, G., Peng, S., Lu, M., Zhang, W., Tong, X., and Mills, J.: Global land cover mapping at 30m resolution: A POK-based operational approach, ISPRS-J. Photogramm. Remote Sens., 103, 7-27, https://doi.org/10.1016/j.isprsjprs.2014.09.002, 2015.

Chen, L., Deng, J., Yang, W., and Chen, H.: Hydrological modelling of large-scale karst-dominated basin using a grid-based distributed karst hydrological model, J. Hydrol., 628, 130459, https://doi.org/10.1016/j.jhydrol.2023.130459, 2024.

FAO and IIASA: Harmonized world soil database (version 2.0) [dataset], https://www.fao.org/soils-portal/data-hub/soil-maps-and-databases/harmonized-world-soil-database-v20/en/, 2023.

Fang, Y. H., Zhang, X., Corbari, C., Mancini, M., Niu, G. Y., and Zeng, W.: Improving the Xin'anjiang hydrological model based on mass–energy balance, Hydrol. Earth Syst. Sci., 21, 3359-3375, https://doi.org/10.5194/hess-21-3359-2017, 2017.

Höge, M., Scheidegger, A., Baity-Jesi, M., Albert, C., and Fenicia, F.: Improving hydrologic models for predictions and process understanding using neural odes, Hydrol. Earth Syst. Sci., 26, 5085-5102, https://doi.org/10.5194/hess-26-5085-2022, 2022.

Lam, S. K., Pitrou, A., and Seibert, S.: Numba: a llvm-based python jit compiler, in: Proceedings of the Second Workshop on the LLVM Compiler Infrastructure in HPC, Austin, Texas, 7, https://doi.org/10.1145/2833157.2833162, 2015.

Li, B., Sun, T., Tian, F., Tudaji, M., Qin, L., and Ni, G.: Hybrid hydrological modeling for large alpine basins: a semi-distributed approach, Hydrol. Earth Syst. Sci., 28, 4521-4538, https://doi.org/10.5194/hess-28-4521-2024, 2024.

Liu, J., Chen, X., Zhang, J., and Flury, M.: Coupling the Xinanjiang model to a kinematic flow model based on digital drainage networks for flood forecasting, Hydrol. Process., 23, 1337-1348, https://doi.org/10.1002/hyp.7255, 2009.

Maidment, D. R., Olivera, F., Calver, A., Eatherall, A., and Fraczek, W.: Unit hydrograph derived from a spatially distributed velocity field, Hydrol. Process., 10, 831-844, https://doi.org/10.1002/(SICI)1099-1085(199606)10:6<831::AID-HYP374>3.0.CO;2-N, 1996.

Miao, Q., Yang, D., Yang, H., and Li, Z.: Establishing a rainfall threshold for flash flood warnings in China's mountainous areas based on a distributed hydrological model, J. Hydrol., 541, 371-386, https://doi.org/10.1016/j.jhydrol.2016.04.054, 2016.

Ouyang, W., Ye, L., Chai, Y., Ma, H., Chu, J., Peng, Y., and Zhang, C.: A differentiable, physics-based hydrological model and its evaluation for data-limited basins, J. Hydrol., 649, 132471, https://doi.org/j.jhydrol.2024.132471, 2025.

Perrini, P., Cea, L., Chiaravalloti, F., Gabriele, S., Manfreda, S., Fiorentino, M., Gioia, A., and Iacobellis, V.: A runoff-on-grid approach to embed hydrological processes in shallow water models, Water Resour. Res., 60, e2023WR036421, https://doi.org/10.1029/2023WR036421, 2024.

Seibert, J., Bergström, S., and Sveriges, L.: A retrospective on hydrological catchment modelling based on half a century with the HBV model, Hydrol. Earth Syst. Sci., 26, 1371-1388, https://doi.org/10.5194/hess-26-1371-2022, 2022.

Shaw, J., Kesserwani, G., Neal, J., Bates, P., and Sharifian, M. K.: LISFLOOD-FP 8.0: the new discontinuous Galerkin shallow-water solver for multi-core CPUs and GPUs, Geosci. Model Dev., 14, 3577-3602, https://doi.org/10.5194/gmd-14-3577-2021, 2021.

Shi, G., Sun, W., Shangguan, W., Wei, Z., Yuan, H., Li, L., Sun, X., Zhang, Y., Liang, H., Li, D., Huang, F., Li, Q., and Dai, Y.: A China dataset of soil properties for land surface modelling (version 2, CSDLv2), Earth Syst. Sci. Data, 17, 517-543, https://doi.org/10.5194/essd-17-517-2025, 2025.

Su, B., Kazama, S., Lu, M., and Sawamoto, M.: Development of a distributed hydrological model and its application to soil erosion simulation in a forested catchment during storm period, Hydrol. Process., 17, 2811-2823, https://doi.org/10.1002/hyp.1435, 2003.

Tong, B.: Fine-scale rainfall-runoff processes simulation using grid Xinanjiang (grid-XAJ) model, Hohai University, Nanjing, Jiangsu, 2022.

Yao, C., Li, Z., Yu, Z., and Zhang, K.: A priori parameter estimates for a distributed, grid-based Xinanjiang model using geographically based information, J. Hydrol., 468-469, 47-62, https://doi.org/10.1016/j.jhydrol.2012.08.025, 2012.

Zhao, J., Duan, Y., Hu, Y., Li, B., and Liang, Z.: The numerical error of the Xinanjiang model, J. Hydrol., 619, 129324, https://doi.org/10.1016/j.jhydrol.2023.129324, 2023.

Zhao, R.: The Xinanjiang model applied in China, J. Hydrol., 135, 371-381, https://doi.org/10.1016/0022-1694(92)90096-E, 1992.

---

## Author Comment (AC3)

**Response to Reviewer CC1**

Title: Two-dimensional Differential-form of Distributed Xinanjiang Model Authors: Jianfei Zhao, Zhongmin Liang\*, Vijay P. Singh, Taiyi Wen, Yiming Hu, Binquan Li, Jun Wang Manuscript ID: hess-2024-377

Thank you very much for your interest and valuable suggestions regarding our manuscript. All your comments have been carefully addressed, and a point-by-point response is provided below.

For better readability, the point-by-point response is formatted as follows:

- Reviewer's comments are shown in black
- Authors' responses are shown in blue
- Revisions to be incorporated in the revised manuscript are highlighted in red

**Overall comments:**

In the study, a two-dimensional differential-form of distributed Xinanjiang Model was developed. This work is interesting and valuable. Through applying the proposed model, a good performance is achieved. But there are still some points that should be explained or revised before publication.

We appreciate your recognition of both the value of our work and the performance achieved by our proposed model. We have carefully considered all of your comments and responded to them in the subsequent specific comments section. We hope that these changes could effectively address your concerns.

**Specific comments:**

1. There are many parameters in the proposed TDD-XAJ model, the authors should state the method for parameter calibration. If the calibration is done as stated in Line 432 (calibrated manually), a lot of work should done.

Thanks for your comments. The TDD-XAJ model comprises 15 parameters, which is one parameter more than the original lumped Xinanjiang model (Zhao et al., 2023), with 11 hydrological parameters for runoff generation and 4 parameters (2 hydrological and 2 hydraulic) for runoff concentration. The two hydraulic parameters are surface roughness coefficient ( $n_s$ ) and channel roughness coefficient ( $n_c$ ). For  $n_s$ , values are assigned based on the land use type of each grid cell, with different land uses corresponding to different roughness coefficients, which are derived from existing literature (Miao et al., 2016; Perrini et

al., 2024). For  $n_c$ , values are obtained from a roughness coefficient table for river channels (Arcement and Schneider, 1989). Since this manuscript primarily focuses on the theoretical aspects of the TDD-XAJ model, we adopted uniform watershed-scale parameter values to simplify the research, thereby keeping the calibration workload manageable.

To determine spatially heterogeneous hydrological parameters, the process is generally based on spatially quantified data of watershed physical characteristics. This work is primarily carried out in two ways:

- (1) Lookup table-based method. Parameters are determined from tables based on watershed physical attributes. Specifically, the ratio of the impervious area  $(A_{imp})$  and coefficient of deep soil layer evapotranspiration (*c*) are determined according to land use types (Yao et al., 2012), while the determination of tension water storage capacity curve exponent (*b*) and free water storage capacity curve exponent (*ex*) are assigned based on soil types.
- (2) **Physical meaning-based method**. Parameter values are calculated using quantitative watershed physical characteristics according to the physical meaning of the parameters. Specifically:

a. Tension water storage capacity of the upper, lower, and deep soil layer ( $W_{um}$ ,  $W_{lm}$ , and  $W_{dm}$ ). The summation of  $W_{um}$ ,  $W_{lm}$ , and  $W_{dm}$  represents the tension water capacity of the entire soil layer ( $W_m$ ), which can be calculated by the difference between field capacity ( $\theta_f$ ) and residual water content ( $\theta_r$ ) and multiplying the soil layer depth ( $D_s$ , with the unit of mm). Subsequently, two watershed-scale uniform coefficients ( $K_{um}$  and  $K_{lm}$ ) and their derived value (1- $K_{um}$ - $K_{lm}$ ) are used to divide  $W_m$  into  $W_{um}$ ,  $W_{lm}$ , and  $W_{dm}$  accordingly (Yao et al., 2012).

**b**. Free water storage capacity ( $S_m$ ).  $S_m$  usually represents the capacity of free water capacity in the humus layer. It is calculated by multiplying humus layer depth ( $D_h$ , with the unit of mm) and the difference between saturated water content ( $\theta_s$ ) and field capacity ( $\theta_f$ ), as also described by Yao et al. (2012).

**c**. Interflow and groundwater outflow coefficient ( $K_i$  and  $K_g$ ).  $K_i$  and  $K_g$  represent the outflow rate of interflow and groundwater. The method for determining these parameters involves converting the storage of interflow and groundwater linear reservoir to corresponding saturated water depth, based on the hillslope storage-discharge theory and steady-state assumptions. These are then multiplied by the slope gradient and the saturated hydraulic conductivity of the upper (representing interflow) and lower (representing groundwater) layers, using the kinematic wave assumption (Tong, 2022).  $K_i$  and  $K_g$  are finally expressed as the ratios of corresponding flow distance in the time interval of input forces to the slope length.

**d**. Interflow and groundwater storage recession coefficient ( $C_i$  and  $C_g$ ).  $C_i$  and  $C_g$  represent the time delay for interflow and groundwater runoff as they travel from specific locations on the slope to the river channel. These parameters are determined based on the theory of spatially distributed unit hydrograph (Maidment et al., 1996). The grid cells that form the flow path extending from specific

locations on the slope to the river channel is first identified using GIS. Then, using the kinematic wave assumption, the flow velocity of interflow and groundwater runoff through each grid cell is computed based on the saturated hydraulic conductivity of the upper and lower layers and the slope gradient. Finally, the time taken for flow through each grid cell is accumulated, and the parameters for each grid cell are derived using theoretical conversion formulas (Tong, 2022).

e. The remaining parameter is coefficient of potential evapotranspiration to pan evaporation ( $K_e$ ), which is usually treated as watershed-scale uniform parameters.

The primary data used to determine spatially heterogeneous model parameters include soil physical and hydraulic properties, slope gradient, and land use. These can be obtained from open-source datasets, such as Harmonized World Soil Database v2.0 (HWSD v2.0) (FAO and IIASA, 2023), China dataset of soil properties for land surface modelling version 2 (CSDLv2) (Shi et al., 2025), and Global land cover mapping at 30m resolution (GlobeLand30) (Chen et al., 2015). In addition to manual calibration, uniform watershed-scale parameters or coefficients can also be determined using automated optimization algorithms, such as the Covariance Matrix Adaptation Evolution Strategy (CMA-ES) (Hansen et al., 2003). We plan to integrate this approach into the TDD-XAJ model in future developments.

2. Line 423, Daily scale hydrological data were used in the study. I think the constructed model can be used for flood events simulation. Why don't you attempt to use sub-daily hydrological data?

The governing equations of the TDD-XAJ model are transformed into a system of ordinary differential equations after spatial discretization. The model can generate both instantaneous and time-averaged values of state variables and fluxes over a specified time interval through numerical integration, offering flexibility in model temporal resolution. As a result, the model is well-suited for daily-scale continuous simulations as well as flood event-based simulations (which usually use sub-daily data), as you mentioned in your comment. We have collected sufficiently long daily-scale hydrological data (spanning 2007-2019, totaling 13 year), but we did not gather enough sub-daily scale hydrological data. Consequently, this study relies on daily-scale hydrological data. In future work, we plan to collect additional sub-daily data as a foundation for exploring flood events simulation.

3. Line 456, the spatial distribution of the Oi has been zoomed into the upper left corner. I suggest the authors provide the spatial distribution of the entire area, and then zoom the upper left corner.

Thank you for your suggestion. This visualization approach will allow us to present the full spatial distribution of the  $O_i$ , ensuring data integrity while also highlighting the differences in the upper left corner. As reviewer RC1 noted, the contrast between Figures 7b and 7d was unclear, and your

recommended visualization approach effectively addresses this issue. As you suggested, the revised Figure 7 is shown below:

"Line 452-457:

**Figure 7.** The spatial distribution of surface water depth  $(h_s)$  and interflow storage  $(O_i)$  on the left-side hillslope of both single-slope (a-b) and double-slope (c-d) synthetic V-catchment test cases at the 60 minute mark. The state variable distributions shown are simulated using two-dimensional (2D) slope concentration methods. The corresponding results of 1D methods are identical to those obtained from the single-slope case simulated with 2D methods, regardless of the test case used. For a clear comparison, the spatial distribution of  $O_i$  in the upper left corner has been zoomed."

4. In Table 2, average MAE statistics of model fluxes for a total of 500 parameter sets are provided using loosely coupled model. But the reference is the fully-coupled model. This cannot illustrate the better performance of fully-coupled model.

The main difference between the loosely-coupled (LC) and fully-coupled (FC) model lies in their numerical implementation frameworks. In the LC model, the difference-form equations for runoff generation from the original lumped XAJ model are directly adopted, which are derived based on the time interval of input force ( $\Delta T$ ). However, for runoff concentration, the LC model uses differential-form equations; consequently, the generated runoff components (surface runoff, interflow, and groundwater runoff) are averaged over  $\Delta T$  to determine the input intensities for the following runoff concentration and runoff concentration, solving both processes simultaneously as a system of ordinary differential equations (ODEs). In the FC model, the total amount of input force, rather than further calculated runoff components,

is averaged over  $\Delta T$ .

We compared the LC model and FC model on single-slope and double-slope synthetic V-shaped watershed test cases using the same 500 parameter sets. An analytical solution exists for the total amount of surface runoff ( $R_s^*$ ). When the  $\Delta T$  was set to 90 minutes, the average mean absolute error (MAE) for  $R_s^*$  in the LC model was 4.57 mm, compared to  $2.84 \times 10^{-4}$  mm for the FC model. As  $\Delta T$  was reduced to 45 minutes and 15 minutes, the average MAE for  $R_s^*$  in the LC model decreased to 1.21 mm and 0.14 mm, respectively; however, these errors remain significantly higher than those of the FC model.

For hillslope or channel outflow, no analytical solution is available, which makes direct comparison challenging. To address this, we evaluated the convergence of the LC model by progressively reducing  $\Delta T$ . The difference-form runoff generation equations used by the LC model have first-order temporal accuracy, and the FC model provides a high-order approximation of the analytical solution. Theoretically, as  $\Delta T$  decreases, the results of the LC model should converge to those of the FC model. We used MAE to evaluate the consistency between the hillslope and channel outflow hydrographs simulated by the FC and LC models. Our numerical experiment showed that the average MAE decreases as  $\Delta T$  is reduced, indicating that the LC model's results converge toward those of the FC model. Furthermore, significant numerical errors could be observed in the LC model ( $\Delta T$ =90 minutes), whether benchmarked against the LC model ( $\Delta T$ =15 minutes) or the FC model ( $\Delta T$ =90 minutes). In the single-slope test case, when using the LC model ( $\Delta T$ =15 minutes) as the benchmark, the average MAE for channel and hillslope outflow are 1.82 mm and 1.64 mm, respectively, whereas when using the FC model ( $\Delta T$ =90 minutes) as the benchmark, the average MAE are 1.85 mm and 1.68 mm. In the double-slope test case, when using the LC model ( $\Delta T$ =15 minutes) as the benchmark, the average MAE for channel and hillslope outflow are 1.83 mm and 1.68 mm, respectively, while benchmarked against the FC model ( $\Delta T$ =90 minutes) yields average MAE of 1.87 mm and 1.72 mm. Additionally, the outflow hydrograph simulated by the LC model exhibits non-physical steady states and inflection points, which indicate its potential limitations in capturing transient behaviors.

Overall, when an analytical solution is available, the error of the LC model is several orders of magnitude higher than that of the FC model. In cases without an analytical solution, as  $\Delta T$  decreases, results of the LC model converge to those of the FC model. Furthermore, non-physical steady states and inflection points are observed in the hydrograph simulated by the LC model. Consequently, the FC model is considered to performs better in numerical simulations.

5. Line 505, the 2 values cannot be found in Table 2.

Thank you for pointing this out. To keep Table 2 concise, we initially approximated the very small values  $(4.19 \times 10^{-3} \text{ and } 2.84 \times 10^{-4})$  as 0 and used "~0.00" to maintain alignment in the column. The exact values are provided in the following lines for clarity. To avoid any misunderstanding, we have now included the values in scientific notation directly in Table 2. The revised Table is shown below:

**"Line 500:**

| Model               | ΔT
(min) | Statistics | MAE (mm)              | MAE (m 3 /s) |              |                 |              |  |  |
|---------------------|-------------|------------|-----------------------|-------------------------|--------------|-----------------|--------------|--|--|
|                     |             |            | R s *      | Hillslop                | e outflow    | Channel outflow |              |  |  |
|                     |             |            |                       | Single-slope            | Double-slope | Single-slope    | Double-slope |  |  |
| Loosely-
coupled | 90          | Max        | 5.93                  | 2.19                    | 2.20         | 2.01            | 2.06         |  |  |
|                     |             | Average    | 4.57                  | 1.85                    | 1.87         | 1.68            | 1.72         |  |  |
|                     | 45          | Max        | 2.01                  | 1.60                    | 1.62         | 1.31            | 1.41         |  |  |
|                     |             | Average    | 1.21                  | 0.81                    | 0.82         | 0.70            | 0.73         |  |  |
|                     | 15          | Max        | 0.33                  | 0.56                    | 0.61         | 0.36            | 0.39         |  |  |
|                     |             | Average    | 0.14                  | 0.16                    | 0.17         | 0.11            | 0.12         |  |  |
| Fully-
coupled   | 90          | Max        | 4.19×10 -3 | a                       | —            | —               | —            |  |  |
|                     |             | Average    | 2.84×10-4             |                         |              |                 |              |  |  |

Table 2. MAE statistics of model fluxes in numerical implementation comparison experiment.

a. The results of the fully-coupled model are used as references to calculate the MAE values for hillslope and channel outflow, so the corresponding value is empty."

6. The simulation in the Tunxi watershed was only compared with 1 previous study in the same watershed. Is it possible to compare the results with previous research using other lumped or distributed models in the same or adjacent watershed?

Thank you for your suggestion. For the present study, we focused on comparing our simulation in the Tunxi watershed with a well-documented previous study in the same area to ensure consistency in benchmark data. The primary focus of this study is on the theoretical aspect of the TDD-XAJ model. We acknowledge that further validation, including comparisons with other models, remains necessary. However, difficulties in data availability, model structure, and parameterization among studies made such a comparison challenging at this stage. While inter-model comparison was not implemented, validation targeting a hydrological station within the Tunxi Watershed was executed to strengthen model performance evaluation.

We introduced the Yuetan hydrological station—a station within the Tunxi watershed (Figure 5), and compared its simulation results with observed data. Details of performance metrics are provided in Table 4. As shown in Table 4, the average values of the Nash-Sutcliffe efficiency (NSE), Kling-Gupta efficiency (KGE), the absolute flood volume relative error (|FVRE|), and the coefficient of determination ( $R^2$ ) for Yuetan station (across all years) are 0.83, 0.78, 6.2%, and 0.86, respectively. The corresponding values for Tunxi station are 0.87, 0.80, 6.7%, and 0.90. In summary, these metrics indicate that the TDD-XAJ

model provides robust streamflow simulations at both stations in the Tunxi watershed. The revisions to be implemented are detailed below:

---

## Author Response (AR1)

**Response to comments on Manuscript hess-2024-377**

**Title**: Two-dimensional Differential-form of Distributed Xinanjiang Model
**Authors**: Jianfei Zhao, Zhongmin Liang*, Vijay P. Singh, Taiyi Wen, Yiming Hu, Binquan Li, Jun Wang
**Manuscript ID**: hess-2024-377

Dear Editor and Reviewers,

Please find enclosed our responses to the manuscript assessment entitled "Two-dimensional Differential-form of Distributed Xinanjiang Model".

We sincerely appreciate the Editor, the two Referees, and the community reviewer for their invaluable support, critical comments and constructive suggestions, as well as for providing us with the opportunity to revise our work. Upon the comments and suggestions, we have made a very careful revision of the manuscript accordingly. The major changes are summarized as follows:

(1) We have added Section 4.4 to clarify the computational efficiency of the TDD-XAJ model we proposed. This section explains the deficiency of Python language is mitigated by using NumPy library, and it also provides additional runtime information of the numerical experiment. Two appendices are included to detail the model's implementation details and its input/output files.

(2) We have investigated the spatial simulation performance of the TDD-XAJ model in three ways: evaluating the hydrograph simulation ability for station within the application watershed, providing the spatial distribution of model simulated state variables, and conducting further validation using more complex numerical test cases.

(3) We have added Section 4.5 to discuss the limitations of the current study and outline potential future research works. The relevant content in previous conclusion section has been relocated here and expanded upon. This section also includes a discussion around the model implementation.

(4) We have added a Supplement, which primarily includes the method for determining model parameters under spatially distributed condition, as well as a nomenclature to clarify the mathematical variables used in the manuscript. Additionally, Figure 7 and Figure 8 have been updated for improved clarity.

We believe the quality of the manuscript has significantly improved through this review process. We are also happy to address any additional comments.

For better readability, the point-by-point response is formatted as follows:
- Original comments are shown in black. The nth comment from the Editor, Referee 1, Referee 2, and Community 1 are numbered as EC-n, RC1-n, RC2-n, and CC1-n, respectively.

- Authors' responses are shown in blue
- Revisions are highlighted in red
- Figures used only in this response will have the prefix "R", such as Figure R1
- Figures without prefix "R" are numbered as they appear in the manuscript

We thank you for your consideration,

Jianfei Zhao (First contact)

Email: jianfei.zhao@hhu.edu.cn

on behalf of all co-authors

and

Zhongmin Liang (Corresponding author)

Email: zmliang@hhu.edu.cn

**Response to Editor:**

EC-1. Thank you for submitting your manuscript. I have received very detailed comments from two Referees, which provide valuable suggestions on the improvement of the manuscript. I would like to invite you to revise your manuscript. The revised version will be sent out for another round of review to make sure it meets the requirements of the Referees. Thanks.

We would like to express our sincere gratitude to you for your timely handing of our manuscript and for the opportunity to revise it. We have carefully addressed the comments from two Referees and the community in a point-by-point manner. The revised manuscript, including a tracked version highlighting the changes and a clean version without highlights, will be submitted alongside this response file.

Thank you for your patience while we made the revisions. We believe that our manuscript has significantly improved through this review process, and we remain open to addressing any further comments that could enhance its quality. If there are any questions or suggestions, please feel free to contact us.

**Response to RC1:**

We would like to sincerely thank you for thoroughly reviewing our manuscript. All your comments have been carefully addressed, and a point-by-point response is provided below.

**Overall comments:**

RC1-1. This work is a typical representation of the forward development in the current hydrological modeling field. By leveraging existing advanced numerical techniques, it reconstructs the traditional empirical, conceptual, and lumped hydrological model, transforming it into modern distributed hydrological model that can represent spatial heterogeneity and are based on complete systems of differential equations. Overall, I think this work is of great practical significance. In addition, the author's writing is also quite mature and concise. Nevertheless, I still believe that this work should undergo major revision before it is suitable for publication, as the presentation of the results section remains insufficient for a mature study.

Thank you for your positive remarks on our work, particularly for acknowledging and highlighting our efforts. Your comments are invaluable, and we have carefully considered each point to enhance the clarity and maturity of the revised manuscript. We hope that these changes could effectively address your concerns.

**Major concerns:**

RC1-2. I have reviewed the model code provided by the author and noticed that it is relatively concise pure Python code and does not include many additional dependencies. I believe this is a strong advantage, as it suggests broader usage scenarios and greater potential for expansion. However, as an interpreted language, Python has a typical limitation in comparison to compiled languages like C, namely its lower efficiency. As a potential application for large domain modeling, I am concerned about the computational efficiency of the TDD-XAJ model. The author should at least provide a reference for efficiency, and clarify whether transitioning from a 1D modeling framework to a 2D framework would result in a significant decrease in computational efficiency. I believe the author should add a section to address these issues.

In addition, please include a description of the programming environment of the model, as well as the dependencies used, to help users clearly understand the usage scenarios.

Thank you for taking the time and effort to review the code we provided. The model was implemented solely in Python, and to ensure readability, understandability, and ease of application and potential future

extensions, we used only essential libraries, keeping the code concise while maintaining completeness. We are grateful for your recognition of this manner and your positive feedback.

We understand your concern regarding the computational efficiency of the TDD-XAJ model. Just as you pointed out, Python is an interpreted language, does lag behind compiled languages such as C or Fortran in terms of execution speed. We have added a section (Section 4.4, Line 614-638 in tracked version) in the revised manuscript to address this issue. Besides, transitioning from a 1D modeling framework to a 2D framework does not result in a significant decrease in computational efficiency. To clarify this, we repeated the models used in the slope concentration methods comparison experiment and recorded the runtime. The details and analysis are also added to the Section 4.4 of the manuscript. The Section 4.4 is attached below:

Line 614-638 in tracked version:

**4.4 Computational efficiency of the model**

The implementation of the TDD-XAJ model is detailed in Appendix A. The Python language was chosen for its robust ecosystem in hydrology and earth system sciences (Stacke and Hagemann, 2021; Murphy et al., 2024), which facilitates rapid development and experimentation. While Python, as an interpreted language, is generally less computationally efficient than compiled languages, it benefits from libraries like NumPy, which rely on compiled languages for underlying computations. To enhance computational performance, we leveraged NumPy's vectorized operations and optimized functions, which are generally faster than native Python code.

To compare the computational efficiency of the 1D and 2D slope runoff concentration methods, we repeated the models used in corresponding comparison experiment for 10 times, and recorded the runtime for each. The average runtime values, obtained using an AMD 5950X CPU (3.4 GHz) in single-core mode, are summarized in Table 5.

**Table 5**. Average runtime recorded in the slope runoff concentration methods comparison numerical experiment.

| Surface comparison scenario (s) | | | Subsurface comparison scenario (s) | | |
|---|---|---|---|---|---|
| Method | Single-slope case | Double-slope case | Method | Single-slope case | Double-slope case |
| 1D diffusion wave | 28.41 | 28.32 | 1D linear reservoir | 2.10 | 2.06 |
| 2D diffusion wave | 29.02 | 29.80 | 2D linear reservoir | 1.13 | 1.12 |

As shown in Table 5, replacing 1D slope runoff concentration methods with 2D methods incurs minimal computational overhead while potentially enhancing efficiency. For diffusion wave method used for surface runoff concentration, the runtime for the 2D form increases 2.1% and 5.2% compared to the 1D form in the single-slope and double-slope case, respectively. Conversely, for linear reservoir method

applied to subsurface runoff concentration, switching from 1D to 2D reduces runtime by at least 45.6% across both test cases. The numerical computation procedure for the slope runoff concentration methods mainly involves two steps: flux calculation and state variable update. In flux calculation step, the 1D and 2D diffusion wave method require calculating once per cell (based on GIS-derived flow direction) or twice (in both x and y directions), respectively. For linear reservoir method, both the 1D and 2D forms require calculation only once per cell, and the outflow flux evaluated is further decomposed into x- and y- directional components based on slope aspect for the 2D form. When updating state variables, both the diffusion wave and linear reservoir methods in their 1D form require logical judgments based on GIS-derived flow direction to identify the inflow cells. In their 2D form, this can be directly determined using cell indexing. The element-wise logical judgment is relatively time-consuming and is comparable to a one-time flux calculation per cell. Therefore, for the diffusion wave method, transitioning from 1D to 2D form does not result in a significant increase in runtime, but for the linear reservoir method, the computational overhead is effectively reduced.

We have further discussed the limitations of using Python and potential directions for future improvements. This content has been added to the new section suggested by RC2 (Section 4.5, Line 639-657 in tracked version), which is shown below:

Line 647-652 in tracked version:
In terms of computational efficiency, although we make extensive use of the NumPy library, which is generally faster than native Python code, we recognize that NumPy may still be slower in certain operations (e.g., element-wise calculations) compared to compiled languages. This is a well-known limitation within the Python community, and solutions like Just-In-Time (JIT) compilation have been proposed (Lam et al., 2015), which convert frequently executed script code into machine code with further automatic optimizations. Although this manuscript primarily focuses on presenting the theoretical aspects of the TDD-XAJ model, we plan to optimize the code, including the implementation of parallelization, in future work.

As suggested, we have added an appendix (Appendix A, Line 687-693 in tracked version) to the manuscript, providing details of the programming environment and library dependencies used in the model. The appendix is provided below:

Line 687-693 in tracked version:
**Appendix A: Implementation details of the TDD-XAJ model**
The TDD-XAJ model is implemented in Python 3.9, taking advantage of its extensive ecosystem as well as cross-platform capabilities. To maintain readability and facilitate understanding, only essential libraries

are used, ensuring the model is both accessible and easily extendable in the future. A list of these libraries is provided in Table A1. NumPy plays a crucial role, providing vectorized operations and optimized computational functions that enhance performance compared to native Python code. Other libraries serve supporting roles, such as model configuration and handling file input/output.

**Table A1.** Libraries utilized in the implementation of the TDD-XAJ Model.

| Category | Name | Version | Description |
|---|---|---|---|
| Core | NumPy | 1.26.4 | A basic scientific computing library for efficient numerical operations |
| Auxiliary | PyYAML | 6.0.1 | Library for reading YAML files used for model configuration and settings |
| | Pandas | 2.2.1 | A high-level library for data manipulation and analysis |
| | OpenPyxl | 3.1.5 | Backend support for reading and writing Excel files in Pandas |
| | treelib | 1.7.0 | Efficient implementation of tree data structure |

RC1-3. I have some confusion regarding the channel unit modeling in Figure 4. In the previous sections (Section 2.1 and Figure 1), the channel units are explicitly defined as segments, with their lengths clearly specified in three ways. In these cases, the width of the channel units is significantly smaller than the grid size so that the rainfall can be neglected. However, in this test case (Figure. 4), the grid size (5m x 5m) is significantly larger than the channel width (20m). Does this mean that the channel units need to encompass multiple grids? I would ask the author to provide a corresponding diagram to clarify how spatial discretization and channel unit modeling are handled in the test cases.

A similar issue also applies to real basin modeling. Could you provide a diagram to illustrate how spatial discretization is carried out in the Tunxi watershed, including the definition of grid units and channel units?

Thanks for your comments. The TDD-XAJ model uses DEM grid cells as its fundamental computational units, which means the simulation domain is spatially discretized with square structure grid (see Figure R1 below). These cells are explicitly categorized into two types: (1) channel cells, which containing river channels, where both slope concentration and channel concentration processes are considered, including water exchange between the slope and the channel, and (2) non-channel cells, which do not contain river channels and only slope concentration process is considered. In channel cells, the channel concentration process is governed by one-dimensional diffusion wave equations. This allows the channel to be conceptualized as a line feature within the model, with its width treated as a cross-sectional attribute of the one-dimensional equations. The spatial relationship between the channel line feature and channel cells adopts two topologies: intersecting (channel passes through cell interiors) and adjacent (channel aligns with cell boundaries). In real-world watershed applications, the channel width is generally smaller than the grid size. Thus, we implement the intersecting topology by positioning the channel line through cell centers (Figure R1a). In the synthetic V-shaped watershed test case, where the channel width (20m) is

significantly larger than the grid resolution (5m), we adopt the adjacent topology to keep physical realism and avoid representing the channel with multiple cells (Figure R1b).

[Figure]

**Figure R1.** Diagram of spatial discretization and the arrangement of channel line features (with two spatial topological relationships between the channels and slopes).

Figure 5 in our manuscript initially presented the location of the Tunxi watershed and its gauging stations. To illustrate the spatial discretization in the Tunxi watershed, including the definition of grid (non-channel) units and channel units, we have made further extensions based on this figure (see Figure 5b). The revised Figure 5 is shown below:

Line 432-435 in tracked version:

[Figure]

Figure 5. Location and gauging station distribution of the Tunxi watershed (a), and the spatial discretization of the watershed, including channel and non-channel cells (b).

RC1-4. Since the author emphasizes the advantages of the 2D method in capturing microtopography and presents it as a typical case of distributed modeling, the author should at least provide some results for

spatial simulations. This would better illustrate the model's advantages and provide supporting evidence for the attribution.

Specifically, the author could provide some degree of spatial validation (for example), which is highly valued in the hydrological modeling community. If this proves challenging (due to data limitations or other reasons), an alternative could be to include a test case with more varied slopes (such as two or three different slope changes in the y-direction). In this more complex test case, testing the consistency between slope variations and simulation results would better highlight the advantages of 2D modeling.

Thanks for your valuable comments. To address this, we have further investigated three aspects: (a) the introduction of a hydrological station within the watershed and the evaluation of the simulation performance of the TDD-XAJ model at this station, (b) the inclusion of the spatial simulation results in addition to the simulated hydrograph, and (c) further validation by varying the y-directional slope of the test case, as per you suggestion.

Firstly, we introduced the Yuetan hydrological station—a station within the Tunxi watershed (Figure 5), and compared its simulation results with observed data. Details of performance metrics are provided in Table 4. As shown in Table 4, the average values of the Nash-Sutcliffe efficiency (NSE), Kling-Gupta efficiency (KGE), the absolute flood volume relative error (|FVRE|), and the coefficient of determination (R2) for Yuetan station (across all years) are 0.83, 0.78, 6.2%, and 0.86, respectively. The corresponding values for Tunxi station are 0.87, 0.80, 6.7%, and 0.90. In summary, these metrics indicate that the TDD-XAJ model provides robust streamflow simulations at both stations in the Tunxi watershed. The revisions are detailed below:

Line 432-435 in tracked version:

[revised manuscript text omitted]

The application results indicate a good and consistent agreement between the observation and the simulations of the TDD-XAJ model in the Tunxi watershed, demonstrating a slight improvement in performance metrics compared to the GXAJ model, particularly for flood volume. Additionally, the spatial analysis of simulation results demonstrates that the TDD-XAJ model captures spatial variability in water storage effectively in response to varying precipitation patterns.

Due to data limitations, we regret that we are unable to provide further spatial validation. As an alternative, and based on your suggestion, we have expanded the range of slope in the $y$-direction ($S_{oy}$). The original used $S_{oy}$ of 0.00 (single-slope case) and 0.02 (double-slope case) have been extended to 0.00, 0.01, 0.02, 0.03, 0.04, and 0.05, while the slope in the $x$-direction ($S_{ox}$) is fixed to 0.05. For each case, both 1D and 2D runoff concentration methods are applied. The 1D methods use GIS-derived flow directions, and the comparison between the calculated and real flow directions is shown in Figure R2, and they are identical only when $S_{oy}$ equals 0.00 and 0.05.

[Figure]

| Flow direction | $S_{oy}$ | | | | | |
|---|---|---|---|---|---|---|
| | 0.00 | 0.01 | 0.02 | 0.03 | 0.04 | 0.05 |
| Real | 0° | 11.31° | 21.80° | 30.96° | 38.66° | 45° |
| GIS-derived | 0° | 0° | 0° | 45° | 45° | 45° |

Figure R2. The comparison of real and GIS-derived flow direction under different $y$-directional slope ($S_{oy}$).

For surface runoff concentration, the 1D and 2D forms of diffusion wave (DW) method are used. Figure R3 and R4 illustrate the comparison of hillslope and channel outflow hydrograph for different $S_{oy}$, respectively. In the case of subsurface runoff concentration, the 1D and 2D forms of linear reservoir method (LR) are implemented. The hillslope outflow hydrograph is demonstrated in Figure R5, with channel outflow hydrograph further detailed in Figure R6. We used mean absolute error (MAE) to assess the difference between hydrographs simulated by the 1D and 2D forms of runoff concentration methods. From these figures, we can see that when the GIS-derived flow direction aligns with the actual flow direction ($S_{oy}$=0.00, and $S_{oy}$=0.05), the results of 1D form methods match those of the 2D form methods (see Figure R3a and R3f, Figure R4a and R4f, Figure R5a and R5f, Figure R6a and R6f). However, when there are discrepancies in derived and real flow directions, certain components in 1D form methods are neglected and its results show deviations.

[Figure]

**Figure R3.** The hillslope outflow hydrograph of synthetic V-catchment test case with varying $y$-directional slope ($S_{oy}$) in surface comparison scenario. The one-dimensional (1D) and two-dimensional

(2D) form diffusion wave methods (DW) were evaluated in this scenario. The difference between the outflow of 1D and 2D form methods was assessed with mean absolute error (MAE).

[Figure]

**Figure R4.** The channel outflow hydrograph of synthetic V-catchment test case with varying $y$-directional slope ($S_{oy}$) in surface comparison scenario. The one-dimensional (1D) and two-dimensional (2D) form diffusion wave methods (DW) were evaluated in this scenario. The difference between the outflow of 1D and 2D form methods was assessed with mean absolute error (MAE).

[Figure]

**Figure R5.** The hillslope outflow hydrograph of synthetic V-catchment test case with varying $y$-directional slope ($S_{oy}$) in subsurface comparison scenario. The one-dimensional (1D) and two-dimensional (2D) form linear reservoir methods (LR) were evaluated in this scenario. The difference between the outflow of 1D and 2D form methods was assessed with mean absolute error (MAE).

[Figure]

**Figure R6.** The channel outflow hydrograph of synthetic V-catchment test case with varying $y$-directional slope ($S_{oy}$) in subsurface comparison scenario. The one-dimensional (1D) and two-dimensional (2D) form linear reservoir methods (LR) were evaluated in this scenario. The difference between the outflow of 1D and 2D form methods was assessed with mean absolute error (MAE).

RC1-5. The differences between Figure. 7b and Figure. 7c suggest that 1D modeling overlooks the interflow component in the y-direction, although this is not particularly clear. Could you include one or two time profile plots corresponding to different stages in Figure. 6g to further support this conclusion?

Thanks for your valuable feedback. In Figure 7, we illustrate the differences between the 1D and 2D forms of the linear reservoir method for subsurface runoff concentration through Figure 7b and Figure 7d. As you pointed out, the contrast between these two sub-figures is not very clear. To address this, we have plotted the distribution of interflow storage completely and then zoomed in on the areas where differences are more noticeable (as suggested by CC1). This visualization approach helps to highlight how the 1D model overlooks the interflow component in the $y$-direction. The updated Figure 7 is shown below:

Line 468-474 in tracked version:

[Figure]

**Figure 7.** The spatial distribution of surface water depth ($h_s$) and interflow storage ($O_i$) on the left-side hillslope of both single-slope (a-b) and double-slope (c-d) synthetic V-catchment test cases at the 60 minute mark. The state variable distributions shown are simulated using two-dimensional (2D) slope concentration methods. The corresponding results of 1D methods are identical to those obtained from the single-slope case simulated with 2D methods, regardless of the test case used. For a clear comparison, the spatial distribution of $O_i$ in the upper left corner has been zoomed.

Furthermore, we selected three points (A, B, and C, see Figure R7) from the zoomed area in the updated Figure 7. For both single-slope and double-slope cases, we generated time profile plots of the interflow storage using both the 1D and 2D linear reservoir method at these points (Figure R8). The comparative results clearly reveal the simulated differences between the 1D and 2D linear reservoir methods in the double-slope test case (Figures R8d to R8f).

[Figure]

**Figure R7**. Diagram of the spatial location for points A, B, and C.

[Figure]

**Figure R8**. Timeseries of interflow storage ($O_i$) simulated by 1D and 2D linear reservoir (LR) methods for single-slope (a-c) and double-slope (d-f) synthetic V-catchment test cases (at points shown in Figure R7).

Below are some of my suggestions regarding the writing or presentation (although the paper is already well-written):

**Specific issues:**

RC1-6. Line 13-14: Please emphasize both the 2D diffusion wave and the 2D linear reservoir method in this sentence to enhance consistency with the paper.

Thanks for your suggestion. We have revised the sentence into:

Line 13-15 in tracked version:

We also introduced two-dimensional (2D) diffusion wave equations for surface slope concentration, and derived 2D linear reservoir equations for subsurface slope concentration, to replace their 1D counterparts.

RC1-7. Line 58-59: Beven's alternative blueprint encompasses several key concepts, including Bayesian philosophy, model equivalence, and an emphasis on uniqueness. It would be beneficial to clarify which of these are closely related to the development of XAJ, although this may be considered a matter of fine-tuning.

Thanks for your suggestion. Beven's alternative blueprint indeed encompasses several key perspectives, including those you mentioned, as well as an evaluation on the physical-based property of distributed hydrological model (Beven, 2002). Before the alternative blueprint, physical-based model generally emphasized on applying rigorous mathematical-physical equations deriving from established physical principles under appropriate assumptions, such as different simplified form of the Navier-Stokes equations. However, the alternative blueprint, while still adhering to fundamental physical principles and conservation laws, places greater emphasis on consistency with observational data. This shift effectively relaxes the strict dependence on mathematical-physical equations, enabling the integration of conceptual parameterization equations from the lumped hydrological model, thereby expanding the available equation space for distributed hydrological model. This laid the foundation for the development of the distributed version of the XAJ model, making it closely linked to the evolution of the XAJ model. According to your suggestion, we have revised it to:

Line 61-64 in tracked version:

…, 2002 was a notable milestone due to Beven's alternative blueprint (Beven, 2002). This blueprint emphasized the importance of observational consistency as a core requirement of the physical-based property in DHMs, enabling the integration of parameterized equations from lumped hydrological models.

RC1-8. Line 342: Please clarify the model's inputs and outputs, and whether they differ from the original XAJ model.

As suggested, we have included an appendix (Appendix B, Line 694-704 in tracked version) in the manuscript to clarify the inputs and outputs of the TDD-XAJ model, along with an analysis of whether they differ from the original XAJ model. The appendix is provided below:

Line 694-704 in tracked version:

**Appendix B: Inputs and outputs of the TDD-XAJ model**

The inputs and outputs of the TDD-XAJ model are listed in Table B1. Its inputs consist of four categories: (1) model configuration dictionary, (2) spatiotemporal multidimensional arrays of precipitation and evaporation as forcing inputs, (3) raster datasets providing spatial attributes (including elevation, river network, aspect, etc.), and (4) lookup tables storing hydraulic parameters for surface runoff and channel concentration. The outputs comprise spatiotemporal distributions of state variables and model fluxes at pre-defined temporal resolutions (instantaneous or time-averaged values). Compared to the original XAJ model, which is a lumped hydrological model, its inputs are watershed-scale parameters and timeseries for precipitation and evaporation, with output mainly including simulated discharge hydrograph. The TDD-XAJ model shares largely similar input and output information with existing distributed XAJ model,

the aspect data introduced by the 2D linear reservoir method can be derived from elevation raster.

**Table B1.** Detailed information on input and output files of the TDD-XAJ model.

| Class | Category | Name | Format | Description |
|---|---|---|---|---|
| Input | Dictionary | Model configuration | YAML | (1) Paths for the other files in this table, temp and output folders
(2) Numerical calculation settings (e.g., CFL coefficient)
(3) Model parameters
(4) Location of hydrological stations |
| | Array | Forcings | NPY | Spatiotemporal distribution of precipitation and evaporation |
| | Raster | Elevation | Esri ASCII | Use projected coordinate system, with unit of m |
| | Raster | Aspect | Esri ASCII | Derived from elevation raster via GIS analysis |
| | Raster | Flow accumulation | Esri ASCII | Derived from elevation raster via GIS analysis |
| | Raster | Flow direction | Esri ASCII | Derived from elevation raster via GIS analysis |
| | Raster | River network | Esri ASCII | Mark the grid where channel is located and corresponding code |
| | Raster | Land use | Esri ASCII | Aligned with elevation raster, and store land use code |
| | Lookup table | Surface hydraulics | CSV/XLSX | Land use code and corresponding surface roughness coefficient |
| | Lookup table | Channel hydraulics | CSV/XLSX | Channel code and corresponding attributes (e.g., longitudinal slope and channel roughness coefficient) |
| Output | Array | State variables | NPY | Spatiotemporal distributions of state variables at pre-defined temporal resolutions (instantaneous or time-averaged values) |
| | Array | Model fluxes | NPY | Spatiotemporal distributions of model fluxes at pre-defined temporal resolutions (instantaneous or time-averaged values) |

RC1-9. Line 359: "The synthetic V-catchment, first proposed by (Overton and Brakensiek, 1970)", This appears to be an incorrect citation format.

Thanks for pointing this out, we have corrected this (Line 370 in tracked version).

RC1-10. Figure 6: The lower-left corner should be "subsurface" rather than "surface."

Thank you for pointing this out, we have corrected this (Line 461-462 in tracked version).

RC1-11. Figure 8: Highlight the inflection.

Thanks for your suggestion. We have highlighted the inflection point in Figure 8 (a) using a red dashed box and plotted the first-order numerical derivative to emphasize this phenomenon. The updated Figure 8 is shown below:

Line 536-543 in tracked version:

[Figure]

**Figure 8.** The simulated hillslope outflow (a) and channel outflow (b) of the loosely-coupled (LC) model and fully-coupled (FC) model in the double-slope case. The time in parentheses denotes the time interval of input forces ($\Delta T$) used for the model. The FC model uses 90 min only, while the LC model uses 90 min, 45 min, and 15 min for convergence test. The parameters used are the first set of the parameters sampled with the SLH method from their ranges listed in Table 2. The inflection point at 75 minute mark in the hillslope outflow hydrograph of the LC model (using $\Delta T$=15min) is highlighted with a red dashed box, along with timeseries of the first-order numerical derivative of hillslope outflow ($\Delta Q$).

**Response to RC2:**

We deeply appreciate your detailed review on our manuscript. All your comments have been carefully addressed, and a point-by-point response is provided below.

**Overall comments:**

RC2-1. This manuscript not only summarizes the progress achieved during past decades but also advances the field by establishing a complete system of differential equations for the distributed XAJ model, along with its numerical implementation. The manuscript provides a thorough analysis of both one-dimensional and two-dimensional concentration methods while addressing numerical error issues. The manuscript is well-written with promising application prospects. To strengthen the manuscript, I would like to propose several comments in the order of line numbers.

Thank you very much for your concise paper summary and positive feedback on our research. We have carefully considered all of your comments and addressed them in the subsequent specific comments section. We hope that these changes could effectively address your concerns and enhance our manuscript.

**Specific comments:**

RC2-2. Line 55: It is more appropriate to describe this as "the widely-used three-water-sources lumped XAJ model."

Thanks for your suggestion. Including the term "three-water-sources" makes the sentence more precise. We have revised the sentence into:

Line 56-57 in tracked version:
…, leading to the formation of the widely-used three-water-sources lumped XAJ model (Zhao, 1992).

RC2-3. Line 56-66: While I agree the third phase highlighted the development of the distributed XAJ model, it is important to note that the original lumped XAJ model has continued to evolve in parallel. The authors should briefly acknowledge this progression.

Thanks for pointing this out. We agree that it is important to acknowledge that the original lumped XAJ model has continued to evolve in parallel with the development of the distributed XAJ model. According to your suggestion, we have revised it to:

Line 58-61 in tracked version:

**Phase 3 (2002-present):** The third phase is characterized by efforts to transition the XAJ model from a lumped to a distributed version, aligning it with other contemporary models like TOPMODEL and HBV (Beven et al., 2021; Seibert et al., 2022). It is noteworthy that the original lumped XAJ model has undergone continuous evolution alongside (Ouyang et al., 2025).

RC2-4. Line 61: I believe the term "concentration method" is more appropriate here to maintain terminology consistency throughout the manuscript.

Thank you for your feedback regarding terminology consistency. We agree that aligning terminology strengthens clarity. In addition to Line 61, we have checked the terminology throughout the manuscript. The principle is to replace "route" with "runoff concentration" when used in conjunction with "runoff generation". According to your suggestion, we have revised it to:

Line 64-67 in tracked version:
The transformation involved the application of the runoff generation modules to smaller computational units (e.g., sub-basins or grids) and implementing distributed hydrological or hydraulic runoff concentration methods (Chen et al., 2024; Fang et al., 2017; Liu et al., 2009; Su et al., 2003).

RC2-5. Line 126: "Deeper" in Figure 1 should be corrected into "Deep", as referenced in Section 2.2.1. Besides, the slope-channel coupling diagram in this figure should be described in the text for clarity.

Thanks for pointing this out. As you suggested, we have revised it to:

Line 129-135 in tracked version:
Bidirectional water exchange between slope surface and channel is considered under varying hydraulic conditions. Specifically, when the channel water surface elevation exceeds that of the slope surface, water flows from the channel back onto the slope surface. The diagram of the TDD-XAJ is shown in Fig. 1.

[Figure]

**Figure 1.** Diagram of the two-dimensional differential-form of distributed Xinanjiang model.

RC2-6. Line 145: The term "slope" is recommended instead of "hillslope".

Thanks for your suggestion. We have revised it accordingly (Line 152-153 in tracked version).

RC2-7. Line 153: Given the number of equations and variables, it is suggested to include a nomenclature section.

We agree that including a nomenclature section would enhance readability, and we have added it in the supplementary material (Table S1, Line 72-73 in the Supplement). Thanks for your suggestion.

RC2-8. Line 343: The determination methods of 15 model parameters listed in the Table 1 requires detailed explanation. In addition, it is also necessary to explain the differences in model parameter from the original XAJ model.

Thanks for your comments. The TDD-XAJ model comprises 15 parameters, which is one parameter more than the original lumped Xinanjiang model (Zhao et al., 2023), with 11 hydrological parameters for runoff generation and 4 parameters (2 hydrological and 2 hydraulic) for runoff concentration. These 13 hydrological parameters are consistent with those from the original lumped model. The remaining hydrological parameter of the original lumped model is used to represent the channel concentration process. The channel concentration process is represented with one-dimensional diffusional wave equations in the TDD-XAJ model, thus transforming the hydrological parameter into a hydraulic parameter (channel roughness coefficient, $n_c$). The additional hydraulic parameter (surface roughness

coefficient, $n_s$) in the TDD-XAJ model is included to facilitate two-dimensional diffusional wave equations for slope surface concentration.

We have added a column to the model parameter table (Table 1), to distinguish whether the parameters are spatially uniform or spatially distributed. For the spatially distributed parameters, we introduced their determination methods in the supplementary material (see Section S2). As suggested, we have revised it to:

Line 351-356 in tracked version:

The TDD-XAJ model has 15 tunable parameters, which is one more than in the original lumped Xinanjiang model (Zhao et al., 2023). The signification and value range of these parameters are listed in Table 1. Parameters that can be spatially uniform are consistent with those from the original lumped model. The methods for determining spatially distributed parameters are provided in Sect. S2 of the Supplement.

**Table 1.** Parameters and parameter range of the TDD-XAJ model.

| Components | Symbol | Signification | Range | Spatially | Unit |
|---|---|---|---|---|---|
| Evapotranspiration and runoff calculation | $K_e$ | Coefficient of potential evapotranspiration to pan evaporation | [0.6, 1.5] | Uniform | - |
| | $c$ | Coefficient of deep soil layer evapotranspiration | [0.01, 0.2] | Uniform/ Distributed | - |
| | $W_{um}$ | Tension water storage capacity of upper soil layer | [5, 30] | Uniform/ Distributed | mm |
| | $W_{lm}$ | Tension water storage capacity of lower soil layer | [60, 90] | Uniform/ Distributed | mm |
| | $W_{dm}$ | Tension water storage capacity of deep soil layer | [15, 60] | Uniform/ Distributed | mm |
| | $A_{imp}$ | The ratio of the impervious area | [0.01, 0.2] | Uniform/ Distributed | - |
| | $b$ | Tension water storage capacity curve exponent | [0.1, 0.4] | Uniform/ Distributed | - |
| Runoff separation | $S_m$ | Free water storage capacity | [10, 50] | Uniform/ Distributed | mm |
| | $ex$ | Free water storage capacity curve exponent | [1.0, 1.5] | Uniform/ Distributed | - |
| | $K_i$ | Interflow outflow coefficient | [0.10, 0.55] | Uniform/ Distributed | - |
| | $K_g$ | Groundwater outflow coefficient | $0.7 - K_i$ | Uniform/ Distributed | - |
| Slope concentration | $n_s$ | Surface roughness coefficient | [0.01, 0.80] | Distributed | m s$^{-1/3}$ |
| | $C_i$ | Interflow storage recession coefficient | [0.5, 0.9] | Uniform/ Distributed | - |
| | $C_g$ | Groundwater storage recession coefficient | [0.98, 0.998] | Uniform/ Distributed | - |
| Channel concentration | $n_c$ | Channel roughness coefficient | [0.01, 0.05] | Distributed | m s$^{-1/3}$ |

Line 16-71 in the Supplement:

**Section S2: The determination methods of spatially distributed model parameter**

To determine spatially distributed model parameters, the process is generally based on spatially quantified data of watershed physical characteristics. This work is primarily carried out in two ways:

(1) **Lookup table-based method**. Parameters are determined from tables based on watershed physical attributes. Specifically, the ratio of the impervious area ($A_{imp}$) and coefficient of deep soil layer evapotranspiration ($c$) are determined according to land use types (Yao et al., 2012), while the determination of tension water storage capacity curve exponent ($b$) and free water storage capacity

curve exponent ($ex$) are assigned based on soil types. The value of surface roughness coefficient ($n_s$) is assigned based on the land use type of each grid cell, with different land uses corresponding to different roughness coefficients, which are derived from existing literature (Miao et al., 2016; Perrini et al., 2024). For channel roughness coefficient ($n_c$), values are obtained from a roughness coefficient table for river channels (Arcement and Schneider, 1989).

(2) **Physical meaning-based method**. Parameter values are calculated using quantitative watershed physical characteristics according to the physical meaning of the parameters. Specifically:

**a**. Tension water storage capacity of the upper, lower, and deep soil layer ($W_{um}$, $W_{lm}$, and $W_{dm}$). The summation of $W_{um}$, $W_{lm}$, and $W_{dm}$ represents the tension water capacity of the entire soil layer ($W_m$), and it can be determined according to soil hydrological parameters and soil layer depth (Yao et al., 2012), which could be expressed as:

$$W_m = (\theta_f - \theta_r)D_s, \tag{S4}$$

where $\theta_f$ is field capacity, $\theta_r$ is residual water content, $D_s$ is soil layer depth (mm). Subsequently, two watershed-scale uniform coefficients ($K_{um}$ and $K_{lm}$) and their derived value ($1 - K_{um} - K_{lm}$) are used to divide $W_m$ into $W_{um}$, $W_{lm}$, and $W_{dm}$ accordingly, which are given as:

$$W_{um} = W_m K_{um}, \tag{S5}$$

$$W_{lm} = W_m K_{lm}, \tag{S6}$$

$$W_{dm} = W_m(1 - K_{um} - K_{lm}). \tag{S7}$$

**b**. Free water storage capacity ($S_m$). $S_m$ usually represents the capacity of free water in the humus layer. Thus, it can be determined according to soil hydrological parameters and the humus layer depth (Yao et al., 2012), which could be expressed as:

$$S_m = (\theta_s - \theta_f)D_h, \tag{S8}$$

where $\theta_s$ is saturated water content, $\theta_f$ is field capacity, $D_h$ is humus layer depth (mm).

**c**. Interflow and groundwater outflow coefficient ($K_i$ and $K_g$). $K_i$ and $K_g$ represent the outflow rate of interflow and groundwater. The method for determining $K_i$ and $K_g$ involves converting the free water storage to corresponding saturated water depth, based on the hillslope storage-discharge theory and steady-state assumptions, which is then multiplied by the slope gradient and saturated hydraulic conductivity using the kinematic wave assumption (Tong, 2022). $K_i$ and $K_g$ are finally expressed as the ratios of corresponding flow distance in the time interval of input forces to the slope length, which could be given as:

$$K_i = \frac{2S_0 K_{su} S_{hill} \Delta T}{1000(\theta_s - \theta_f)L_{hill}^2}, \tag{S9}$$

$$K_g = \frac{2S_0 K_{sl} S_{hill} \Delta T}{1000(\theta_s - \theta_f)L_{hill}^2}, \tag{S10}$$

where $S_0$ is free water storage (mm), $K_{su}$ and $K_{sl}$ is saturated hydraulic conductivity of the upper (representing interflow) and lower (representing groundwater) soil layer respectively (m s$^{-1}$), $S_{hill}$ is

the gradient of the slope, $\Delta T$ is the time interval of input forces (s), and $L_{hill}$ is the length of the slope (m).

**d**. Interflow and groundwater storage recession coefficient ($C_i$ and $C_g$). $C_i$ and $C_g$ represent the time delay for interflow and groundwater runoff as they travel from specific locations on the slope to the river channel. These parameters are determined based on the theory of spatially distributed unit hydrograph (Maidment et al., 1996; Tong, 2022). The grid cells that form the flow path extending from specific locations on the slope to the river channel is first identified using GIS. Then, using the kinematic wave assumption, the flow velocity of interflow and groundwater runoff through each grid cell is computed based on the saturated hydraulic conductivity of the upper and lower layers and the slope gradient. Finally, the time taken for flow through each grid cell is accumulated, which could be expressed as:

$$T_i = \sum_{j=1}^{N_{hill}} L_{hill}^j / (K_{su}^j S_{hill}^j), \tag{S11}$$

$$T_g = \sum_{j=1}^{N_{hill}} L_{hill}^j / (K_{sl}^j S_{hill}^j), \tag{S12}$$

where $T_i$ and $T_g$ is the accumulated travel time from specific locations on the slope to the river channel through interflow and groundwater respectively (s), $N_{hill}$ is the count of grid cells that form the flow path. $C_i$ and $C_g$ for each grid cell are further derived using theoretical conversion, which could be given as:

$$C_i = \exp(-\Delta T / T_i), \tag{S13}$$

$$C_g = \exp(-\Delta T / T_g). \tag{S14}$$

The primary data used to determine spatially distributed model parameters include soil physical and hydraulic properties, slope gradient, and land use. These can be obtained from open-source datasets, such as Harmonized World Soil Database v2.0 (HWSD v2.0) (FAO and IIASA, 2023), China dataset of soil properties for land surface modelling version 2 (CSDLv2) (Shi et al., 2025), and Global land cover mapping at 30m resolution (GlobeLand30) (Chen et al., 2015).

RC2-9. Line 439: The equation of FVRE should be provided. Similarly, it is also suggested to provide the formulas of three channel cross-sectional hydraulic elements mentioned in Line 282.

Thanks for your comment. As suggested, the equation of FVRE is provided below:

Line 453-455 in tracked version:
The equation of FVRE could be given as:

$$FVRE = \left( \sum_{i=1}^{n} Q_{sim,i} - \sum_{i=1}^{n} Q_{obs,i} \right) / \sum_{i=1}^{n} Q_{obs,i} * 100\% \tag{32}$$

where $Q_{sim,i}$ and $Q_{obs,i}$ represent simulated and observed discharge at time step i (m$^3$ s$^{-1}$), and $n$ is the length of the sequence.

Furthermore, we have provided the formulas of three channel cross-sectional hydraulic elements (cross-sectional area, water surface width, and wetted perimeter) in supplementary material. The revisions are detailed below:

Line 289-291 in tracked version:
The cross-section is generalized into a trapezoid, and the formulas of cross-sectional hydraulic elements including $A$, $B$, and $\chi$ can be derived accordingly (see Eq. S1, S2, and S3 in the Supplement).

Line 5-15 in the Supplement:
**Section S1: Cross-sectional generalization and hydraulic parameters of river channel**
In the two-dimensional differential-form of distributed Xinanjiang model (TDD-XAJ), the cross-section of river channel is generalized into a trapezoid (Fig. S1). Thus, the formulas for cross-sectional area, water surface width, and channel wetted perimeter could be given as:

$$A = \begin{cases} \varsigma h_c + h_c^2/\tan\beta & 0<\beta<90° \\ \varsigma h_c & \beta = 90° \end{cases}, \tag{S1}$$

$$B = \begin{cases} \varsigma + 2h_c/\tan\beta & 0<\beta<90° \\ \varsigma & \beta = 90° \end{cases}, \tag{S2}$$

$$\chi = \begin{cases} \varsigma + 2h_c/\sin\beta & 0<\beta<90° \\ \varsigma + 2h_c & \beta = 90° \end{cases}, \tag{S3}$$

where $A$ is cross-sectional area (m²), $B$ is water surface width (m), $\chi$ is channel wetted perimeter (m), $\varsigma$ is channel bottom width (m), $h_c$ is channel water depth (m), $\beta$ is river bank slope gradient (°).

[Figure]

**Figure S1.** Diagram of trapezoidal cross-sectional generalization of river channel.

RC2-10. Line 452: In the slope concentration methods comparison experiment, the authors systematically compared the 1D and 2D forms of the diffusion wave and linear reservoir methods based on idealized test cases. For the diffusion wave method, significant differences were observed between the 1D and 2D form, both in terms of hydrographs and surface storage. However, for the linear reservoir method, while the differences in hydrographs were noticeable (Figure 6g and 6h), the contrast in storage was less evident (Figure 7b and 7d). The authors should improve the visualization approach for Figure 7 such as by changing color schemes to make the comparison more clear.

We are sorry for not clearly presenting the differences between the spatial distributions of interflow storage simulated by the 1D and 2D linear reservoir methods, as also noted by other reviewers. Following the suggestion of CC1, we have provided the complete spatial distribution of interflow storage and then zoom in on areas with significant differences. This approach allows for a clearer illustration while preserving data integrity. The revised Figure 7 is shown below:

Line 468-474 in tracked version:

[Figure]

**Figure 7.** The spatial distribution of surface water depth ($h_s$) and interflow storage ($O_i$) on the left-side hillslope of both single-slope (a-b) and double-slope (c-d) synthetic V-catchment test cases at the 60 minute mark. The state variable distributions shown are simulated using two-dimensional (2D) slope concentration methods. The corresponding results of 1D methods are identical to those obtained from the single-slope case simulated with 2D methods, regardless of the test case used. For a clear comparison, the spatial distribution of $O_i$ in the upper left corner has been zoomed.

RC2-11. Line 547-565: The authors analyzed the model's performance by applying it to the Tunxi watershed and examining the flow hydrograph at the outlet station, and the overall simulation results were satisfactory. However, as a distributed hydrological model, the authors should provide more details regarding the spatial simulation. Furthermore, it is suggested that the authors could include comparative results from stations within the watershed, if possible, as this would provide a more comprehensive evaluation of the performance of the TDD-XAJ model.

Thanks for your feedback on enhancing the spatial evaluation of the TDD-XAJ model. To address this, we have further included the spatial simulation results in addition to the simulated hydrograph in 2008

(Figure 10). Moreover, to strengthen the assessment of model performance, we introduced the Yuetan hydrological station—a station within the Tunxi watershed (Figure 5), and compared its simulation results with observed data. Details of performance metrics are provided in Table 4. As shown in Table 4, the average values of the Nash-Sutcliffe efficiency (NSE), Kling-Gupta efficiency (KGE), the absolute flood volume relative error (|FVRE|), and the coefficient of determination ($R^2$) for Yuetan station (across all years) are 0.83, 0.78, 6.2%, and 0.86, respectively. The corresponding values for Tunxi station are 0.87, 0.80, 6.7%, and 0.90. In summary, these metrics indicate that the TDD-XAJ model provides robust streamflow simulations at both stations in the Tunxi watershed. The revisions are shown below:

Line 432-435 in tracked version:

[revised manuscript text omitted]

The application results indicate a good and consistent agreement between the observation and the simulations of the TDD-XAJ model in the Tunxi watershed, demonstrating a slight improvement in performance metrics compared to the GXAJ model, particularly for flood volume. Additionally, the spatial analysis of simulation results demonstrates that the TDD-XAJ model captures spatial variability in water storage effectively in response to varying precipitation patterns.

RC2-12. Line 591-595: In the final paragraph of the conclusion, the authors summarize the limitations of this study. The manuscript encompasses extensive research efforts. I understand that the journey from proposing a model to its refinement and maturity is a lengthy process, and this manuscript has done an excellent job methodologically, providing a solid foundation for future application and research. I recommend relocating this paragraph to the discussion part, where the potential application scenarios and future research directions of the model can be further explored.

Thanks for your encouragement. As suggested, we have added a section (Section 4.5, Line 639-657 in tracked version) in Result and Discussion to discuss limitations and future research in the revised manuscript, which is shown below:

Line 639-657 in tracked version:

**4.5 Limitations and future research**

The main limitation of the TDD-XAJ model is that it addresses only the numerical errors on the time scale from the ODE's numerical solution, while neglecting errors arising from the spatial discretization of the PDE via the FVM method. Future research should consider using methods like manufactured solutions (Bisht and Riley, 2019) as an alternative for evaluating numerical errors arising from both spatial and temporal discretization when exact solutions are difficult to obtain. Additionally, exploring more advanced spatial discretization techniques, such as second-order FVM or discontinuous Galerkin method (Shaw et al., 2021), along with more sophisticated temporal integration methods, could help control numerical errors more effectively.

In terms of computational efficiency, although we make extensive use of the NumPy library, which is generally faster than native Python code, we recognize that NumPy may still be slower in certain operations (e.g., element-wise calculations) compared to compiled languages. This is a well-known limitation within the Python community, and solutions like Just-In-Time (JIT) compilation have been proposed (Lam et al., 2015), which convert frequently executed script code into machine code with further automatic optimizations. Although this manuscript primarily focuses on presenting the theoretical aspects of the TDD-XAJ model, we plan to optimize the code, including the implementation of parallelization, in future work.

It is essential to validate the model with more real-world watershed data and evaluate its uncertainty, particularly for watersheds with varying scales, diverse underlying physical conditions, and hydrological data at different temporal resolutions. The differential-form mathematical equations established for the TDD-XAJ model provide a solid foundation for future research, including combining deep learning for better model parameterization and process understanding (Höge et al., 2022; Li et al., 2024).

**Response to CC1:**

Thank you very much for your interest and valuable suggestions regarding our manuscript. All your comments have been carefully addressed, and a point-by-point response is provided below.

**Overall comments:**

CC1-1. In the study, a two-dimensional differential-form of distributed Xinanjiang Model was developed. This work is interesting and valuable. Through applying the proposed model, a good performance is achieved. But there are still some points that should be explained or revised before publication.

We appreciate your recognition of both the value of our work and the performance achieved by our proposed model. We have carefully considered all of your comments and responded to them in the subsequent specific comments section. We hope that these changes could effectively address your concerns.

**Specific comments:**

CC1-2. There are many parameters in the proposed TDD-XAJ model, the authors should state the method for parameter calibration. If the calibration is done as stated in Line 432 (calibrated manually), a lot of work should done.

Thanks for your comments. The TDD-XAJ model comprises 15 parameters, which is one parameter more than the original lumped Xinanjiang model (Zhao et al., 2023), with 11 hydrological parameters for runoff generation and 4 parameters (2 hydrological and 2 hydraulic) for runoff concentration. The two hydraulic parameters are surface roughness coefficient ($n_s$) and channel roughness coefficient ($n_c$). For $n_s$, values are assigned based on the land use type of each grid cell, with different land uses corresponding to different roughness coefficients, which are derived from existing literature (Miao et al., 2016; Perrini et al., 2024). For $n_c$, values are obtained from a roughness coefficient table for river channels (Arcement and Schneider, 1989).

Since this manuscript primarily focuses on the theoretical aspects of the TDD-XAJ model, we adopted uniform watershed-scale parameter values to simplify the research, thereby keeping the calibration workload manageable. In addition to manual calibration, uniform watershed-scale parameters or coefficients can also be determined using automated optimization algorithms, such as the Covariance Matrix Adaptation Evolution Strategy (CMA-ES) (Hansen et al., 2003). We plan to integrate this approach into the TDD-XAJ model in future developments.

To determine spatially distributed hydrological parameters, the process is generally based on spatially quantified data of watershed physical characteristics. This work is primarily carried out in two ways: the lookup table-based and the physical meaning-based method. For further details, please refer to RC2-8, which are also provided in Section S2 (Line 16-71 in the Supplement).

CC1-3. Line 423, Daily scale hydrological data were used in the study. I think the constructed model can be used for flood events simulation. Why don't you attempt to use sub-daily hydrological data?

The governing equations of the TDD-XAJ model are transformed into a system of ordinary differential equations after spatial discretization. Thus, the model can generate both instantaneous and time-averaged values of state variables and fluxes over a specified time interval through numerical integration, offering flexibility in model temporal resolution. As a result, the model is well-suited for daily-scale continuous simulations as well as flood event-based simulations (which usually use sub-daily data), as you mentioned in your comment. We have collected sufficiently long daily-scale hydrological data (spanning 2007-2019, totaling 13 year), but we did not gather enough sub-daily scale hydrological data. Consequently, this study relies on daily-scale hydrological data. In future work, we plan to collect additional sub-daily data as a foundation for exploring flood events simulation.

CC1-4. Line 456, the spatial distribution of the Oi has been zoomed into the upper left corner. I suggest the authors provide the spatial distribution of the entire area, and then zoom the upper left corner.

Thank you for your suggestion. This visualization has the advantage of presenting the full spatial distribution of $O_i$, ensuring data integrity while also highlighting the differences in the upper left corner. As RC1 and RC2 noted, the contrast between Figures 7b and 7d was unclear, and your recommended visualization approach effectively addresses this issue. As suggested, the revised Figure 7 is shown below:

Line 468-474 in tracked version:

[Figure]

**Figure 7.** The spatial distribution of surface water depth ($h_s$) and interflow storage ($O_i$) on the left-side hillslope of both single-slope (a-b) and double-slope (c-d) synthetic V-catchment test cases at the 60 minute mark. The state variable distributions shown are simulated using two-dimensional (2D) slope concentration methods. The corresponding results of 1D methods are identical to those obtained from the single-slope case simulated with 2D methods, regardless of the test case used. For a clear comparison, the spatial distribution of $O_i$ in the upper left corner has been zoomed.

CC1-5. In Table 2, average MAE statistics of model fluxes for a total of 500 parameter sets are provided using loosely coupled model. But the reference is the fully-coupled model. This cannot illustrate the better performance of fully-coupled model.

The main difference between the loosely-coupled (LC) and fully-coupled (FC) model lies in their numerical implementation frameworks. In the LC model, the difference-form equations for runoff generation from the original lumped XAJ model are directly adopted, which are derived based on the time interval of input force ($\Delta T$). However, for runoff concentration, the LC model uses differential-form equations; consequently, the generated runoff components (surface runoff, interflow, and groundwater runoff) are averaged over $\Delta T$ to determine the input intensities for the following runoff concentration equations. In contrast, the FC model adopts differential-form equations for both runoff generation and runoff concentration, solving both processes simultaneously as a system of ordinary differential equations (ODEs). In the FC model, the total amount of input force, rather than further calculated runoff components, is averaged over $\Delta T$.

We compared the LC model and FC model on single-slope and double-slope synthetic V-shaped watershed test cases using the same 500 parameter sets. An analytical solution exists for the total amount

of surface runoff ($R_s^*$). When the $\Delta T$ was set to 90 minutes, the average mean absolute error (MAE) for $R_s^*$ in the LC model was 4.57 mm, compared to $2.84 \times 10^{-4}$ mm for the FC model. As $\Delta T$ was reduced to 45 minutes and 15 minutes, the average MAE for $R_s^*$ in the LC model decreased to 1.21 mm and 0.14 mm, respectively; however, these errors remain significantly higher than those of the FC model.

For hillslope or channel outflow, no analytical solution is available, which makes direct comparison challenging. To address this, we evaluated the convergence of the LC model by progressively reducing $\Delta T$. The difference-form runoff generation equations used by the LC model have first-order temporal accuracy, and the FC model provides a high-order approximation of the analytical solution. Theoretically, as $\Delta T$ decreases, the results of the LC model should converge to those of the FC model. We used MAE to evaluate the consistency between the hillslope and channel outflow hydrographs simulated by the FC and LC models. Our numerical experiment showed that the average MAE decreases as $\Delta T$ is reduced, indicating that the LC model's results converge toward those of the FC model. Furthermore, significant numerical errors could be observed in the LC model ($\Delta T$=90 minutes), whether benchmarked against the LC model ($\Delta T$=15 minutes) or the FC model ($\Delta T$=90 minutes). In the single-slope test case, when using the LC model ($\Delta T$=15 minutes) as the benchmark, the average MAE for channel and hillslope outflow are 1.82 mm and 1.64 mm, respectively, whereas when using the FC model ($\Delta T$=90 minutes) as the benchmark, the average MAE are 1.85 mm and 1.68 mm. In the double-slope test case, when using the LC model ($\Delta T$=15 minutes) as the benchmark, the average MAE for channel and hillslope outflow are 1.83 mm and 1.68 mm, respectively, while benchmarked against the FC model ($\Delta T$=90 minutes) yields average MAE of 1.87 mm and 1.72 mm. Additionally, the outflow hydrograph simulated by the LC model exhibits non-physical steady states and inflection points, which indicate its potential limitations in capturing transient behaviors.

Overall, when an analytical solution is available, the error of the LC model is several orders of magnitude higher than that of the FC model. In cases without an analytical solution, as $\Delta T$ decreases, results of the LC model converge to those of the FC model. Furthermore, non-physical steady states and inflection points are observed in the hydrograph simulated by the LC model. Consequently, the FC model is considered to performs better in numerical simulations.

CC1-6. Line 505, the 2 values cannot be found in Table 2.

Thank you for pointing this out. To keep Table 2 concise, we initially approximated the very small values ($4.19 \times 10^{-3}$ and $2.84 \times 10^{-4}$) as 0 and used "~0.00" to maintain alignment in the column. The exact values are provided in the following lines for clarity. To avoid any misunderstanding, we have now included the values in scientific notation directly in Table 2. The revised Table is shown below:

**Table 2.** MAE statistics of model fluxes in numerical implementation comparison experiment.

| Model | $\Delta T$ (min) | Statistics | MAE (mm) | MAE (m³/s) | | | |
|---|---|---|---|---|---|---|---|
| | | | $R_s^*$ | Hillslope outflow | | Channel outflow | |
| | | | | Single-slope | Double-slope | Single-slope | Double-slope |
| Loosely-coupled | 90 | Max | 5.93 | 2.19 | 2.20 | 2.01 | 2.06 |
| | | Average | 4.57 | 1.85 | 1.87 | 1.68 | 1.72 |
| | 45 | Max | 2.01 | 1.60 | 1.62 | 1.31 | 1.41 |
| | | Average | 1.21 | 0.81 | 0.82 | 0.70 | 0.73 |
| | 15 | Max | 0.33 | 0.56 | 0.61 | 0.36 | 0.39 |
| | | Average | 0.14 | 0.16 | 0.17 | 0.11 | 0.12 |
| Fully-coupled | 90 | Max | $4.19\times10^{-3}$ | —[a] | — | — | — |
| | | Average | $2.84\times10^{-4}$ | — | — | — | — |

a. The results of the fully-coupled model are used as references to calculate the MAE values for hillslope and channel outflow, so the corresponding value is empty.

CC1-7. The simulation in the Tunxi watershed was only compared with 1 previous study in the same watershed. Is it possible to compare the results with previous research using other lumped or distributed models in the same or adjacent watershed?

Thank you for your suggestion. For the present study, we focused on comparing our simulation in the Tunxi watershed with a well-documented previous study in the same area to ensure consistency in benchmark data. The primary focus of this study is on the theoretical aspect of the TDD-XAJ model. We acknowledge that further validation, including comparisons with other models, remains necessary. However, difficulties in data availability, model structure, and parameterization among studies made such a comparison challenging at this stage. While inter-model comparison was not implemented, validation targeting a hydrological station within the Tunxi Watershed was executed to strengthen model performance evaluation.

We introduced the Yuetan hydrological station—a station within the Tunxi watershed (Figure 5), and compared its simulation results with observed data. Details of performance metrics are provided in Table 4. As shown in Table 4, the average values of the Nash-Sutcliffe efficiency (NSE), Kling-Gupta efficiency (KGE), the absolute flood volume relative error (|FVRE|), and the coefficient of determination ($R^2$) for Yuetan station (across all years) are 0.83, 0.78, 6.2%, and 0.86, respectively. The corresponding values for Tunxi station are 0.87, 0.80, 6.7%, and 0.90. In summary, these metrics indicate that the TDD-XAJ model provides robust streamflow simulations at both stations in the Tunxi watershed. The revisions are detailed below:

[Figure]

**Figure 5.** Location and gauging station distribution of the Tunxi watershed (a), and the spatial discretization of the watershed, including channel and non-channel cells (b).

Line 574-593 in tracked version:

**Table 4.** Annually evaluated simulation performance metrics of the TDD-XAJ model in the Tunxi watershed.

| Period | Year | Tunxi station | | | | Yuetan station | | | |
|---|---|---|---|---|---|---|---|---|---|
| | | NSE | KGE | FVRE(%) | $R^2$ | NSE | KGE | FVRE(%) | $R^2$ |
| | 2008 | 0.94 | 0.91 | -8.26 | 0.94 | 0.90 | 0.89 | -1.33 | 0.90 |
| | 2009 | 0.88 | 0.90 | -6.95 | 0.88 | 0.82 | 0.80 | -16.43 | 0.83 |
| | 2010 | 0.85 | 0.78 | -16.88 | 0.90 | 0.82 | 0.78 | -18.99 | 0.87 |
| Calibration | 2011 | 0.89 | 0.78 | 7.53 | 0.89 | 0.78 | 0.76 | -3.46 | 0.80 |
| | 2012 | 0.82 | 0.84 | -7.64 | 0.83 | 0.74 | 0.80 | -5.97 | 0.74 |
| | 2013 | 0.87 | 0.80 | -10.75 | 0.92 | 0.88 | 0.79 | -3.99 | 0.92 |
| | 2014 | 0.88 | 0.79 | 0.20 | 0.91 | 0.84 | 0.75 | 0.72 | 0.87 |
| | 2015 | 0.85 | 0.77 | -9.92 | 0.92 | 0.85 | 0.80 | -8.48 | 0.88 |
| | 2016 | 0.88 | 0.78 | -6.92 | 0.92 | 0.86 | 0.79 | -4.37 | 0.89 |
| Validation | 2017 | 0.88 | 0.76 | 1.62 | 0.92 | 0.84 | 0.72 | 4.66 | 0.86 |
| | 2018 | 0.87 | 0.77 | 1.58 | 0.89 | 0.87 | 0.78 | -2.75 | 0.90 |
| | 2019 | 0.85 | 0.74 | -2.24 | 0.89 | 0.79 | 0.74 | -3.70 | 0.81 |

For Tunxi station at the outlet of Tunxi watershed, Table 4 indicates that the values of the FVRE metric are all within ±20 %, with the absolute values of the FVRE (|FVRE|) averaging 8.3% and 4.5% for the calibration and validation period, respectively. In terms of hydrograph evaluation, the average values of NSE and KGE are 0.88 and 0.83 for the calibration period and 0.87 and 0.76 for the validation period, which is slightly better for the calibration period than for the validation period. The minimum value of $R^2$ is 0.83 for all years, and the average value for all years is 0.90. In a direct comparison, Tong (2022) conducted a similar daily simulation in the same watershed using the GXAJ model, reporting average NSE and |FVRE| values of 0.85 and 11.0% between 2008 and 2017, respectively. In contrast, the TDD-XAJ model achieved average values of 0.87 for NSE and 7.7% for |FVRE| in the same period. For Yuetan station within Tunxi watershed, Table 4 shows that FVRE metric values remain within ±20 %. The

average |FVRE| is 7.3% and 4.8% for the calibration and validation periods, respectively. Meanwhile, the average value of NSE is 0.82 for the calibration period and 0.84 for the validation period, and the average KGE is 0.80 and 0.77 for calibration period and validation period, respectively. The average $R^2$ Across all years is 0.86. Fig. 9 provides an example of the simulated hydrograph at Tunxi and Yuetan station of the TDD-XAJ model in 2008.

[Figure]

**Figure 9.** The simulated hydrograph at Tunxi (a) and Yuetan (b) station of the Tunxi watershed in 2008 using the TDD-XAJ model.

**References mentioned in the response**

Arcement, G. J. and Schneider, V. R.: Guide for selecting Manning's roughness coefficients for natural channels and flood plains, 2339, https://doi.org/10.3133/wsp2339, 1989.

Beven, K., 2002. Towards an alternative blueprint for a physically based digitally simulated hydrologic response modelling system. Hydrol. Process., 16(2): 189-206.

Beven, K. J., Kirkby, M. J., Freer, J. E., and Lamb, R.: A history of TOPMODEL, Hydrol. Earth Syst. Sci., 25, 527-549, https://doi.org/10.5194/hess-25-527-2021, 2021.

Bisht, G. and Riley, W. J.: Development and verification of a numerical library for solving global terrestrial multiphysics problems, J. Adv. Model. Earth Syst., 11, 1516-1542, https://doi.org/10.1029/2018MS001560, 2019.

Chen, J., Chen, J., Liao, A., Cao, X., Chen, L., Chen, X., He, C., Han, G., Peng, S., Lu, M., Zhang, W., Tong, X., and Mills, J.: Global land cover mapping at 30m resolution: A POK-based operational approach, ISPRS-J. Photogramm. Remote Sens., 103, 7-27, https://doi.org/10.1016/j.isprsjprs.2014.09.002, 2015.

Chen, L., Deng, J., Yang, W., and Chen, H.: Hydrological modelling of large-scale karst-dominated basin using a grid-based distributed karst hydrological model, J. Hydrol., 628, 130459, https://doi.org/10.1016/j.jhydrol.2023.130459, 2024.

FAO and IIASA: Harmonized world soil database (version 2.0) [dataset], https://www.fao.org/soils-portal/data-hub/soil-maps-and-databases/harmonized-world-soil-database-v20/en/, 2023.

Fang, Y. H., Zhang, X., Corbari, C., Mancini, M., Niu, G. Y., and Zeng, W.: Improving the Xin'anjiang hydrological model based on mass–energy balance, Hydrol. Earth Syst. Sci., 21, 3359-3375, https://doi.org/10.5194/hess-21-3359-2017, 2017.

Hansen, N., Müller, S. D., and Koumoutsakos, P.: Reducing the Time Complexity of the Derandomized Evolution Strategy with Covariance Matrix Adaptation (CMA-ES), Evol. Comput., 11, 1-18, https://doi.org/10.1162/106365603321828970, 2003.

Höge, M., Scheidegger, A., Baity-Jesi, M., Albert, C., and Fenicia, F.: Improving hydrologic models for predictions and process understanding using neural ODEs, Hydrol. Earth Syst. Sci., 26, 5085-5102, https://doi.org/10.5194/hess-26-5085-2022, 2022.

Lam, S. K., Pitrou, A., and Seibert, S.: Numba: a LLVM-based Python JIT compiler, in: Proceedings of the Second Workshop on the LLVM Compiler Infrastructure in HPC, Austin, Texas, 7, https://doi.org/10.1145/2833157.2833162, 2015.

Li, B., Sun, T., Tian, F., Tudaji, M., Qin, L., and Ni, G.: Hybrid hydrological modeling for large alpine basins: a semi-distributed approach, Hydrol. Earth Syst. Sci., 28, 4521-4538, https://doi.org/10.5194/hess-28-4521-2024, 2024.

Liu, J., Chen, X., Zhang, J., and Flury, M.: Coupling the Xinanjiang model to a kinematic flow model based on digital drainage networks for flood forecasting, Hydrol. Process., 23, 1337-1348, https://doi.org/10.1002/hyp.7255, 2009.

Maidment, D. R., Olivera, F., Calver, A., Eatherall, A., and Fraczek, W.: Unit hydrograph derived from a spatially distributed velocity field, Hydrol. Process., 10, 831-844, https://doi.org/10.1002/(SICI)1099-1085(199606)10:6<831::AID-HYP374>3.0.CO;2-N, 1996.

Miao, Q., Yang, D., Yang, H., and Li, Z.: Establishing a rainfall threshold for flash flood warnings in China's mountainous areas based on a distributed hydrological model, J. Hydrol., 541, 371-386, https://doi.org/10.1016/j.jhydrol.2016.04.054, 2016.

Murphy, B., Yurchak, R., and Müller, S.: GeoStat-Framework/PyKrige (1.7.2), Zenodo [code], https://doi.org/10.5281/zenodo.11360184, 2024.

Overton, D. E. and Brakensiek, D. L.: A kinematic model of surface runoff response, in: Proceedings of the Wellington Symposium, Wellington, New Zealand, 1970.

Ouyang, W., Ye, L., Chai, Y., Ma, H., Chu, J., Peng, Y., and Zhang, C.: A differentiable, physics-based hydrological model and its evaluation for data-limited basins, J. Hydrol., 649, 132471, https://doi.org/j.jhydrol.2024.132471, 2025.

Perrini, P., Cea, L., Chiaravalloti, F., Gabriele, S., Manfreda, S., Fiorentino, M., Gioia, A., and Iacobellis, V.: A runoff-on-grid approach to embed hydrological processes in shallow water models, Water Resour. Res., 60, e2023WR036421, https://doi.org/10.1029/2023WR036421, 2024.

Seibert, J., Bergström, S., and Sveriges, L.: A retrospective on hydrological catchment modelling based on half a century with the HBV model, Hydrol. Earth Syst. Sci., 26, 1371-1388, https://doi.org/10.5194/hess-26-1371-2022, 2022.

Shaw, J., Kesserwani, G., Neal, J., Bates, P., and Sharifian, M. K.: LISFLOOD-FP 8.0: the new discontinuous Galerkin shallow-water solver for multi-core CPUs and GPUs, Geosci. Model Dev., 14, 3577-3602, https://doi.org/10.5194/gmd-14-3577-2021, 2021.

Shi, G., Sun, W., Shangguan, W., Wei, Z., Yuan, H., Li, L., Sun, X., Zhang, Y., Liang, H., Li, D., Huang, F., Li, Q., and Dai, Y.: A China dataset of soil properties for land surface modelling (version 2, CSDLv2), Earth Syst. Sci. Data, 17, 517-543, https://doi.org/10.5194/essd-17-517-2025, 2025.

Stacke, T. and Hagemann, S.: HydroPy (v1.0): a new global hydrology model written in Python, Geosci. Model Dev., 14, 7795-7816, https://doi.org/10.5194/gmd-14-7795-2021, 2021.

Su, B., Kazama, S., Lu, M., and Sawamoto, M.: Development of a distributed hydrological model and its application to soil erosion simulation in a forested catchment during storm period, Hydrol. Process., 17, 2811-2823, https://doi.org/10.1002/hyp.1435, 2003.

Tong, B.: Fine-scale rainfall-runoff processes simulation using grid Xinanjiang (grid-XAJ) model, Hohai University, Nanjing, Jiangsu, 2022.

Yao, C., Li, Z., Yu, Z., and Zhang, K.: A priori parameter estimates for a distributed, grid-based Xinanjiang model using geographically based information, J. Hydrol., 468-469, 47-62, https://doi.org/10.1016/j.jhydrol.2012.08.025, 2012.

Zhao, J., Duan, Y., Hu, Y., Li, B., and Liang, Z.: The numerical error of the Xinanjiang model, J. Hydrol., 619, 129324, https://doi.org/10.1016/j.jhydrol.2023.129324, 2023.

Zhao, R.: The Xinanjiang model applied in China, J. Hydrol., 135, 371-381, https://doi.org/10.1016/0022-1694(92)90096-E, 1992.

Finally, we would like to once again thank the Editor and all Reviewers for their thorough review of our manuscript. If there are any questions, suggestions, or discussions, please feel free to contact us.